# Distinctive aerosol-cloud-precipitation interactions in marine boundary layer clouds from the ACE-ENA and SOCRATES aircraft field campaigns

Xiaojian Zheng[1,a], Xiquan Dong[1], Baike Xi[1], Timothy Logan[2] and Yuan Wang[3]

[1]Department of Hydrology and Atmospheric Sciences, University of Arizona, Tucson, AZ, USA

[2]Department of Atmospheric Sciences, Texas A&M University, College Station, TX, USA

[3]Department of Earth System Sciences, Stanford University, Stanford, CA, USA

[a]Now at: Environmental Science Division, Argonne National Laboratory, Lemont, IL, USA

**Correspondence**: Xiquan Dong (xdong@arizona.edu)

**Abstract.** The aerosol-cloud-precipitation interactions within the cloud-topped Marine Boundary Layer (MBL) are examined using aircraft in-situ measurements from Aerosol and Cloud Experiments in the Eastern North Atlantic (ACE-ENA) and Southern Ocean Clouds Radiation Aerosol Transport Experimental Study (SOCRATES) field campaigns. SOCRATES clouds exhibit a larger number concentration and smaller cloud droplet effective radius (148.3 $cm^{-3}$ and 8.0 μm) compared to ACE-ENA summertime (89.4 $cm^{-3}$ and 9.0 μm) and wintertime clouds (70.6 $cm^{-3}$ and 9.8 μm). The ACE-ENA clouds, especially during the winter, feature stronger drizzle formation via droplet growth through enhanced collision-coalescence, attributed to a relatively cleaner environment and deeper cloud layer. Furthermore, the Aerosol-Cloud Interaction (ACI) indices from the two aircraft field campaigns exhibit distinct sensitivities, indicating different cloud microphysical responses to aerosols. The ACE-ENA winter season features relatively fewer aerosols, which are more likely activated into cloud droplets under the conditions of sufficient water vapor availability and strong turbulence. The enriched aerosol loading during ACE-ENA summer and SOCRATES generally leads to smaller cloud droplets competing for the

limited water vapor and exhibiting a stronger ACI. Notably, the precipitation susceptibilities are stronger during the ACE-ENA than during the SOCRATES campaigns. The in-cloud drizzle behavior significantly alters sub-cloud cloud condensation nuclei (CCN) budgets through the coalescence-scavenging effect, and in turn, impact the ACI assessments. The results of this study can enhance the understanding and aid in future model simulation and assessment of the aerosol-cloud interaction.

## 1. Introduction

Marine boundary layer (MBL) clouds substantially impact the Earth's climate system (Dong and Minnis, 2022). Sustained by large-scale subsidence and cloud-top longwave radiative cooling, MBL clouds, typically located beneath the temperature inversion at the MBL top, persistently reflect the incoming solar radiation and modulate the radiative balance (Lilly, 1968; Albrecht et al., 1995; Wood et al., 2015; Dong et al., 2023). The climatic significance of MBL cloud radiative effects, which remains largely uncertain (IPCC, 2022), is closely linked to cloud microphysical properties that are substantially influenced by surrounding aerosol conditions (Chen et al., 2014; Feingold and McComiskey, 2016). Observational evidence demonstrates that cloud microphysical responses to aerosols, defined as the aerosol-cloud interaction (ACI), can be typically viewed as decreased cloud droplet effective radii ($r_c$) and increased number concentrations ($N_c$) with more aerosol intrusion under conditions of comparable cloud water content (Feingold and McComiskey, 2016). The ACIs have been extensively investigated by different observational platforms, such as aircraft (Hill et al., 2009; Diamond et al., 2018; Gupta et al., 2022), ground-based and satellite observations (Painemal et al., 2020; Zhang et al., 2022; Zheng et al., 2022a), and model simulations (Wang et al., 2020; Christensen et al., 2023) over different maritime regions like the southeast Pacific (Painemal and Zuidema, 2011), northeast Pacific (Braun et al., 2018), southeast Atlantic (Gupta et al., 2022), and eastern North Atlantic (Zheng et al., 2022a).

Furthermore, a larger number of small cloud droplets can sometimes extend cloud longevity and spatial coverage and modulate the precipitation processes in the MBL clouds, reflecting the cloud adjustments to aerosol disturbances (Albrecht, 1989; Bellouin et al., 2020). Precipitation, particularly in the form of drizzle, is common in MBL clouds (Wood et al., 2015; Wu et al., 2020), and the turbulence forced by stratocumulus cloud-top radiative cooling can increase the cloud liquid water path and contribute to drizzle production (Ghate et al., 2019, 2021). The drizzle formation and growth processes are deeply entwined with the MBL aerosols and dynamics. Frequent aerosol intrusions in the MBL have been found to lower the efficiency of collision-coalescence which results in the suppression of precipitation frequency and strength. Such phenomenon can be quantified and assessed via the cloud precipitation susceptibility (Feingold and Seibert, 2009; Lu et al., 2009; Sorooshian et al., 2009; Duong et al., 2011). The assessments of precipitation susceptibility are examined to be under the influences of methodology (Terai et al., 2012), cloud morphology (Sorooshian et al., 2009; Jung et al., 2016), ambient aerosol concentrations (Duong et al., 2011; Jung et al., 2016; Gupta et al., 2022), and cloud thickness (Terai et al., 2012; Jung et al., 2016; Gupta et al., 2022). The in-cloud turbulence and wind shear can effectively enhance collision-coalescence efficiency, stimulate drizzle formation and growth, and consequently lead to enhanced precipitation rate and amount (Chen et al., 2011; Wu et al., 2017). Cloud-top entrainment of dryer and warmer air can potentially deplete small cloud droplets and shrink large droplets via evaporation, thereby impacting cloud-top microphysical processes depending on the homogeneous or inhomogeneous mixing regimes (Lehmann et al., 2009; Jia et al., 2019).

Conversely, precipitation has been shown to exert a substantial influence on the MBL aerosol and cloud condensation nuclei (CCN) budget through the coalescence-scavenging effect. The coalescence-scavenging refers to the process in which cloud or drizzle droplets, containing aerosol particles inside, merge with each other. Upon the collision-coalescence of cloud droplets, the dissolved aerosol masses within the cloud droplets also collide and merge into a larger aerosol core, leading to larger aerosol particles upon droplet evaporation. The sub-cloud aerosols are then replenished into the cloud layer,

experiencing growth within the cloud through cloud and drizzle droplet collision-coalescence and subsequently falling and evaporating outside the cloud again. Eventually, the residual aerosols undergoing this cloud-processing cycle will gradually decrease in number concentration and increase in size (Flossmann et al., 1985; Feingold et al., 1996; Hudson and Noble, 2020; Hoffmann and Feingold, 2023). In addition, the drizzle drops, upon falling out of the cloud base, can result in net reductions in sub-cloud aerosols and CCN budgets via precipitation scavenging processes (Wood, 2006; Zheng et al., 2022b). Quantitative estimates of these effects remain ambiguous and inconclusive, which are subject to multiple factors such as aerosol physicochemical characteristics, cloud morphology, and MBL dynamics and thermodynamics conditions (Sorooshian et al., 2009; Duong et al., 2011; Diamond et al., 2018; Brunke et al., 2022). Thus, more studies on the aforementioned processes regarding MBL aerosols and clouds over different maritime regions are warranted to pursue an in-depth understanding of aerosol-cloud-precipitation interactions (ACPIs).

The Eastern North Atlantic (ENA) stands as a desirable region for exploring MBL clouds in the mid-latitude, with Graciosa Island in the Azores (39.09°N, 28.03°W) representing a focal point for studies of ACPIs. Located between the mid-latitude and subtropical climate zones, Graciosa Island is subject to the meteorological influence of both the Icelandic Low and the Azores High, and the influence of aerosols ranging from pristine marine air masses to those heavily influenced by continental emissions from North America and Northern Europe (Logan et al., 2014; Wood et al., 2015; Wang et al., 2020). Addressing the need for sustained research into the MBL clouds, the recent Aerosol and Cloud Experiments in the Eastern North Atlantic (ACE-ENA) aircraft campaign (J. Wang et al., 2022) was conducted in the summer (June and July) 2017 (ACE-ENA Sum) and winter (January and February) 2018 (ACE-ENA Win). During these two intensive operation periods (IOPs) of ACE-ENA, the research aircraft accrued abundant in-situ measurements of aerosols, clouds, and drizzle properties, providing invaluable resources for studying the ACI and ACPI processes. During the summer, the Azores is located at the eastern part of the high-pressure system, while during the winter, the center of the Azores high

shifts to the eastern Atlantic and is primarily located directly over the Azores (Mechem et al., 2018; J. Wang et al., 2022). Furthermore, both ACE-ENA Sum and ACE-ENA Win IOPs of featured anomalously strong high-pressure systems, compared to the 20-year climatology, as shown in Figure S1. This meteorological pattern is favorable to the prevailing and persistent stratocumulus clouds observed during ACE-ENA, especially during the winter IOP, where the enhanced large-scale subsidence can lead to stronger and sharper temperature inversions above the stratocumulus-topped MBL (Rémillard and Tselioudis, 2015; Jensen et al., 2021; Marcovecchio et al., 2022). The ACE-ENA Sum is characterized by anomalously low MBL heights and substantial MBL decoupling (Miller et al., 2021; J. Wang et al., 2022). The winter IOP was under the frequent impacts of the mid-latitude systems and prevalently featured precipitation-generated cold pools, where evaporative cooling alters the thermodynamical structure of the MBL, sustains and enhances turbulence mixing, hence contributes to dynamical perturbations that can influence the behavior of the MBL (Terai and Wood, 2013; Zuidema et al., 2017; Jenson et al., 2021; J. Wang et al., 2022; Smalley et al., 2024). In recent years, many observational studies based on ACE-ENA data have focused on the seasonal contrasts of the aerosol distributions and sources (Y. Wang et al., 2021b; Zawadowicz et al., 2021), the cloud and drizzle microphysics vertical distributions (Wu et al., 2020a; Zheng et al., 2022b), and the impacts of MBL conditions on the cloud structure and morphology (Jensen et al., 2021). However, they seldom analyze the comprehensive interactions between aerosol, clouds, and precipitation.

Over the Southern Ocean (SO), the Southern Ocean Clouds Radiation Aerosol Transport Experimental Study (SOCRATES) field campaign (McFarquhar et al., 2021) was conducted during the austral summer (January and February 2018), which marks another valuable piece of the MBL cloud research. The SO, being one of the cloudiest regions globally, is predominantly influenced by naturally produced aerosols originating from oceanic sources due to its remoteness, where the anthropogenic and biomass burning aerosols exert minimal influence over the region (McCoy et al., 2021; Sanchez et al., 2021; Twohy et al., 2021; Zhang et al., 2023). The aerosol budget in this region is primarily shaped by

biological aerosols, which nucleate from the oxidation products of dimethyl sulfide (DMS) emissions, as well as by sea spray aerosols. Hence, the SO provides an unparalleled natural laboratory for discerning the influence of these natural aerosol emissions on the MBL clouds under a pre-industrial natural environment. The summertime SO region, particularly near the SOCRATES focus area, is characterized by more frequently closed-cell mesoscale cellular convection structures (Danker et al., 2022; Lang et al., 2022). Furthermore, the MBL clouds over the SO predominantly consist of supercooled liquid water droplets, which coexist with mixed- and ice-phase processes (Y. Wang et al., 2021a; Xi et al., 2022), while the precipitation phases are examined to be primarily dominated by liquid hydrometeors (Tansey et al., 2022; Kang et al., 2024). The in-situ measurements collected from SOCRATES have cultivated many studies on aerosols, clouds, and precipitation over the SO using both in-situ measurements and model simulations (McCoy et al., 2020; Altas et al., 2021; D'Alessandro et al., 2021), and provides an opportunity to study the liquid cloud processes under a colder climate. As shown in Figure S1c, our composite analysis of the synoptic pattern shows that the SOCRATES cloud cases used in this study are located ahead of the anomalously strong thermal ridge and behind the thermal trough, providing an environment favorable to closed cellular MBL cloud structures (McCoy et al., 2017; Lang et al., 2022). Since the region of selected SOCRATES cloud cases crosses a larger latitudinal zone and is under more consistent influence of mid-latitude cyclone systems than the ACE-ENA during the summer IOP, the cloud sampling periods used in this study majority reside in the closed-cell MBL stratocumulus decks.

The cloud cases selected from the ACE-ENA and SOCRATES campaigns share similar cloud morphology (stratocumulus) while experiencing different aerosol sources and meteorological conditions. A synergistic approach that compares data from these different field campaigns can provide valuable insights to the community regarding the dominant physical processes of the interactions between aerosols, clouds, and precipitation under the influence of different MBL dynamic and thermodynamic conditions. This study targets the similarities and differences in the MBL aerosol, cloud, and drizzle properties, their distribution and evolution, and more appealingly, the ACIs and ACPIs between the two campaigns. The

data and methods used in this study are introduced in section 2. The aerosol and CCN properties in the above- and sub-cloud regimes, as well as the vertical distributions of MBL cloud and drizzle properties, are examined in section 3. The ACI, precipitation susceptibility and drizzle impacts on the sub-cloud aerosols and CCN (ACPI) are discussed in section 4. Finally, the results are summarized, and the importance of this study is discussed in section 5.

## 2. Data and methods

### 2.1 Cloud and drizzle properties

The in-situ measurements of MBL cloud properties are temporally synchronized to 1 Hz resolution, corresponding to approximately 100 m (5 m) of horizontal (vertical) sampling. The sampling locations of the selected cases are indicated by the white dots in Figure S1. The Fast Cloud Droplet Probe (FCDP) onboard the aircraft during ACE-ENA can detect droplets with diameter ($D_p$) ranging from 1.5 µm to 50 µm, with the size bins of the probe between 1 and 3 µm (Glienke and Mei, 2020). SOCRATES used a similar CDP to measure droplets from 2 µm to 50 µm at a 2 µm probe size bin width. Both ACE-ENA and SOCRATES leverage the Two-Dimensional Stereo Particle Imaging Probe (2DS) to discern droplets with diameters from 5 µm to 1280 µm (Lawson et al., 2006; Glienke and Mei, 2019). The 2DS in-situ measurements are used as additional screening to eliminate the ice particles with diameters larger than 200 µm. Moreover, the University of Washington Ice–Liquid Discriminator product, which is a Machine-learning-based single-particle phase classification of the 2DS images (Atlas et al., 2021), is used to identify small ice crystals when available. Through these three datasets, we can tease out the ice-dominated period to the highest extent possible and focus on the liquid cloud processes and ACI during SOCRATES (Wang et al., 2021).

Although these in-situ measurements can provide "ground-truth" datasets, their uncertainties must be properly analyzed and data quality must be controlled before being applied to scientific studies. The uncertainties of FCDP in sizing and concentration are approximately 30% and 20%, respectively

(Baumgardner et al., 2017). Considering the significant uncertainty in the concentration of smaller particles from a photodiode probe such as 2DS (Baumgardner & Korolev, 1997; Wang et al., 2021), a diameter of 40 μm is used as the demarcation line between cloud droplets and drizzle drops (Wood et al., 2005). Then droplet number concentrations in the overlapping size bin between FCDP and 2DS are redistributed assuming a gamma distribution, thereby a complete size spectrum of cloud and drizzle can be merged from FCDP and 2DS measurements. Hence, the cloud and drizzle microphysical properties can be calculated.

The cloud droplet number concentration ($N_c$) is given by:

$$N_c = \int_2^{40} n(D_p)\, dD_p, \tag{1}$$

The cloud droplet effective radius ($r_c$, Hansen and Travis, 1974) is given by:

$$r_c = \frac{\int_2^{40} r_p^3\, n(D_p)\, dD_p}{\int_2^{40} r_p^2\, n(D_p)\, dD_p}, \tag{2}$$

The cloud liquid water content ($LWC_c$) can be calculated by:

$$LWC_c = \frac{4}{3}\pi\rho_w \int_2^{40} D^3\, n(D_p)\, dD_p, \tag{3}$$

where $\rho_w$ is water density.

Similarly, the drizzle drop number concentration ($N_d$) and liquid water content ($LWC_d$) can be calculated using the size distribution from 40 μm to 1280 μm. Particularly, the drizzle mean mass diameter ($D_{mmd}$) is given by:

$$D_{mmd} = \left(\frac{\int_{40}^{1280} D_p^3\, n(D_p)\, dD_p}{\int_{40}^{1280} n(D_p)\, dD_p}\right)^{1/3}, \tag{4}$$

This quantity is chosen because the $D_{mmd}$ denotes the diameter of average mass (the third-moment average) of the drizzle size distribution, which provides the link between the number concentration and the mass concentration of drizzle droplets in a sample (Hinds, 1999).

Adapting the method in Zheng et al. (2022b), the cloud base precipitation rate ($R_{CB}$) is given by:

$$R_{CB}(mm/hr) = 6\pi * 10^{-4} \int_{40\mu m}^{1280\mu m} D_{p,mm}^{3}\, n(D_{p,mm})\, U_{\infty}(D_{p,mm})\, dD_{p,mm}, \quad (5)$$

in order to match the unit conversion, the $D_{p,mm}$ is diameter in unit of mm, $n(D_{p,mm})$ is drizzle number concentration in every size bin with a unit of # m$^3$ mm$^{-1}$, and $U_{\infty}(D_{p,mm})$ is terminal velocity in given size bin, which is calculated from the full Reynolds number theory as in Pruppacher and Klett (2010).

The combined threshold of $N_c > 5$ cm$^{-3}$ and $LWC_c > 0.01$ g m$^{-3}$ is used for determining the valid cloud samples and cloud boundaries (Wood, 2005; Zheng et al., 2022b). The complete cloud vertical profiles from sub-cloud to the above-cloud are selected during the ACE-ENA and SOCRATES IOPs, in which the flight strategy includes sawtooth and spiral cloud transects and ramping cloud sampling. The precipitation conditions are determined by whether samples of $N_d > 0.001$ cm$^{-3}$ exists below the cloud base height. In total, the selected numbers of cloud (precipitating cloud) profiles are 18 (13), 26 (13), and 28 (24) for ACE-ENA Sum and ACE-ENA Win, and SOCRATES, respectively. The detailed selected cloud profiles, with their cloud-base heights ($z_t$), cloud-top heights ($z_b$) and cloud thicknesses ($H_c = z_t - z_b$) are listed in Table S1, along with the cloud profile macrophysics.

Furthermore, the assessments of ACI are significantly impacted by the MBL dynamic and thermodynamic conditions. Jones et al. (2011) suggested that the MBL would be in a well-mixed and coupled condition when the difference in liquid water potential temperature ($\theta_L$) and total water mixing ratio ($q_t$) between the bottom of MBL and the inversion layer are less than 0.5 K and 0.5 g kg$^{-1}$, respectively. The cases selected for this study feature both coupled and decoupled MBL conditions, particularly during ACE-ENA Sum, which is characterized by anomalously low MBL heights and substantial MBL decoupling. Previous studies found that under decoupled conditions the aerosols, CCN, and moisture sources near the surface are disconnected from the cloud layer aloft, hence exerting a much less effective impact on cloud microphysics (Zheng et al., 2022a; Christensen et al., 2023; Su et al., 2024). Therefore, we adapt and modify the metric in Jones et al. (2011) to calculate the sub-cloud coupled layer, in order to quantify the degree to which aerosols and CCN measured sub-cloud are in a well-mixed state

and can represent the actual interaction (or contact) with the cloud layer. In this study, the $q_t$ and $\theta_L$ at the cloud base are calculated, and then their vertical variations are examined starting from the altitude of cloud base ($z_b$) and looking downward. As such, the coupled point height ($z_{cp}$) is defined as the altitude where the downward vertical changes in $q_t$ and $\theta_L$ exceed 0.5 K and 0.5 g kg$^{-1}$, respectively. Hence, the coupled layer thickness ($H_{cp} = z_t - z_{cp}$) is defined as the layer between the cloud top height ($z_t$) and coupled point height ($z_{cp}$), hence the selection of the aerosols and CCN within the below-cloud part of the coupled layer can be viewed as in contact with the cloud. An example of the coupled layer identification is shown in Figure S2. Therefore, the degree of MBL decoupling ($D_{cp}$) can be quantified as the ratio of the coupled point height ($z_{cp}$) to the cloud base height ($z_b$) , where $D_{cp} = z_{cp}/z_b$. As shown in Table S1, the ACE-ENA Sum feature with highest degree of decoupling (averaged $D_{cp}$ = 0.504), compared to the ACE-ENA Win ($D_{cp}$ = 0.370) and SOCRATES ($D_{cp}$ = 0.277).

### 2.2 Aerosol properties

The total aerosol number concentrations ($N_a$) from ACE-ENA and SOCRATES are measured by the airborne Condensation Particle Counter (CPC) models 3772 and 3760A, which count the number of aerosols with diameter ($D_p$) larger than 3 nm and 11 nm, respectively (Kuang and Mei, 2019; SOCRATES Low Rate Data, 2022). Additionally, the Passive Cavity Aerosol Spectrometer (PCASP) onboard the ACE-ENA aircraft is capable of sizing the aerosol with $D_p$ ranging from 0.1 μm to 3.2 μm (Goldberger, 2020). The ultra-high sensitivity aerosol spectrometer (UHSAS) measures the size-resolved aerosol distribution from 0.06 μm to 1.0 μm during SOCRATES (Uin, 2016). Therefore, the number concentrations of accumulation mode aerosols ($N_{ACC}$, 0.1 μm-1 μm) can be discerned from the PCASP and UHSAS aerosol size distributions. Aitken mode aerosols ($N_{Ait}$, < 0.1 μm) from ACE-ENA are retrieved by the fast integrated mobility spectrometer (FIMS), which can size the aerosol down to 9 nm (Olfert et al., 2008), while the $N_{Ait}$ from SOCRATES is limited to 0.06 μm – 0.1 μm due to the limitation

of UHSAS. As for the CCN measurements, the ACE-ENA utilized the Dual-Column CCN Counter at two constant supersaturation levels of 0.15% and 0.35% (Uin and Mei, 2019), while the CCN number concentration ($N_{CCN}$) during SOCRATES was measured under various supersaturation levels from 0.06% to 0.87% using a scanning CCN counter (Roberts and Nenes, 2005). In this study, $N_{CCN}$ at 0.35% supersaturation ($N_{CCN0.35\%}$) is used to ensure a direct comparison between ACE-ENA and SOCRATES. The aerosol measurements are in the temporal resolution of 1 Hz. Note that the aerosol and CCN data are quality-controlled by removing the data point where the $N_c + N_d$ greater than 5 cm$^{-3}$ or $N_d$ greater than 0.01 cm$^{-3}$, to filter out the contamination of the cloud droplets, and drizzle water splashing.

The sub-cloud aerosols and CCN are selected within the below cloud base part of the coupled layer, which is described in the last section, in order to better assess the aerosol-cloud interactions. The above-cloud aerosols and CCN are selected between the cloud top and 200 m above. Note that the selection criteria of 200 m above the cloud top would inevitably induce uncertainty in the cloud top ACI assessment, depending on the vertical trend of the individual aerosol profile. Over the Southeast Atlantic, Gupta et al. (2021) conducted an analysis focusing particularly on the differing impacts when biomass burning aerosols are in contact with marine stratocumulus cloud tops, using 100 m above as the demarcation, versus when they are separated by various distances, and found that significant differences were observed in cloud microphysics, owing to different droplet evaporation and nucleation, compared to profiles in which aerosols and cloud layer are separated. That result is in agreement with the modeling sensitivity study over the Eastern North Atlantic by Wang et al. (2020), who found that aerosol plumes can exert impacts on the cloud-top microphysics only when they are in close contact with the cloud layer. During much of the ACE-ENA campaign, nearly constant (and sometimes decreasing) vertical atmospheric profiles of aerosol concentration were observed within a few hundred meters above the cloud top. Aerosol intrusions due to long-range transport, particularly during the summer season, were observed to induce an elevated aerosol layer in higher altitudes that was not in contact with the cloud layer. The frequent new particle formation events during SOCRATES significantly alter the free-

troposphere Aitken mode aerosol budget, but the aerosols would need to further subside to impact the cloud (McCoy et al., 2021; Zhang et al., 2023). Note that from previous studies on ACE-ENA and SOCRATES, the aerosol vertical profiles within ~200 m above the cloud layers are typically found to have less variation (Wang et al., 2020; Wang et al., 2021; McCoy et al., 2021; Zhang et al., 2023), hence representing the aerosol layers in contact with the cloud. Hence, the 200 m criterion used in this study provides a sufficient sample size population for statistical analysis.

## 3. Aerosol, cloud, and drizzle properties of selected cases

### 3.1 Aerosols and CCN in above- and sub-cloud regimes

The probability density functions (PDFs) of aerosols, CCN, and cloud microphysical properties from selected cases during the ACE-ENA and SOCRATES field campaigns are presented in Figure 1. Notably, the $N_a$, $N_{Acc}$ and $N_{CCN0.35\%}$ values from the SOCRATES are the highest among the three IOPs, followed by the ACE-ENA Sum and ACE-ENA Win as illustrated in both above-cloud (Figs. 1a-1c) and sub-cloud regimes (Figs. 1d-1f). Such variations can be linked to the different aerosol sources in the ACE-ENA and SOCRATES regions, especially during the summer and winter seasons over the Azores.

In the SOCRATES region, according to the previous studies involving back-trajectory analyses, dominant air masses within the MBL primarily originate from the south or from the west, skirting the Antarctic coast (Zhang et al., 2023), while the air masses above the MBL follow a similar transport pathway, they can also originate from the tip of southern Africa and be transported southeast along the warm conveyor belt (McCoy et al., 2021). Above-cloud aerosol $N_{CCN0.35\%}$ values analyzed during SOCRATES (674.6 $cm^{-3}$) are primarily constituted by the Aitken mode aerosols because the mean $N_{Acc}$ is only 62.5 $cm^{-3}$. Previously, McCoy et al. (2021) reported average $N_{CCN0.35\%}$ values of 680.69 $cm^{-3}$, 546.28 $cm^{-3}$ and 465.05 $cm^{-3}$ for mid-troposphere, above and below cloud for the multiple SOCRATES cases, respectively. For individual cases, the above cloud aerosols vary from a couple hundred to over a

thousand particles per cubic centimeter (McCoy et al., 2021; Zhang et al., 2023). These aerosols are predominantly produced from the oxidation of biogenic gases, notably dimethyl sulfide (DMS) emitted by marine biological productivity (Sanchez et al., 2018; McCoy et al., 2020). The rising air currents in MBL transport these particles into the free troposphere with dominant aerosol population over the SO (McCoy et al., 2021; Sanchez et al., 2021). Hence, it reinforces the notion that the SO represents a pre-industrial marine environment where the influence of anthropogenic and biomass-burning aerosols is mostly negligible (McCoy et al., 2020, 2021).

Conversely, the ENA region experiences aerosols of varied origins, spanning maritime air masses to those heavily influenced by continental emissions from North America or Northern Europe, especially during the summer season (Logan et al., 2014; Wang et al., 2020). The summer air mass back-trajectories within the MBL strongly feature recirculating flow around the Azores high. During the wintertime, however, the air masses predominantly originate in the free troposphere, are transported above the MBL, and are then further entrained to the MBL by large-scale subsidence, indicating less influence from continental pollution (Y. Wang et al., 2021b). During the ACE-ENA Sum, the MBL is enriched by sulfate and carbonaceous particles (Y. Wang et al., 2021b; Zawadowicz et al., 2021). This enhancement is attributed both to local generation from DMS and to the long-range transport from the continental air masses, resulting in the mean $N_a$ of 312.6 cm$^{-3}$ and 301.5 cm$^{-3}$ for above- and sub-cloud regimes, respectively. The ACE-ENA Win exhibits the lowest aerosol and CCN concentrations, predominantly sourced from local maritime influences, and coupled with reduced continental air mass intrusions (Zheng et al., 2018; Y. Wang et al., 2021b).

Figure 1a and 1d reveal that there are more above-cloud $N_a$ during the three IOPs than sub-cloud values, especially during the SOCRATES. The higher above-cloud $N_a$ values from the three IOPs are primarily contributed by Aitken mode aerosols because their corresponding $N_{Acc}$ values are much lower (Figs. 1a & 1b). It is interesting to note that the above-cloud $N_{CCN0.35\%}$ values exceed the $N_{Acc}$ for all three IOPs (Figs. 1b & 1c), implying that a significant fraction of Aitken mode aerosols can be activated

to become CCN, corroborating findings from earlier studies (McCoy et al., 2021; Zheng et al., 2021). For the sub-cloud regime, the $N_a$ values for SOCRATES and ACE-ENA Win are ~70-80% of their corresponding above-cloud values, and the $N_a$ during ACE-ENA Sum is almost identical to its above-cloud value. Notice that the sub-cloud $N_{Acc}$ values from the three IOPs are more than double the above-cloud $N_{Acc}$ values, and most of the sub-cloud accumulation mode aerosol can be activated to become CCN at SS of 0.35%. It is interesting to note that the higher $N_{CCN0.35\%}$ at the sub-cloud layer during SOCRATES may partially be a result of aerosols being positively impacted by cloud dynamic processes (Figs. 1e & 1f), which is suggested by previous studies (McCoy et al., 2021; Zhang et al., 2023) and will be further discussed in the following paragraphs.

To further investigate the above- and sub-cloud aerosol properties from the three IOPs, the aerosol droplet size distributions are analyzed in Figure 2. It is evident that SOCRATES aerosols have the highest concentrations of Aitken mode particles ($D_p = 0.06 – 0.1$ µm, given that the $<$ 0.06 µm is not available from UHSAS) for the above- and sub-cloud regimes. McCoy et al. (2021) and Zheng et al. (2021) identified analogous origins and formations of the above-cloud Aitken mode aerosols over the SO and ENA regions and concluded that these aerosols primarily originate from the nucleation of photo-oxidation products of DMS, notably $H_2SO_4$ and MSA, in the free troposphere. The differential concentrations can be ascribed to the fact that sea-surface DMS concentrations in the SO are generally higher than those in the ENA region (Aumont et al., 2002; Zhang et al., 2023). Moreover, DMS emissions in the ENA during the summer season surpass those during winter (Zawadowicz et al., 2021). For the accumulation mode aerosols (0.1 – 1 µm), the $N_{Acc}$ values for both above- and sub-cloud regimes during SOCRATES decrease monotonically with particle size. The results in Figure 2 further support the finding that Aitken mode aerosols are dominant over the SO. The $N_{Acc}$values during ACE-ENA show slight uplifts for the small accumulation mode aerosols (< 0.3 µm), particularly during the summer, reflecting the signal of potential long-range transport of fine-mode aerosols (Wang et al., 2020; Y. Wang et al.,

2021b). Consequently, such comparison reinforces the notion that the SO represents a largely pre-industrial marine environment, wherein the influence of anthropogenic and biomass-burning aerosols is minimal (McCoy et al., 2020, 2021; Zhang et al., 2023).

When contrasting the aerosol size distributions in the sub-cloud regime (Fig. 2b) with those in the above-cloud regime, the influence of cloud processing on aerosols is discernibly non-trivial, particularly under the cloud-topped MBL conditions examined in this study. The free tropospheric aerosols can be entrained and contribute to the population of Aitken mode aerosols within the MBL, and the sub-cloud aerosols can also be subject to the influence of new particle formation in the upper MBL, though arguably less effective than those within the free troposphere (Zheng et al., 2021). Additionally, in-cloud Brownian capture can lead to a substantial reduction in Aitken mode aerosols (Hudson et al., 2015; Wyant et al., 2022), providing the rationale for the observed decrease in Aitken mode aerosols from above- to the sub-cloud regime, especially for particles smaller than 0.07 µm. In addition, cloud chemical processing, such as the aqueous-phase condensation of sulfuric gas onto the aerosol cores inside the cloud droplets, is particularly pronounced during the transitioning of Aitken mode aerosols to accumulation mode aerosols (Hudson et al., 2015; Zhang et al., 2023).

The larger Aitken mode aerosols (> 0.07 µm) in the above- and sub-cloud regimes can effectively grow to accumulation mode aerosols through coagulation and water vapor diffusional growth (Covert et al., 1996), contributing to the elevated accumulation mode aerosol distribution and increased $N_{Acc}$ in the sub-cloud regime. These processes are evident by the decrease of critical supersaturations from above-cloud (between 0.35% - 0.4%) to sub-cloud (between 0.3% - 0.35%) during SOCRATES (Fig. S3) because the aerosol droplet sizes are enlarged and more readily become CCN. Furthermore, the collision-coalescence combines mixtures of large and small cloud droplets, and results in the sub-cloud aerosol residuals shifting towards the larger size upon the drizzle droplet evaporation below the cloud. This partially elucidates the observed increase in the tail-end of the accumulation mode aerosol distribution for all three IOPs. The elevation in sub-cloud coarse mode aerosols observed for both ACE-ENA IOPs

(as seen in Fig. 2) can be attributed to the evaporation of collision-coalescence-enlarged drizzle droplets and the intrusion of sea spray aerosols (e.g., sea salt), as illustrated and analyzed based on a summertime case study that exhibits the signal of cloud-processing aerosols (Zheng et al., 2022b), and the long-term aerosol physicochemical properties over the ARM-ENA ground-based observatory (Zheng et al., 2018) particularly during the winter season where the production of sea spray aerosol is prevalent.

### 3.2 Distribution of bulk cloud microphysical properties

The PDFs of MBL cloud microphysical properties ($N_c$, $r_c$, $LWC_c$) derived from aircraft in-situ measurements from the three IOPs are shown in Figures 1g-1i. The mean microphysical properties for the individual cloud profiles are listed in Table S2. The SOCRATES has the highest sub-cloud aerosols and CCN, and subsequently feature a larger number of smaller cloud droplets, given the highest $N_c$ (148.3 cm$^{-3}$) and smallest $r_c$ (8 μm) among the three IOPs. These results have further confirmed and reassured our understanding of the aerosol first indirect effect: a larger population of aerosols induce a higher number concentration of small cloud droplets under constrained liquid water content conditions, and thus the MBL clouds reflect more incoming solar radiation (Twomey, 1977). The ACE-ENA Win clouds feature the fewest $N_c$ (70.6 cm$^{-3}$) and largest $r_c$ (9.8 μm), while the $N_c$ and $r_c$ (89.4 cm$^{-3}$ and 9 μm) during ACE-ENA Sum fall between the SOCRATES and ACE-ENA Win values. Considering the aerosol competing effect against the available water vapor, the relatively abundant aerosols in SOCRATES might account for the observed narrower $r_c$ distribution, which peaks between 6 – 10 μm. SOCRATES has a lower cloud-layer water vapor mixing ratio (figure not shown) compared to ACE-ENA because the SO region has been observed to contain less precipitable water vapor than the ENA region due to the colder sea surface temperatures (Marcovecchio et al., 2023). Therefore, the aerosol and cloud properties in Figure 1 promise further examination of different cloud microphysical responses to aerosols via the ACI process. Note that the $N_{CCN0.35\%}$ are lower than $N_c$ values during the ACE-ENA

Win, which is also confirmed in previous studies (J. Wang et al., 2022; Wang et al., 2023). This interesting phenomenon can potentially be attributed to a combination of factors, including lower MBL aerosol sources, stronger in-cloud coalescence-scavenging depletion of sub-cloud aerosols, and the aircraft snapshots capturing the equilibrium states of aerosols and cloud due to enhanced aerosol activations induced by stronger updrafts during the ACE-ENA Win (J. Wang et al., 2022). This thereby compels further investigation into the potential impacts of precipitation on the MBL CCN budget which is further discussed in Section 4.

### 3.3 Vertical distributions of cloud and drizzle microphysics

The vertical distributions of the cloud and drizzle microphysical properties within the cloud layer from the three IOPs are shown in Figure 3. To ensure the representativeness of the vertical profiles, all the in-cloud samples are vertically smoothed using a triangular moving average method, and are inverse-distance weighted in every 50 m moving altitude windows. Furthermore, the altitude is then normalized by $z_i = \frac{Z - Z_{base}}{Z_{top} - Z_{base}}$, where $z_i = 0$ denotes cloud base and $z_i = 1$ denotes cloud top. Consistent with previous discussions on the bulk microphysics distribution, the mean $N_c$ values from SOCRATES are consistently higher than ACE-ENA Sum, and ACE-ENA Win for the entire cloud layer, with a slight increase ranging from the cloud base to the upper-middle part ($z_i \approx 0.85$) and then decreasing toward the cloud top (Fig. 3a). All $r_c$ values from the three IOPs show a near-linear increase from cloud base to top, with the smallest values observed during SOCRATES and the largest values observed during ACE-ENA Win (Fig. 3b).

The warmer and drier air near the cloud top entrains into the cloud layer and further mixes downward, often resulting in the evaporation of small cloud droplets and the shrinking of droplet sizes, which oppose condensational growth (Desai et al., 2021). Decreases in both $N_c$ and $LWC_c$, and the reduced growth of $r_c$ near the cloud top ($z_i > 0.85$) support signals of cloud-top entrainment mixing

during all three IOPs. It is interesting to note that the $r_c$ values from SOCRATES increase monotonically from cloud base to top, while the $r_c$ values from both ACE-ENA Sum, and ACE-ENA Win increase until $z_i \approx 0.8$ and then remain nearly constant, although all of their $N_c$ values (at $z_i \approx 0.8$) decrease towards the cloud top. When dry air entrainment occurs at the cloud top, some of the upper-level smaller cloud droplets will evaporate, which leads to decreases in $N_c$ (Fig. 3a). As a result, the nearly constant $r_c$ values (at $z_i > 0.8$) might represent the equilibrium balance between two competing processes: cloud droplet condensational and collision-coalescence growths and the entrainment mixing evaporation effects.

Carrying the distinct discrepancies in the mean values for all layers, the $N_c$ and $r_c$ from ACE-ENA Sum, and ACE-ENA Win clouds experienced similar vertical evolutions as SOCRATES. The increases of $r_c$ ($\Delta r_c$) from cloud base to cloud top are 4.03 μm, 4.78 μm and 5.85 μm, with percentage increases of 66%, 68% and 79%, for SOCRATES, ACE-ENA Sum, and ACE-ENA Win, respectively. Even though, theoretically, the condensational growth effect would be more pronounced on smaller cloud droplets due to their smaller surface area (Wallace and Hobbs, 2006), SOCRATES exhibits the thickest mean cloud thickness but experienced the least $r_c$ increase among the three IOPs. This suggests that high aerosol loading limits the overall growth of the cloud droplet size distribution (DSD) in SOCRATES clouds, while the ACE-ENA Win clouds show the strongest $r_c$ increase, in contrast. This comparison indicates different cloud microphysical responses to aerosol perturbations in the three IOPs, which will be further discussed in Section 4.1. The $LWC_c$ values from the three IOPs are comparable to each other. The vertical distributions of MBL cloud microphysical properties examined in this study are in good agreement with the previous studies conducted on these two field campaigns (Wu et al., 2020a; Y. Wang et al., 2021a; J. Wang et al., 2021; Wang et al., 2023). Cloud adiabaticity is a key parameter as it provides insight into the degree of mixing and microphysical processes occurring within clouds. The sub-adiabatic conditions indicate that the $LWC_c$ is less than what would be expected in an adiabatic scenario, often due to processes such as in-cloud collision-coalescence and entrainment mixing (Hill et al., 2009; Braun et

al., 2018; Gao et al., 2020; Wu et al., 2020b). In addition, the cloud adiabaticity is defined as $f_{ad} = LWC_c/LWC_{ad}$, where the $LWC_{ad}$ denotes adiabatic LWC (Wu et al., 2020b). As shown in Figure S4, the clouds from all three IOPs feature certain levels of sub-adiabaticity above the cloud base. Considering the inter-cloud layer-mean $f_{ad}$, the campaign-mean $f_{ad}$ values are 0.689±0.229, 0.542±0.143, and 0.490±0.207 for SOCRATES, ACE-ENA Sum, and ACE-ENA Win, respectively.

To quantitatively evaluate the impact of cloud-top entrainment mixing rate on cloud droplets, we adapt the method of Albrecht et al. (2016), where the cloud-top entrainment rate ($w_e$) can be expressed as

$$w_e = A_\sigma * \sigma_w / R_{i\sigma} \ , \tag{6}$$

where the turbulence kinetic energy (TKE) dissipation coefficient $A_\sigma$ is empirically taken as 26 as in Albrecht et al. (2016), and the $R_{i\sigma}$ is the buoyancy Richardson number calculated by $(g/\theta_0) * (\Delta\theta_v h/\sigma_w^2)$. $\sigma_w$ denotes the standard deviation of vertical velocities taken near the cloud top ($z_i$ > 0.9), and $h$ is the MBL height. $\theta_0$ is the reference potential temperature and $\Delta\theta_v$ is the virtual potential temperature difference across the temperature inversion layer above the cloud. Given the valid cloud-top virtual potential temperature and vertical velocity measurements for the selected cloud cases, the averaged $w_e$ values are 0.570±0.834 cm $s^{-1}$, 0.581±0.560 cm $s^{-1}$, and 0.960±1.127 cm $s^{-1}$ for SOCRATES, ACE-ENA Sum, and ACE-ENA Win, respectively. The stronger $w_e$ during ACE-ENA Win might be induced by the generally weaker cloud-top inversions and stronger near-cloud top turbulence (Fig. 5a), compared to the summertime when the ENA is dominated by the large-scale high-pressure system (Ghate et al., 2021). Within the above-cloud inversion layer, the temperature (water vapor mixing ratio) differences ΔT (Δq) are 1.76 K (-1.75 g $kg^{-1}$), 1.54 K (-1.66 g $kg^{-1}$) and 1.48 K (-1.09 g $kg^{-1}$) for SOCRATES, ACE-ENA Sum, and ACE-ENA Win, respectively. The virtual potential temperature differences $\Delta\theta_v$ are 4.90 K, 5.16 K, and 3.82 K, for SOCRATES, ACE-ENA Sum, and ACE-ENA Win, respectively, indicating relatively dryer entrained airmasses during SOCRATES and ACE-ENA Sum.

Considering the near cloud-top proportion of cloud where the $LWC_c$ experienced decrease, the difference in $LWC_c$ (between the cloud-top value and the upper-middle cloud maximum for the mean profiles) for the ACE-ENA Sum (-0.032 g m$^{-3}$) is higher than the reductions in winter (-0.018 g m$^{-3}$) and SOCRATES (-0.009 g m$^{-3}$), albeit that the $w_e$ for ACE-ENA Sum is comparable to SOCRATES, and much lower than ACE-ENA Win values. The warmer and dryer entrained air can partially contribute to the greater $LWC_c$ reduction and the lower $f_{ad}$ (0.39) during the ACE-ENA Sum than those during the ACE-ENA Win ($f_{ad}$ = 0.45) and SOCRATES ($f_{ad}$ = 0.66) near the cloud top (Fig. S4). For the three IOPs, the $N_c$ and $LWC_c$ exhibit stable trends from the cloud base, followed by noticeable decreases near the cloud top mixing zone, while the changes in $r_c$ trends near the cloud top were not as dramatic as the others. Such characteristics of the cloud microphysics vertical profiles indicate the signal of inhomogeneous mixing, which occurs when dry and warm air mixes unevenly and slowly with the cloud air, hence partially evaporating the cloud droplets (Lehmann et al., 2009; Lu et al., 2011). The results are consistent with previous research results regarding stratocumulus clouds over multiple field campaigns (Brenguier et al., 2011; Jia et al., 2019) and with the findings for selected cases during ACE-ENA (Yeom et al., 2021) and SOCRATES (Sanchez et al., 2020). The near-cloud top $r_c$ profiles ($z_i$ > 0.8) for the ACE-ENA cases exhibit fewer increases compared to SOCRATES, which could be possibly attributed to more effective mixing due to the stronger entrainment rate, particularly during ACE-ENA Win, eventually reaching a smaller equilibrium in terms of mean sizes.

Figures 3d-3f illustrate the normalized profiles of MBL drizzle microphysical properties. The $N_d$ values from the three IOPs mimic each other, which all maximize at the cloud top and then monotonically decrease toward the cloud base (Fig. 3d), while their $LWC_d$ values follow a similar trend, albeit with relatively large differences (Fig. 3f). In contrast to the $N_d$ and $LWC_d$ trends, the $D_{mmd}$ gradually increase from cloud top to cloud base (Fig. 3e), making physical sense since the drizzle droplets are typically formed near the cloud top and continuously grow via collision-coalescence process while falling. The

ACE-ENA Win drizzle $D_{mmd}$ and $LWC_d$ values are distinctively larger than those in ACE-ENA Sum and SOCRATES. It is interesting to note that near the cloud top ($z_i > 0.9$), ACE-ENA Win has comparable $N_d$ but much larger $D_{mmd}$ than the other two IOPs, suggesting that there were more large drizzle embryos formed from large cloud droplets (Fig. 3b) during ACE-ENA Win. It is noteworthy that the $D_{mmd}$ in the lower-half region of the ACE-ENA Win clouds experienced rapid growth from ~80 μm to ~105 μm (Fig. 3e), and this increment of ~25 μm contributed to most of the $D_{mmd}$ growth from cloud top to cloud base (33.5 μm), indicating a stronger warm-rain process during the winter.

In order to further analyze the cloud-to-drizzle conversion processes, the cloud and drizzle DSDs are categorized into four segments based on their relative position within the cloud layer (Fig. 4): upper cloud ($z_i > 0.8$, Fig. 4a), upper-middle cloud ($0.5 \leq z_i < 0.8$, Fig. 4b), lower-middle cloud ($0.2 \leq z_i < 0.5$, Fig. 4c) and lower cloud ($z_i < 0.2$, Fig. 4d). The cloud DSDs ($D_p < 40$ μm) from the three IOPs gradually shift towards larger sizes, moving from the lower to the upper cloud regions. This is accompanied by the narrowing of the cloud DSD ranges, as evidenced by the decline in the relative dispersion of cloud droplets (ε), which is defined as the ratio between the standard deviation and the mean radius of the distribution. At the lower portion of the cloud (Fig. 4d), the relatively greater value of ε represents the co-existence of the newly formed small cloud droplets from recently activated CCN and the sedimentation of larger droplets from the upper sections of the cloud. In addition, the discrepancies in ε between the three IOPs may be attributed to the sub-cloud aerosol differences, which essentially resided in different microphysical regimes. Y. Wang et al. (2021a) stated that higher aerosol loading would lead to increased ε due to the water vapor competition effect, supporting the discrepancy between SOCRATES and ACE-ENA Sum, which can be categorized as a water vapor-limited regime. Meanwhile, the ACE-ENA Win exhibits characteristics of an aerosol-limited regime, in which the cloud DSDs tend to be narrower than in the water-limited regime, due to enhanced droplet growth, and the ε values further decrease with height via the condensational narrowing effect (J. Chen et al., 2018).

Notably, for the four cloud portions from cloud base to cloud top, the skewness of summer (winter) cloud DSDs are 0.627 (0.271), 0.358 (0.175), 0.098 (-0.063), and -0.362 (-0.554), respectively. The cloud DSDs during ACE-ENA Win exhibit a more pronounced negative skew (to the left) than those during ACE-ENA Sum, which can be partially attributed to the activation of more sub-cloud coarse mode aerosols becoming larger cloud embryos, as demonstrated in Fig. 2. These coarse mode aerosols, whether from primary production of sea spray or the residuals of evaporated drizzle drops, are more easily activated (or re-activated) into larger cloud droplets when they intrude (or recirculate) into the cloud layer (Hudson and Noble, 2020; Hoffmann and Feingold, 2023). Nevertheless, it is challenging to pinpoint the actual origins of coarse mode aerosols from the perspective of aircraft observational snapshots, thus requiring further numerical modeling work. Ascending within the cloud, the process of water vapor condensation perpetually pushes the DSD towards larger sizes, culminating in a more negatively skewed DSD. Concurrently, the cloud-top entrainment mixing plays a pivotal role in minimizing ε in the upper cloud region, as elaborated by Lu et al. (2023).

In the upper region of the cloud (Fig. 4a), the ACE-ENA Win clouds contain more cloud droplets close to 40 μm, albeit the mean $N_c$ is lower. This scenario is conducive to the formation of larger drizzle embryos compared to summertime clouds, as depicted in Fig. 3e. In comparison, the SOCRATES clouds feature a pronounced log-normal DSD than during ACE-ENA, as the DSDs peak at $D_p$~15 μm throughout the cloud, and subsequently, the lack of larger cloud droplets resulted in the smaller drizzle embryos near the cloud top. As the newly formed drizzle drops descend and continuously grow through the collision-coalescence process, the drizzle DSDs ($D_p > 40$ μm) are noticeably broadened. From upper to lower cloud regions, the longer tails of the drizzle DSDs expand at the cost of smaller drizzle drops and cloud droplets via the collision-coalescence process. The clouds observed during ACE-ENA, especially in wintertime, contain more large drizzle drops ($D_p > 200$ μm) than SOCRATES, which is reflected in the distinct differences in the vertical $D_{mmd}$ as shown in Fig. 3e.

It has been intensively studied that in-cloud turbulence can stimulate collision-coalescence and consequently enhance the drizzle evolution processes (Pinsky et al., 2007; Grabowski and Wang, 2013; Wu et al., 2017; S. Chen et al., 2018). The turbulence strength is characterized by the turbulence kinetic energy (TKE), which is calculated as:

$$TKE = \frac{1}{2}(\overline{u'^2} + \overline{v'^2} + \overline{w'^2}), \quad (7)$$

where the turbulent perturbations of vertical ($\overline{w'^2}$) and horizontal ($\overline{u'^2}$ and $\overline{v'^2}$) components are calculated as the simple moving variance in a 10 s window centered at the measurement time, without window weighting function, using 1 Hz data for all three IOPs. The $w$ data is confined to an absolute aircraft roll angle of less than 5° (Cooper et al., 2016). Given the average aircraft ground speed of ~140 m $s^{-1}$ and vertical speed of ~5 m $s^{-1}$ (Atlas et al., 2020), the smallest resolved wavelength is 140 m. Hence, within the 10s moving window, the ~50 m in the integral vertical range is able to resolve the eddies up to ~1400 m in size, and preserve the potential of capturing the inertial subrange.

As shown in Figure 5, the vertical wind variances (Fig. 5b) in ACE-ENA Win (layer-mean of 0.244 $m^2$ $s^{-2}$) are generally higher than those in ACE-ENA Sum (0.153 $m^2$ $s^{-2}$) and SOCRATES (0.147 $m^2$ $s^{-2}$), while the horizontal wind variances (Figs. 5c & 5d) are comparable between ACE-ENA Sum and ACE-ENA Win but much higher than the SOCRATES, resulting in higher TKE during ACE-ENA. Note that the higher $w'^2$ near cloud top corresponds to the stronger entrainment rate in ACE-ENA Win. Near the cloud top, turbulence effectively enhances coalescence between the larger cloud droplets, primarily by increasing the relative velocities between droplets (Magaritz-Ronen et al., 2016; Ghate and Cadeddu, 2019), and this is especially true for the vertical component $w'^2$ of TKE. The horizontal turbulence components, the $u'^2$ and $v'^2$ can also play a role in mixing the ambient air masses and contribute to the broadening of DSD (Wu et al., 2017). The use of TKE provides an illustration that in-cloud turbulence during ACE-ENA is stronger than that observed during SOCRATES. That being said, the quantitative evaluation of the turbulent enhancement of collision-coalescence requires access to the

eddy dissipation rate, as typically used in model parameterizations (Grabowski and Wang, 2013; Witte et al., 2019). The smallest scales resolvable with the 1 Hz measurement used in this study are on the order of 140 meters, thus capturing only the larger-scale end of the inertial subrange and larger turbulent motions. Consequently, the ability to resolve smaller eddies and turbulent structures, crucial for understanding the energy cascade within the inertial subrange, is limited by coarse spatial and temporal resolutions and aliasing issues (Siebert et al., 2010; Muñoz-Esparza et al., 2018; Kim et al., 2022). Therefore, to fully resolve the spectrum of turbulence and quantitatively examine energy dissipation and mixing processes, access to higher-frequency measurements is required to capture smaller eddies within the inertial subrange (Siebert et al., 2010; Lu et al., 2011; Waclawczyk et al., 2017). Additionally, further quantifying the entrainment-mixing mechanisms also requires high-frequency eddy dissipation and accurate examination of the mixing time scale (Lehmann et al., 2009; Lu et al., 2011) for individual profiles. Though currently beyond the scope of this study, utilizing the high-rate measurements of velocities available from SOCRATES (at 25 Hz) and ACE-ENA (at 20 Hz) to explore those mechanisms further will be of interest to future investigations.

Drizzle formation and evolution in the ACE-ENA Win clouds are noticeably stronger than in the other two IOPs, which could be attributed to multiple factors. First, the ambient aerosols and CCN during winter are substantially fewer, featuring clean environments that promote the formation of generally larger cloud droplets due to the availability of more water content per droplet. Larger cloud droplets are more likely to collide and coalesce into drizzle drops, leading to relatively heavier precipitation (Chen et al., 2011; Duong et al., 2011; Mann et al., 2014). Furthermore, deeper cloud layers with mean thickness of (392.4 m) during ACE-ENA Win were observed when compared to the ACE-ENA Sum clouds (336.3 m). In a thicker cloud layer with sufficient turbulence, the residence times of large cloud droplets and drizzle drops would become longer, and the chance of collision-coalescence growth could be effectively increased by recirculating the drizzle drops (Brost et al., 1982; Feingold et al., 1996; Magaritz et al., 2009; Ghate et al., 2021). Additionally, the prevalence of winter season precipitation-evaporation-

induced MBL cold pools disturbs the MBL thermodynamics and contribute to turbulent mixing (Zuidema et al., 2017) can provide a strong dynamical forcing perturbation to the warm-rain process (Jenson et al., 2021; J. Wang et al., 2022; Smalley et al., 2024). The physical hypotheses from previous studies could potentially serve as the explanation for the phenomena that the ACE-ENA Win drizzle DSD is sufficiently broadened, and the $D_{mmd}$ is enlarged toward the cloud base. In comparison, although the SOCRATES exhibits even thicker clouds (487.4 m), the drizzle processes are seemingly suppressed by the much higher ambient aerosol and CCN concentrations.

## 4 Aerosol-cloud-precipitation interactions (ACPIs)

### 4.1 Cloud microphysical responses on aerosols

The impacts of aerosol loading on cloud microphysical properties can be assessed by the aerosol-cloud interaction (ACI) index, which can be quantified as both:

$$ACI_N = \frac{\partial ln\,(N_c)}{\partial ln\,(N_{CCN,0.35\%})}, \quad (8)$$

and

$$ACI_r = -\frac{\partial ln\,(r_c)}{\partial ln\,(N_{CCN,0.35\%})}, \quad (9)$$

which emphasizes the cloud microphysical responses to CCN via the relative logarithmic change of $N_c$ and $r_c$ to the change in $N_{CCN,0.35\%}$ (Feingold et al., 2003; McComiskey et al., 2009). Physically, the ACI process involves aerosols intruding into the cloud layer, activating as cloud droplets, and subsequently altering cloud DSD and dispersion (Zheng et al., 2022a&b) under varying water vapor conditions. Therefore, the cloud microphysical responses within the lower region of the cloud are assessed, which is the first stage in which the sub-cloud CCN can directly interact with the cloud droplets. Furthermore, the similarity in the vertical integral of $LWC_c$ (as shown in Fig. 3c) provides comparable liquid water between three IOPs for the assessment of newly generated cloud embryos from activated CCN because the $ACI_r$ is normally assessed under a fixed liquid water (Zheng et al., 2020).

Considering all the cases from three IOPs with available CCN measurements (some cases without CCN measurements during SOCRATES), the $N_c$ and $r_c$ at the lower cloud ($z_i < 0.2$) are plotted against the sub-cloud $N_{CCN,0.35\%}$ in Figures 6a and 6b, and the ACI indices are calculated as $ACI_{N,CB}$ and $ACI_{r,CB}$ (CB denoting the assessment near the cloud base). The ACI indices from three IOPs are in the ACI range of the previous studies in MBL clouds (Twohy et al., 2005; Lu et al., 2009; Diamond et al., 2018) using aircraft in-situ measurements. Note that the availability of valid sub-cloud measurements inevitably limits the sample size, especially for SOCRATES, as shown in Table S2. As shown in Figure 6a, the $ACI_{N,CB}$ for ACE-ENA Win (0.748) is higher than ACE-ENA Sum (0.617), indicating that $N_c$ is more sensitive to the sub-cloud $N_{CCN,0.35\%}$ during the winter. In other words, aerosols intruding into the cloud layer are easily activated to become cloud droplets. The $N_c$ sensitivity for the SOCRATES cloud (0.692) lies between the two ACE-ENA IOPs. The $ACI_{N,CB}$ values from three IOPs are generally higher than the $ACI_N$ values from the layer-mean $N_c$ against the sub-cloud $N_{CCN0.35\%}$ (not shown). Previous studies have shown that the enhanced vertical turbulence (updraft velocity) can effectively facilitate CCN replenishment into the cloud layer (Hu et al., 2021; Zheng et al., 2022a&b) and increase the actual in-cloud supersaturation (Brunke et al., 2022), thus leading to a more efficient cloud droplet formation, enhancing the $ACI_{N,CB}$ . By correlating the mean TKE values with the CCN activation ratio ($N_c/N_{CCN,0.35\%}$) for all individual cloud cases, the three IOPs show moderate but statistically significant correlation coefficients of 0.36, 0.55, and 0.51 for ACE-ENA Sum, ACE-ENA Win, and SOCRATES, respectively. This result reinforces the notion that the CCN activation fractions, particularly during the ACE-ENA Win, are significantly correlated with in-cloud turbulence intensities. Furthermore, more coarse mode aerosols during ACE-ENA Win are also favorable to the activation efficiency (Dusek et al., 2006).

As for the $r_c$ responses to CCN (Fig. 6b), the typical Twomey effect, where more CCN compete against available water vapor and result in smaller cloud droplets, is evident by different cloud

susceptibility between the three IOPs. SOCRATES features a higher $ACI_{r,CB}$ (0.311), suggesting that an increase in $N_{CCN,0.35\%}$ can result in a significant decrease in $r_c$, compared to ACE-ENA Sum (0.206) and ACE-ENA Win (0.263). Although the absolute range of variation for $r_c$ during SOCRATES is smaller, the slope is much deeper (Fig. 6b). Recall that the sub-cloud $N_{CCN,0.35\%}$ during SOCRATES is generally higher than ACE-ENA and contains more small-sized aerosols (as indicated in Fig. 2b). Consequently, after activation, the lower part of the cloud exhibits a higher number of smaller cloud droplets for SOCRATES, as shown in Fig. 4d. Therefore, as more CCN intrudes into the cloud, the competition for water vapor among newly-activated cloud droplets becomes more pronounced, given similar water availability. In contrast, the presence of larger cloud droplets near the cloud base, whether activated from coarse-mode aerosols or remaining as residuals from collision-coalescence, would elevate the $r_c$ especially under the relatively less CCN condition, hence inevitably dampening the $ACI_{r,CB}$ during ACE-ENA. However, a more comprehensive investigation into the cloud microphysical responses to CCN intrusions under a larger range of various water supply conditions, and further untangling the ACI from the meteorological influences, will require additional aircraft cases from more field campaigns. Examples include, the VAMOS Ocean-Cloud-Atmosphere-Land Study (VOCALS; Wood et al., 2011), the Cloud System Evolution over the Trades (CSET; Albrecht et al., 2019), the ObseRvations of CLouds above Aerosols and their intEractionS (ORACLES; Redemann et al., 2021), and the Aerosol Cloud meTeorology Interactions oVer the western ATlantic Experiment (ACTIVATE; Sorooshian et al., 2019). Note that the $ACI_{r,CB}$ values in Figure 6b are also larger than the results from the layer-mean $r_c$ against sub-cloud $N_{CCN,0.35\%}$, since the layer-mean microphysics is more subject to the cloud droplet evolution processes such as condensational growth and collision-coalescence.

To investigate the ACI indices at the upper level of the cloud, the $N_c$ and $r_c$ at the upper cloud ($z_i$ > 0.8) are plotted against the above-cloud $N_{CCN,0.35\%}$ in Figures 6c and 6d, and the ACI indices are calculated as $ACI_{N,CT}$ and $ACI_{r,CT}$ (denoting the assessments near the cloud top). Compared to the

$ACI_{N,CB}$ and $ACI_{r,CB}$, the $ACI_{N,CT}$ and $ACI_{r,CT}$ are much weaker, especially for $ACI_{r,CT}$, as the near cloud top droplets are too large for above-cloud aerosols to exert a significant influence on $r_c$ (Diamond et al., 2018; Gupta et al., 2022). The weaker cloud top $N_c$ dependence on the $N_{CCN,0.35\%}$ could be due to the legacy of the sub-cloud CCN impacts on $N_c$ being conveyed to the cloud top. This occurs because free tropospheric aerosols and CCN can be entrained into the MBL before and during the cloud process, as observed in the assessment of inter-cloud cases. Note that the $LWC_c$ near the cloud top for the three IOPs are not comparable to each other, which might also induce uncertainty in the near-cloud-top ACI assessment. These weaker relationships support the notion that although the aerosols entrained into the upper-cloud region can affect the cloud microphysics to a certain degree, the effects are less pronounced than those from the sub-cloud aerosols (Diamond et al., 2018, Wang et al., 2020) because the MBL cloud $N_c$ and $r_c$ variations are dominated by the condensational growth, collision-coalescence, and entrainment mixing processes near the cloud top.

### 4.2 Precipitation susceptibility

The precipitation susceptibility relies on the assessment of relative responses in the precipitation rate to the change in $N_c$ (Feingold and Seibert, 2009; Sorooshian et al., 2009), which is defined as:

$$S_o = -\frac{\partial ln\,(R_{CB})}{\partial ln\,(N_c)}, \quad (10)$$

where the $R_{CB}$ is the cloud base precipitation rate calculated in section 2 (equation 5). By incorporating all the cloud cases, including both precipitating and non-precipitating clouds (the $R_{CB}$ can also be calculated based on the drizzle DSD near the cloud base), the $S_o$ accounts for the impact of cloud droplets on the potential precipitation ability of the cloud (Terai et al., 2012).

As shown in Figure 7a, the $R_{CB}$ values generally have a negative correlation with increased layer-mean $N_c$ for all three IOPs. The $S_0$ values are 0.979, 1.229, and 1.638, with the absolute values of correlation coefficients being 0.33, 0.29, and 0.45 for SOCRATES, ACE-ENA Sum, and ACE-ENA Win,

respectively. The regression relationships are statistically significant with $p<0.05$ for all three IOPs. These correlation coefficient values fall within the reasonable range found in previous studies on precipitation susceptibility in MBL stratus and stratocumulus clouds (Jung et al., 2016; Gupta et al., 2022), and indicate statistically significant dependences of $R_{CB}$ on $N_c$. Previous study by Terai et al. (2012) found that the $S_o$ values decrease with the increasing cloud thickness over the southeast Pacific, and Jung et al. (2016) found that the $S_o$ is more pronounced within the medium-deep clouds with thickness ~300-400 m in the MBL stratocumulus over the eastern Pacific. Gupta et al. (2022) found that the $S_o$ values are generally higher under low ambient $N_a$ condition in the southeastern Atlantic MBL. In this study, $R_{CB}$ for the ACE-ENA Win is more susceptible to the layer-mean $N_c$ than the ACE-ENA Sum and SOCRATES, which can be partially attributed to the existence of more large drizzle drops (as shown in Fig. 4d) near the cloud base in ACE-ENA Win. As previously discussed, the ACE-ENA Win featured enhanced collision-coalescence suggested by the stronger in-cloud turbulence, and a possible drizzle-recirculating process as indicated by the previous study. And such mechanisms might explain the low $N_c$ conditions with more large drizzle drops, leading to the increase of $S_o$ values during ACE-ENA Win. In comparison, the aerosol of SOCRATES is largely composed of fine Aitken mode aerosol, which results in smaller cloud droplets. Thus, collision-coalescence is ineffective during SOCRATES, which leads to the relatively narrower drizzle DSDs, where the warm-rain processes are suppressed, and in turn, diminishing the sensitivity of $R_{CB}$ to $N_c$ (Stevens and Feingold, 2009; Fan et al., 2020; Gupta et al., 2022).

It is well known that the $R_{CB}$ can be parameterized or predicted by assuming an approximate relation with $N_c$ and cloud thickness ($H_c$), which is usually parameterized in the form of $R_{CB} \propto c\ H_c^3\ N_c^{-1}$ (Lu et al., 2009; Kang et al., 2024). Following the same method, we derive the relationships from three IOPs in Figure 7b, where the $R_{CB}$ are positively (negatively) proportional to the $H_c$ ($N_c$), with the exponential parameters in the range of the typical values in the MBL clouds (Comstock et al., 2004; van Zanten et al., 2005; Lu et al., 2009). The statistical coefficient of determination ($R^2$) values of $R_{CB}$ against

$H_c$ ($N_c$) are 0.696 (0.177), 0.419 (0.212) and 0.165 (0.295), for the ACE-ENA Sum, winter and SOCRATES, respectively, suggesting that the $R_{CB}$ in ACE-ENA clouds may be more determined by $H_c$, while the $R_{CB}$ in SOCRATES clouds could be less dependent on both $H_c$ and $N_c$. Note that the relationship for SOCRATES in this study reveals a similar $R_{CB}$ dependence on $N_c$ but a smaller dependence on the cloud thickness than the study by Kang et al. (2024), who concluded a relationship of $R_{CB} = 1.41 \times 10^{-9}\, H_c^{3.1} N_a^{-0.8}$, based on the rain rate retrieved from radar and lidar measurements and the aerosol concentration also from the SOCRATES. The discrepancies are possibly due to the different sample selections and different methods in the $R_{CB}$ calculation. Note that the mean cloud thicknesses of ACE-ENA Sum (336.3 m), ACE-ENA Win (392.4 m) and SOCRATES (487.4 m), are within the thickness range found to exhibit stronger $S_o$ (Terai et al., 2012; Jung et al., 2016; Gupta et al., 2022).

### 4.3 Drizzle impacts on sub-cloud CCN and implication to ACI

Multiple studies on the MBL clouds have concluded that the in-cloud drizzle formation and evolution processes can effectively impact the sub-cloud CCN budgets via the coalescence-scavenging effect (Wood, 2006; Wood et al., 2012; Diamond et al., 2018; Zheng et al., 2022b; Zhang et al., 2023). Drizzle drops are formed and grow via the collision-coalescence process by collecting cloud droplets and small drizzle drops, resulting in the consumption of CCN (the precursor of cloud droplet), but in the meantime, the in-cloud $N_c$ can be continuously buffered by the sub-cloud CCN replenishment. Although the sub-cloud aerosols (especially in large size) would be added if the drizzle fell and evaporated outside the cloud, the increment cannot compensate for the loss. Therefore, the net result of the whole process is usually presented as the depletion of sub-cloud CCN residuals, and such drizzle modulation on the CCN budget could be substantial in moderate-to-light drizzles or even non-precipitating clouds, depending on the collision-coalescence efficiency (Feingold et al., 1996; Wood, 2006; Kang et al., 2022).

The CCN loss rate due to the coalescence-scavenging effect can be calculated as:

$$L_{CCN} = -\frac{K\ H_c}{H_{cp}} * N_c * R_{CB}, \tag{11}$$

where the constant K (2.25 $m^2$ $kg^{-1}$) denotes the drizzle collection efficiency (Wood et al., 2006; Diamond et al., 2018). $H_c$ is cloud thickness, and $H_{cp}$ is the coupled layer thickness to ensure the change in the cloud layer can be sufficiently conveyed throughout the layer. The calculated CCN loss rate for individual cases is listed in Table S2. Considering all cloud (precipitating cloud) scenarios, the mean CCN loss rates are -7.69 ± 13.96 $cm^{-3}$ $h^{-1}$ (-10.45 ± 15.56 $cm^{-3}$ $h^{-1}$), -6.29 ± 11.65 $cm^{-3}$ $h^{-1}$ (-12.11 ± 14.64 $cm^{-3}$ $h^{-1}$), and -4.94 ± 7.96 $cm^{-3}$ $h^{-1}$ (-5.58±8.43 $cm^{-3}$ $h^{-1}$) for ACE-ENA Sum, ACE-ENA Win and SOCRATES, respectively. As the results indicate, the ACE-ENA clouds experience a more substantial sub-cloud CCN loss than SOCRATES, especially in wintertime precipitating clouds. Recall that the assessment of $ACI_{r,CB}$ relies on the relative changes of $r_c$ and $N_{CCN}$, while the different $L_{CCN}$ for individual cases can result in the shrinking of the $N_{CCN}$ variation ranges (imagine the abundant CCN are depleted by the coalescence-scavenging). In other words, the given change in $r_c$ corresponds to a narrowed change in $N_{CCN}$. Mathematically speaking, the assessment of $ACI_{r,CB}$ depends on the ratio of the numerator (change in $r_c$) and the denominator (change in $N_{CCN}$). Under the circumstances of substantial cloud-processing to the aerosols, the altered sub-cloud CCN budgets are reflected as a smaller denominator, versus the less altered numerator, hence mathematically presented as an enlarged $ACI_{r,CB}$. Therefore, the coalescence-scavenging effect can not only deplete the sub-cloud CCN, but also quantitatively amplify the assessment of cloud microphysics susceptibilities (Feingold et al., 1999; Duong et al., 2011; Jung et al., 2016; Zheng et al., 2022b). In order to examine the potential impact of the aforementioned processes on the $ACI$ assessment, a sensitivity analysis is conducted by simply retrospecting the sub-cloud $N_{CCN0.35\%}$ according to their $L_{CCN}$. For each retrospective time step $\Delta T$, the $r_c$ values are held unchanged, and the retrospective $N_{CCN0.35\%}$ values for individual cloud cases are given by $N_{CCN0.35\%} - L_{CCN} * \Delta T$, and then the $ACI_{r,CB}$ can be recalculated. Note that assuming a constant $r_c$ value over time inevitably induces

uncertainty and biases, as it does not consider the microphysical processes affecting the cloud droplet mean size. However, previous numerical experiments show that the noticeable impact on the cloud mean radius through collision-coalescence necessitates a high degree of CCN depletion, and the quantified percentage changes in droplet mean sizes are several times less than the changes in CCN depletion (Feingold et al., 1996). Hence, the retrospective method, from an observational snapshot point of view, provides a direction that enables the assessment of $ACI_{r,CB}$ as if before the sub-cloud aerosols and CCN are scavenged by in-cloud coalescence-scavenging and precipitation scavenging processes.

As shown in Figure 8, the $ACI_{r,CB}$ values tend to decrease with the retrospective time, which indicates the retrospective CCN variation range is enlarged and counteracting the coalescence-scavenging amplification. The detailed illustration of the different $ACI_{r,CB}$ calculated from the scattered $r_c$ and sub-cloud $N_{CCN0.35\%}$ is shown in Figure S5. Note that the $ACI_{r,CB}$ decreasing rates for the precipitating clouds (Fig. 8b) are not as strong as for all clouds because the non-precipitating clouds have smaller $L_{CCN}$ largely due to weaker collision-coalescence. Hence, the retrospective period used here might quickly exceed the actual time of cloud-processing to become effective on aerosol and CCN. In other words, the actual time needed to trace back to the sub-cloud CCN concentration before they were cloud-processed, is shorter than the retrospective time tested here in Figure 8. This results in the faster decrease of $ACI_{r,CB}$ in the non-precipitating cloud. The retrospective of the sub-cloud CCN budget will yield an alternative assessment of ACI, assuming that the drizzle processes have not yet significantly impacted the sub-cloud CCN budget, especially for the assessment under the precipitating clouds. However, examining the exact precipitating timing is challenging since the aircraft provides a snapshot of the cloud and aerosol information. Thus, this retrospective study only provides a possible direction, and the result should be interpreted with caution.

## 5. Summary and Conclusions

Based on the aircraft in-situ measurements during ACE-ENA and SOCRATES, the vertical distributions and the evolutions of the aerosol, cloud, and drizzle properties are investigated under cloud-topped MBL environments. The aerosols and CCN from SOCRATES are the highest among the three IOPs, followed by ACE-ENA Sum, and ACE-ENA Win in descending order in both above- and sub-cloud regimes. The differences can be attributed to the differences in aerosol size distributions between ACE-ENA and SOCRATES, which are largely due to the aerosol sources in those regions. The SOCRATES features the pre-industrial natural environment enriched by aerosols from marine biological productivity and without the contamination of anthropogenic aerosols, while the ACE-ENA features the aerosols from varied sources, including maritime and continental emissions, with distinct seasonal variations. Examining the aerosol size distributions in sub-cloud versus above-cloud regimes manifests the significant influence of cloud processing on aerosols. According to previous studies, physical processing like in-cloud Brownian capture can reduce Aitken mode aerosols, while the chemical processes transform Aitken mode aerosols to larger sizes, moving them toward the accumulation mode. In addition, the in-cloud coalescence processes could also shift sub-cloud aerosol residuals to larger sizes, as multiple aerosols combine into a single aerosol core inside the cloud droplet during collision-coalescence. Those physical mechanisms could potentially explain the observed increase in the tail of the aerosol size distribution for all IOPs, and it will be of interest for future research to prove such hypotheses.

As for the cloud and drizzle properties, the SOCRATES clouds feature a larger number of smaller cloud droplets than the ACE-ENA Sum and ACE-ENA Win clouds, with the $r_c$ growth (and percent increases), from cloud base to top, being 4.03 µm (0.66%), 4.78 µm (0.68%), and 5.85 µm (0.79%) for SOCRATES, ACE-ENA Sum, and ACE-ENA Win, respectively. The cloud-top entrainment mixing is evident in the observed decline of both $N_c$ and $LWC_c$ near the cloud top. The mean cloud-top entrainment rates ($w_e$) are 0.570±0.834 cm s$^{-1}$, 0.581±0.560 cm s$^{-1}$, and 0.960±1.127 cm s$^{-1}$ for SOCRATES, ACE-

ENA Sum, and ACE-ENA Win, respectively. The strongest $w_e$ during ACE-ENA Win is a result of weaker cloud-top inversions and stronger near-cloud-top turbulence. The values of the TKE for three IOPs are generally within the ranges of previous studies (Atlas et al., 2020; Ghate et al., 2021). For drizzle vertical distribution, $N_d$ from the three IOPs all exhibit decreases from cloud top to cloud base, while $D_{mmd}$ are in opposite directions with a maximum at the cloud base. The ACE-ENA Win clouds feature more prominent drizzle formation and evolution owing to the combined effects of relatively cleaner environment, deeper cloud layer, and slightly stronger in-cloud vertical turbulence, which is speculated to substantially enhance the collision-coalescence and the drizzle re-circulating processes, compared to the other two IOPs. Satellite retrievals of droplet number concentration heavily rely on the adiabatic cloud assumption and are usually given as a constant of $f_{ad}$ = 0.8, the in-situ observational evidence found in this study further confirms the unrealistic nature of this assumption. It will be of interest to utilize multiple aircraft measurements (campaigns) to explore the variability of MBL cloud and drizzle microphysical properties over different marine regions. This can help examine potential predictors for $f_{ad}$, which will aid in satellite-based retrievals and aerosol-cloud interaction assessments (Painemal and Zuidema, 2011; Grosvenor et al., 2018; Painemal et al., 2021).

Comparing the seasonality of cloud-base precipitation rate ($R_{CB}$) during ACE-ENA, more cases with large observed $R_{CB}$ during the winter season, which is consistent with J. Wang et al. (2022). Notably, the sensitivity of $R_{CB}$ to $N_c$ is more pronounced for the ACE-ENA during both winter (with precipitation susceptibility $S_o$ = 1.638) and summer ($S_o$ = 1.229) compared to the SOCRATES ($S_o$ = 0.979). This could be possibly hypothesized as the result of turbulence-driven in-cloud droplet interactions, which could result in much higher $R_{CB}$ induced by larger drizzle drops near the cloud base for ACE-ENA, especially under low $N_c$ conditions. Furthermore, $R_{CB}$ can be approximated by a relationship involving $N_c$ and $H_c$, as suggested in prior research. The relationships established in this study indicate that the $S_o$ in ACE-ENA clouds can be partially determined by $H_c$, while in SOCRATES clouds the $S_o$ is less

influenced by $H_c$ and $N_c$. Based on the physical mechanisms found in the previous study, a possible hypothesis can be leveraged to explain the observed results. That is, the combination of a deeper cloud layer and relatively lower ambient aerosol concentration, eventually leading to stronger drizzle production and evolution during ACE-ENA, especially during the winter season, results in more robust precipitation susceptibility. And further numerical simulations and experiments are warranted to prove this hypothesis. Note that considering the combined factors of aerosol loadings, cloud morphology and thicknesses, and the assessment methodology, the derived $S_o$ values in this study are generally higher (or close to the upper end) compared to previous studies (Lu et al., 2009; Duong et al., 2011; Terai et al., 2012; Jung et al., 2016; Gupta et al., 2022).

The investigations of the ACI via the $ACI_{N,CB}$ and $ACI_{r,CB}$ indices reveal that during the ACE-ENA Win, $N_c$ is more sensitive to changes in $N_{CCN0.35\%}$ , indicating aerosols more readily activate to become cloud droplets compared to those in the ACE-ENA Sum, which is consistent with the previous assessment by J. Wang et al. (2022) on the seasonal dependency of the relationship between $N_c$ and aerosols. One influencing factor is the strong dynamic mechanism that speeds up the infusion of CCN into the cloud layer, thus aiding droplet formation. The moderate but statistically significant correlation coefficients between the CCN activation fractions and the TKE agree with a previous study that found the local activation fraction of CCN to be strongly associated with increased updrafts (Hu et al., 2021). Furthermore, the presence of larger aerosols during ACE-ENA Win enhances the droplet activation process. The SOCRATES IOP highlights a higher $ACI_{r,CB}$, indicating a pronounced decrease in $r_c$ with increasing $N_{CCN0.35\%}$. The $ACI_{r,CB}$ in ACE-ENA is dampened by the presence of more large cloud droplets near the cloud base, particularly under relatively higher $N_{CCN0.35\%}$. However, the combined effect of the relatively cleaner environment and sufficient water vapor results in stronger cloud microphysical responses during the ACE-ENA wintertime than in the summertime. Note that the ACI indices from this study lie in the higher end of the ACI ranges estimated via remote sensing (McComiskey

et al., 2009; Dong et al., 2015; Zheng et al., 2022a) possibly because the aircraft assessment of ACI is based on measurements where the aerosols are in direct contact with the cloud layer. Arguably, the assessment of $N_c$ responses to $N_{CCN0.35\%}$ would inevitably be affected by the collision-coalescence process near the cloud base, where simultaneously, the CCN replenishment buffers the $N_c$ and the collision-coalescence process depletes $N_c$. Hence, finding a layer where these two effects maintain a dynamic balance in $N_c$ might aid in a more accurate assessment and more fundamental understanding of the ACI, which might be revealed by the LES or parcel model simulations.

Additionally, the in-cloud drizzle formation and evolution processes significantly influence the sub-cloud CCN budgets via the coalescence-scavenging effect, which can potentially exaggerate the assessment of cloud microphysics susceptibilities. Based on the CCN loss rate ($L_{CCN}$) from ACE-ENA and SOCRATES, a sensitivity analysis is performed focusing on retrospectively adjusting the sub-cloud CCN according to their $L_{CCN}$. Results showed that this adjustment led to a decreased $ACI_{r,CB}$, highlighting the significance of the coalescence-scavenging process on the ACI assessment. However, due to the fact that aircraft only provide a snapshot of the clouds and aerosol information, determining the precise drizzle timing for the individual cloud is challenging. Hence, findings from this retrospective approach provide only a direction or theory, and should be taken cautiously. Nevertheless, pursuing further modeling experiments on this matter may be worthwhile. For example, the exact drizzling time could be pinpointed within a model using an Eulerian framework or traced using a Lagrangian framework. Nevertheless, the CCN adjustment could more accurately reflect the true characteristics of the cloud and the MBL CCN budget, potentially aiding in a more precise assessment of ACI. Therefore, future research would focus on model simulations of MBL clouds from ACE-ENA and SOCRATES and further assess the modeled ACI under the observational constraints, as well as the continuous development of the warm rain microphysical parameterizations, in order to aid in the better represent the MBL clouds in multiple regions.

*Data availability.* The ACE-ENA field campaign data can be accessed from the Department of Energy Atmospheric Radiation Measurement data archive (https://iop.archive.arm.gov/arm-iop-file/2017/ena/aceena/). The SOCRATES field campaign data are publicly archived on the National Center for Atmospheric Research (NCAR) Earth Observing Laboratory (https://data.eol.ucar.edu/master_lists/generated/socrates/).

*Author contributions.* The original idea of this study is discussed by XZ, XD, and BX. XZ performed the analyses and wrote the manuscript. XZ, XD, BX, TL, and YW participated in further scientific discussions and provided substantial comments and edits on the paper.

*Competing interests.* At least one of the (co-)authors is a member of the editorial board of Atmospheric Chemistry and Physics.

*Acknowledgments.* This work was supported by the NSF grants AGS-2031750/2031751/20211752 at the University of Arizona, Texas A&M University and Stanford University, respectively. The authors sincerely thank the investigators and mentors from the ACE-ENA and SOCRATES field campaigns for making the data publicly available. And a special thanks goes to the editor Tak Yamaguchi, and two anonymous reviewers for their constructive comments and suggestions, which helped to improve the manuscript.

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

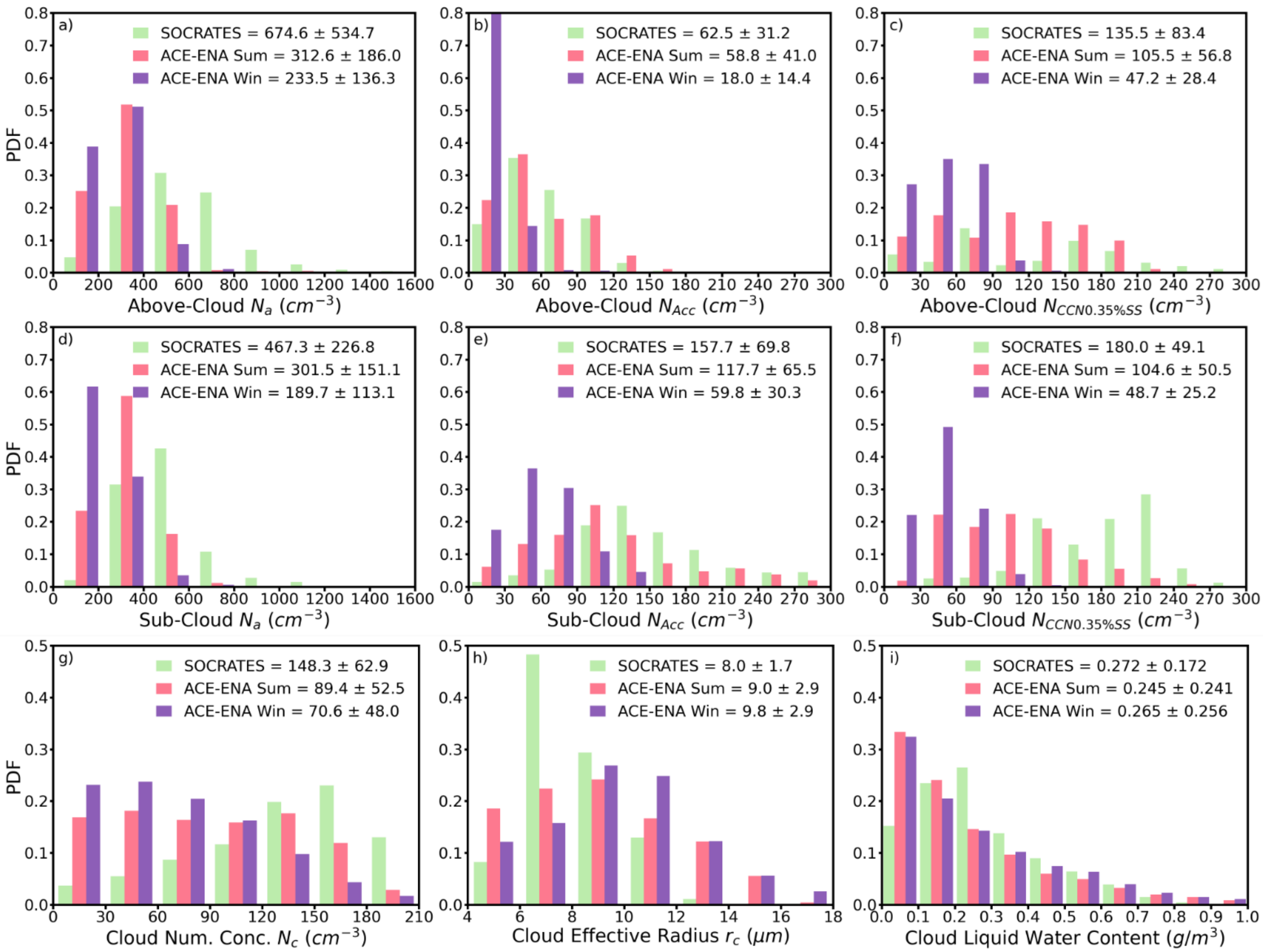

**Figure 1.** Probability Density Functions (PDFs) of $N_a$, $N_{ACC}$ and $N_{CCN0.35\%}$ in the above-cloud (a, b, c) and sub-cloud (d, e, f) regimes; and the cloud microphysical properties of $N_c$ (g), $r_c$ (h), and $LWC_c$ (f) within cloud layer. The statistical metrics in the legends denote the mean and standard deviation values for all samples in three IOPs. The ACE-ENA Sum, ACE-ENA Win, and SOCRATES are color-coded with pink, purple, and green, respectively.

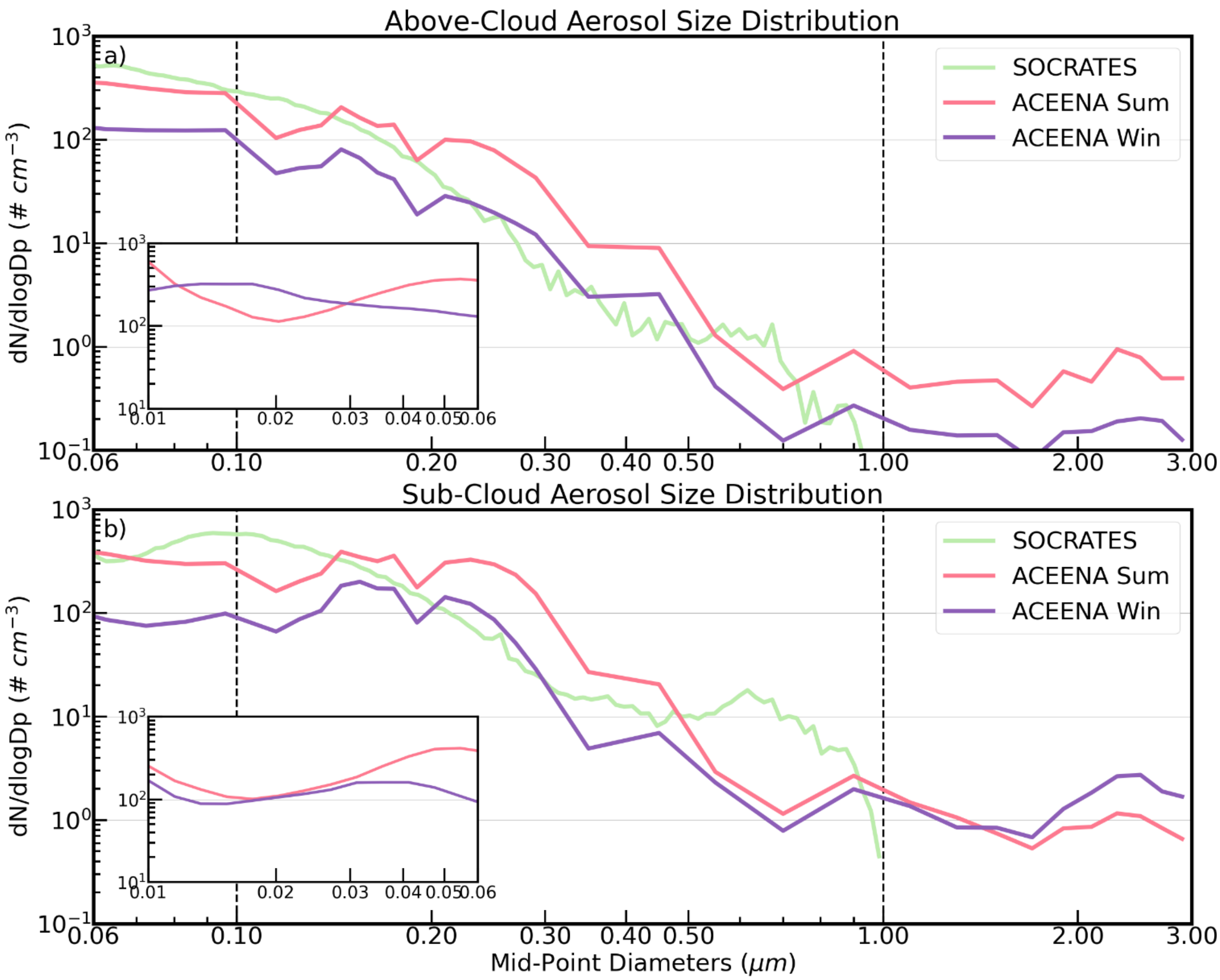

**Figure 2.** Aerosol size distributions ($D_p$ = 0.06 – 3 μm) for above-cloud (a) and sub-cloud (b) regimes. The vertical dashed line at $D_p$ = 0.1 μm and at $D_p$ = 1 μm denotes the demarcations between Accumulation mode, Aitken mode and Coarse mode aerosols. The inner plots denote a smaller range of Aitken mode size distribution ($D_p$ = 0.01 – 0.06 μm) available from ACE-ENA. The ACE-ENA Sum, ACE-ENA Win, and SOCRATES are color-coded with pink, purple, and green, respectively.

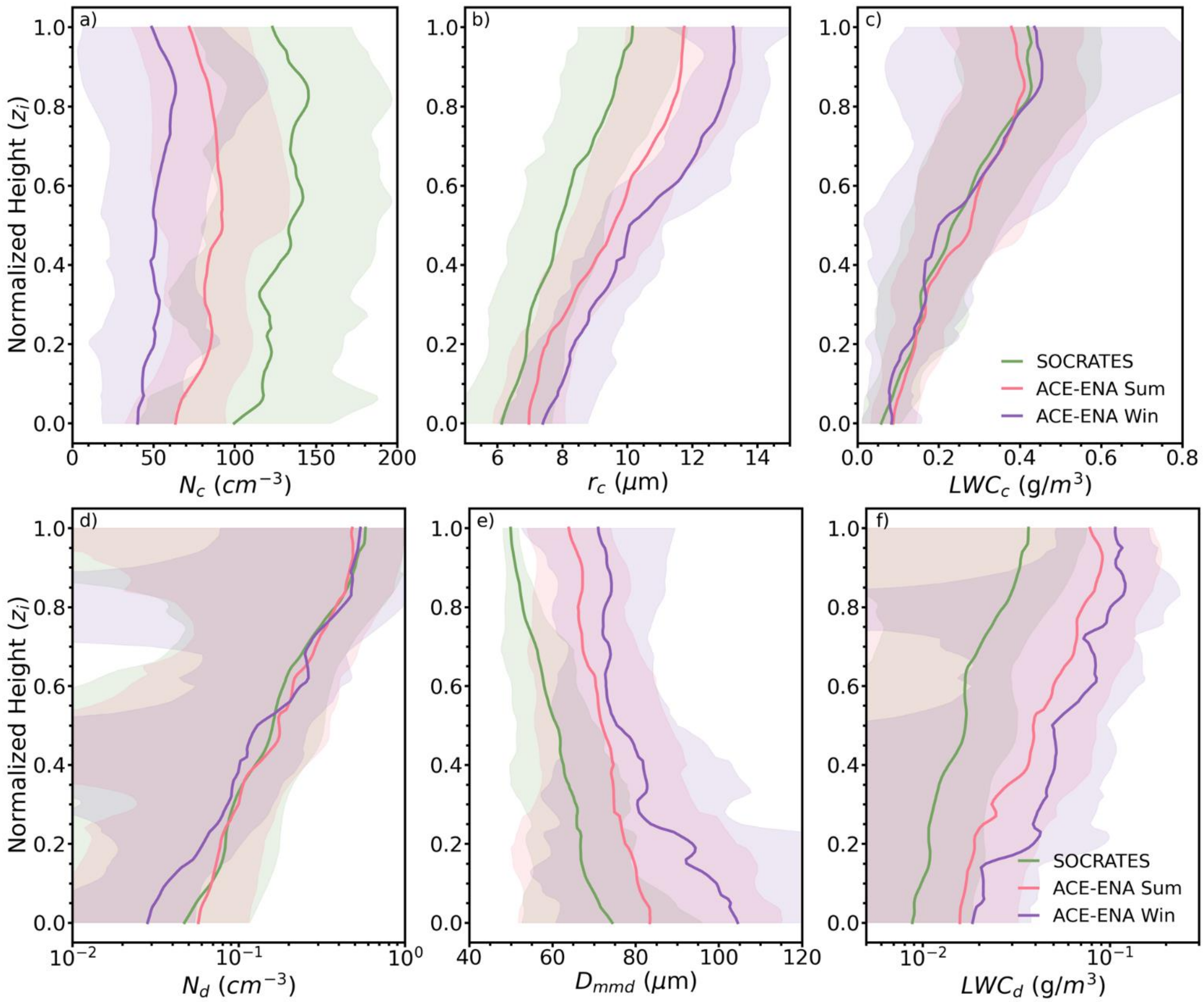

**Figure 3.** Vertical distributions of $N_c$ (a), $r_c$ (b), $LWC_c$ (c), $N_d$ (d), $D_{mmd}$ (e), and $LWC_d$ (f). Here the $z_i = 0$ denotes cloud base and $z_i = 1$ denotes cloud top. Shaded areas denote the inter-cloud-case standard deviations. The ACE-ENA Sum, ACE-ENA Win, and SOCRATES are color-coded with pink, purple, and green, respectively.

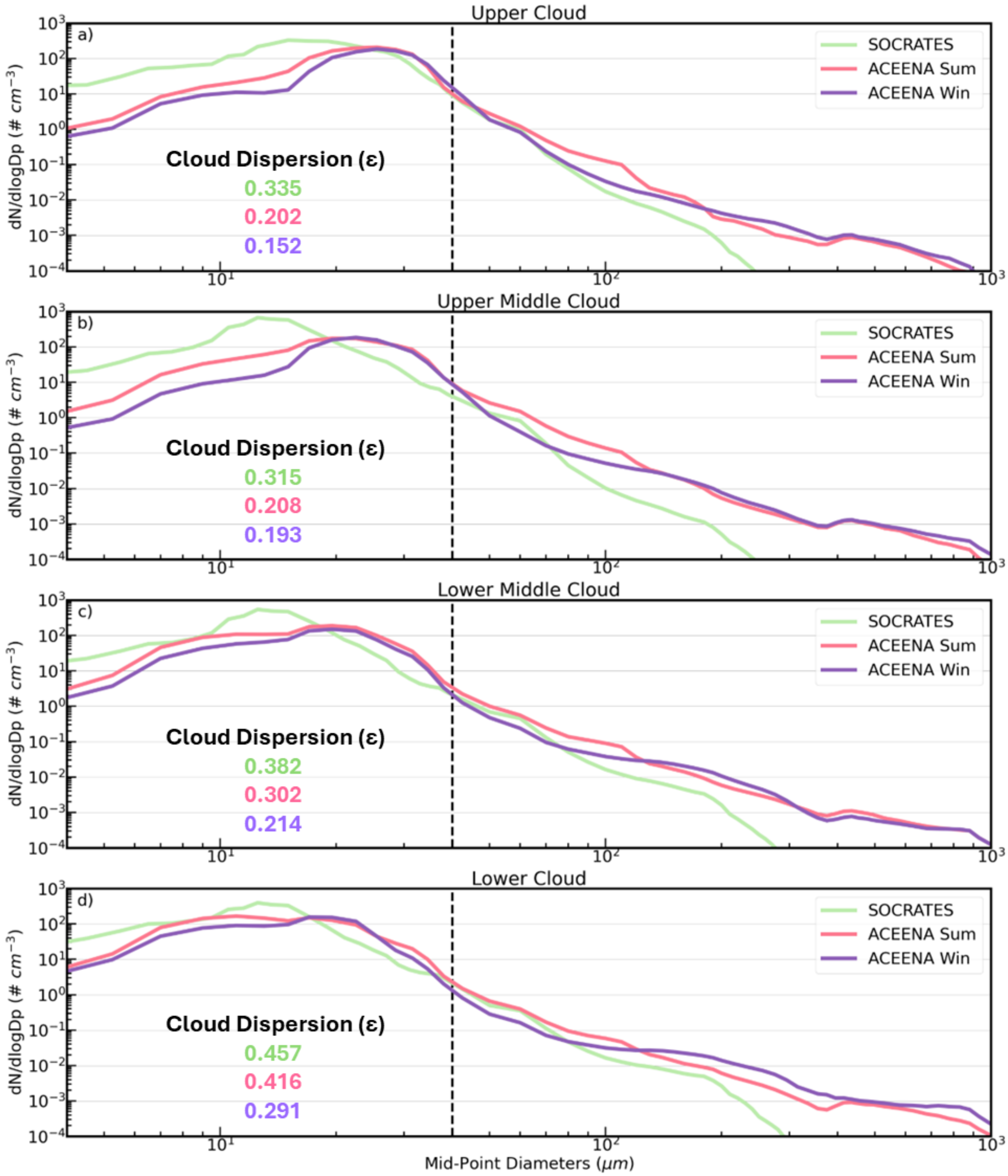

**Figure 4.** Cloud and drizzle size distributions for a) upper cloud ($z_i > 0.8$), b) upper-middle cloud ($0.5 \leq z_i < 0.8$), c) lower-middle cloud ($0.2 \leq z_i < 0.5$) and d) lower cloud ($z_i < 0.2$). The vertical dashed line at $D_p$ = 40 μm denotes the demarcation between cloud droplets and drizzle drops. The ACE-ENA Sum, ACE-ENA Win, and SOCRATES are color-coded with pink, purple, and green, respectively.

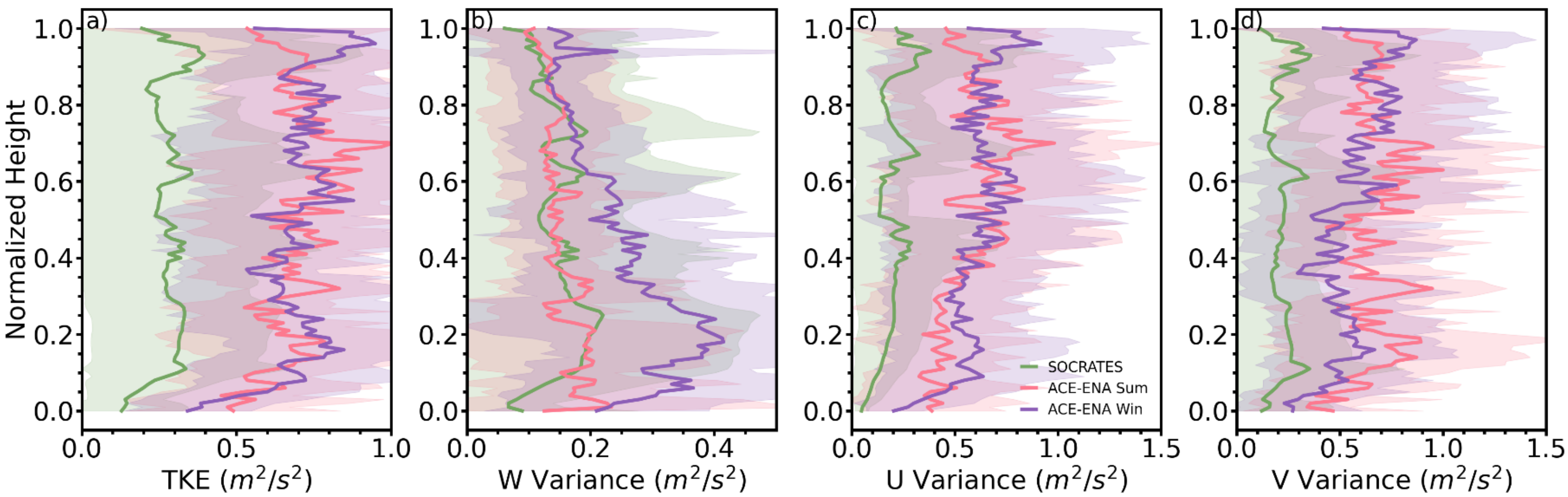

**Figure 5.** Vertical distributions of in-cloud $TKE$ (a), $w'^2$ (b), $u'^2$ (c), and $v'^2$ (d). Shaded areas denote the inter-cloud-case standard deviations. The ACE-ENA Sum, ACE-ENA Win, and SOCRATES are color-coded with pink, purple, and green, respectively.

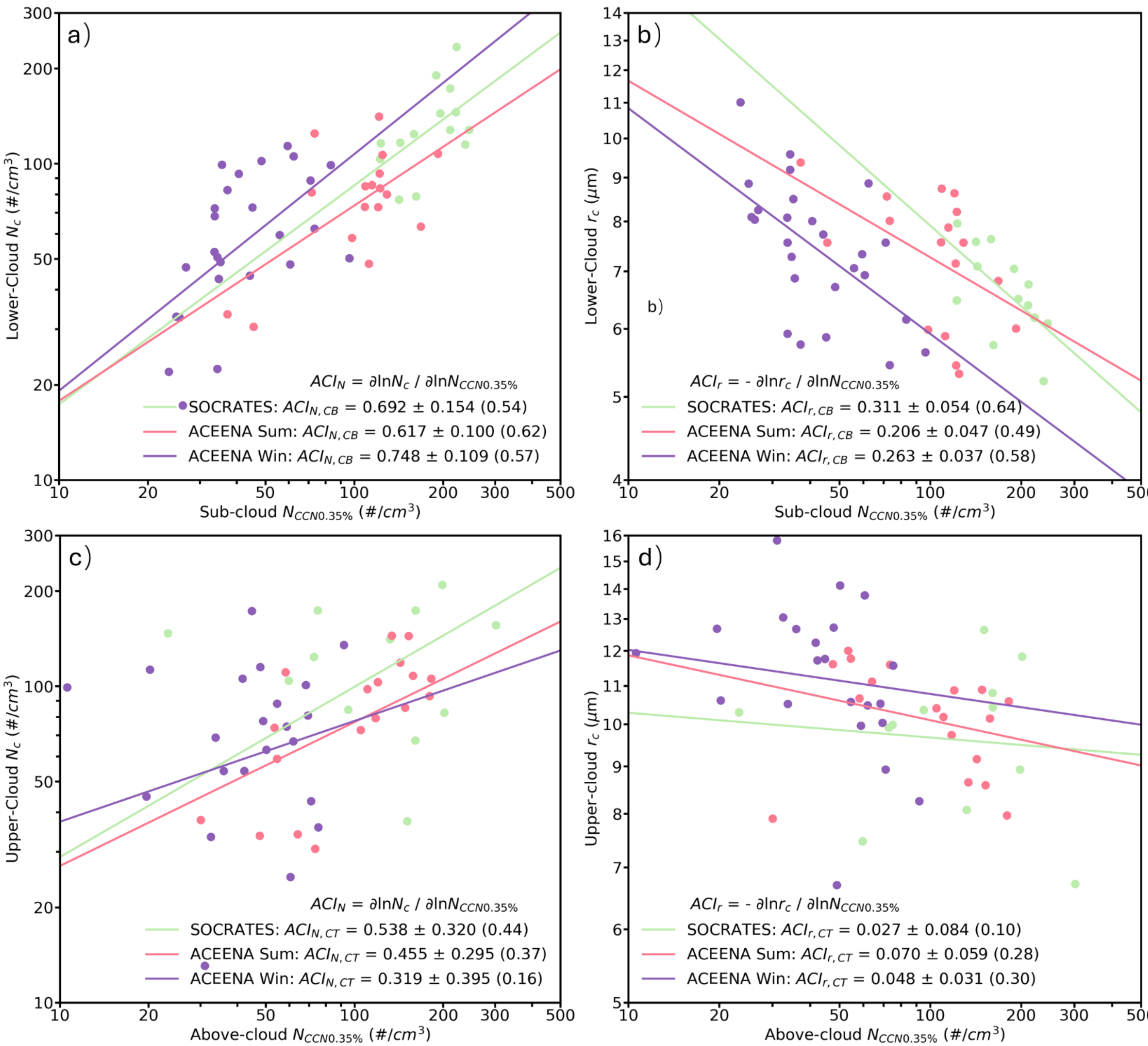

**Figure 6.** Scatterplots of the a) $N_c$ and b) $r_c$ at the lower-cloud ($z_i < 0.2$) against the sub-cloud $N_{CCN0.35\%}$, and the c) $N_c$ and d) $r_c$ at the upper-cloud ($z_i > 0.8$) against the above-cloud $N_{CCN0.35\%}$. The statistical metrics in the legends denote the ACI values and standard errors, and the absolute values of correlation coefficients (in parentheses). The ACE-ENA Sum, ACE-ENA Win, and SOCRATES are color-coded with pink, purple, and green, respectively.

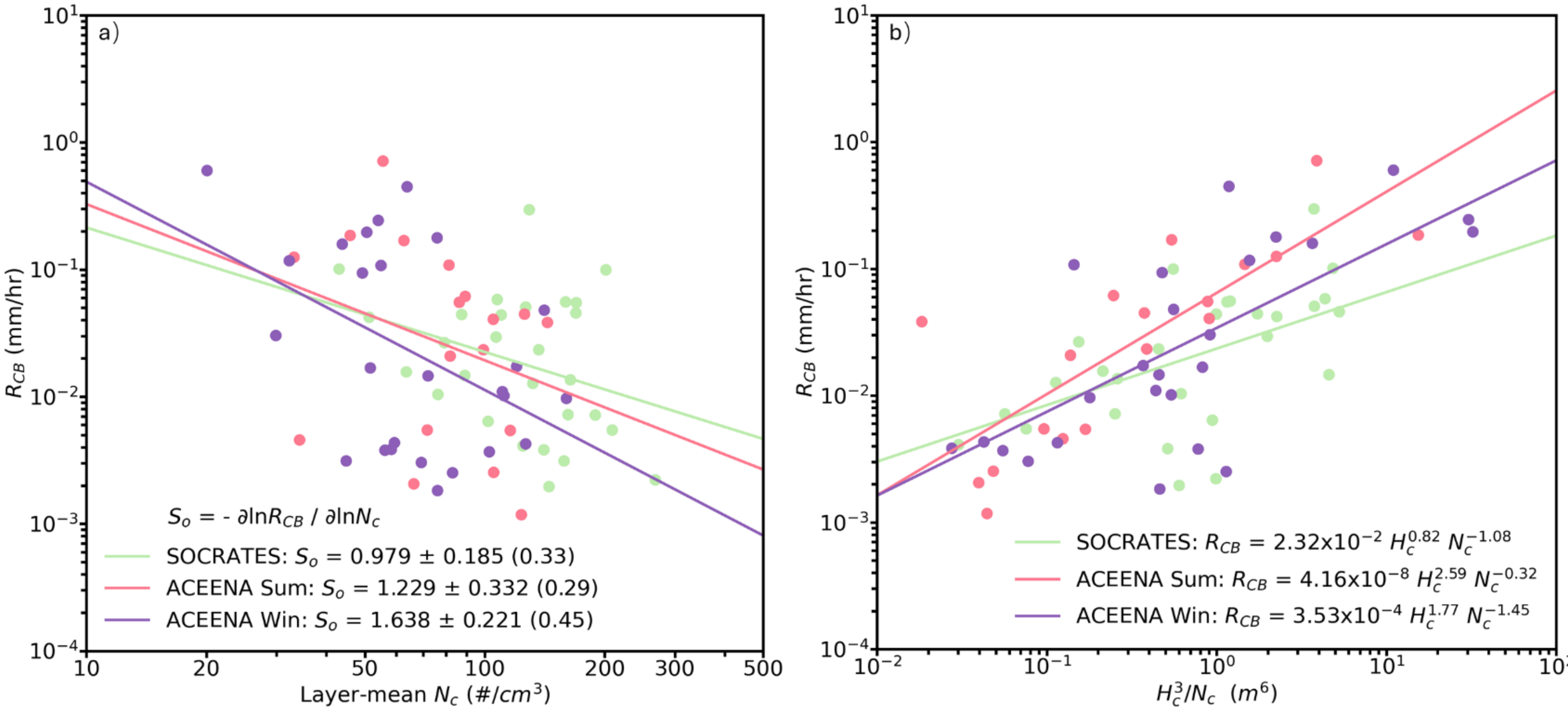

**Figure 7.** Scatterplots of the cloud base precipitation rate $R_{CB}$ against the a) layer-mean $N_c$ and b) $H_c^3/N_c$. The ACE-ENA Sum, ACE-ENA Win, and SOCRATES are color-coded with pink, purple, and green, respectively.

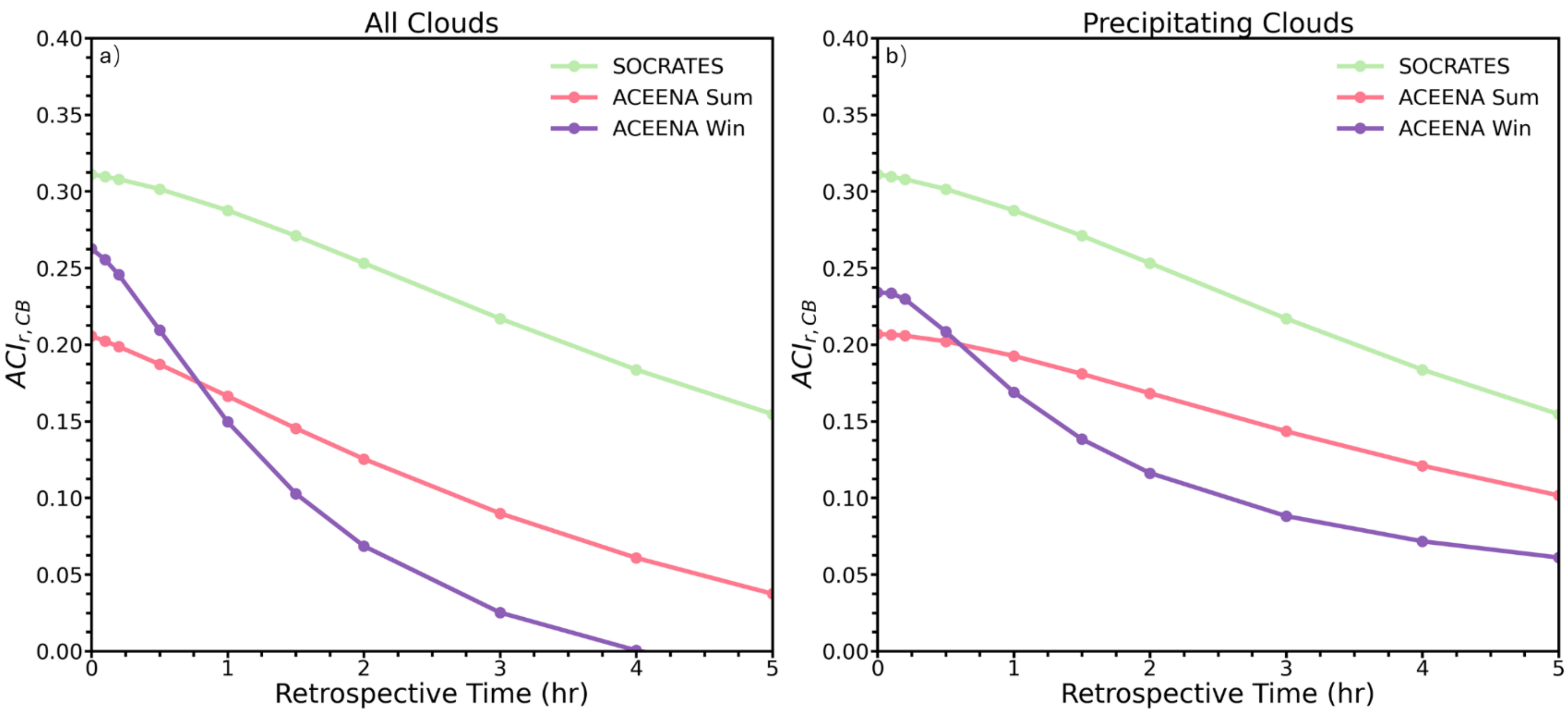

**Figure 8.** $ACI_{r,CB}$ as a function of the sub-cloud $N_{CCN0.35\%}$ retrospective time for a) all clouds and b) precipitating clouds.