# Peer review of "Distinctive aerosol-cloud-precipitation interactions in marine boundary layer clouds from the ACE-ENA and SOCrates aircraft field campaigns"

_EGUsphere, 2023_

## Referee Comment (RC2)

**Review of Zheng et al., "Distinctive aerosol-cloud-precipitation interactions in marine boundary layer clouds from the ACE-ENA and SOCRATES aircraft field campaigns"**

**SUMMARY**

The authors present a comparative study of aircraft in situ aerosol and cloud microphysical measurements of liquid phase boundary layer (BL) clouds from two different regions: over the Eastern North Atlantic (ENA) region near the Azores and the Southern Ocean (SO) in an area spanning south of Tasmania. The ENA measurements are from the summer and winter seasons, while the SO measurements are from summertime only. The overall conclusions are that:
1. Clouds across different regions and seasons have differing collections of microphysical properties
2. These differing microphysical regimes exhibit different susceptibilities to aerosol perturbations
3. Drizzle has a big impact on the BL CCN budget
4. Turbulence plays a leading role in enhanced precipitation seen in the ENA winter regime

The primary data analyzed in the study are aerosol and cloud microphysical properties averaged over all full cloud soundings of warm BL clouds during each campaign. These properties include number concentration (total, modal, fully size-resolved, etc.), measures of drop size distribution (DSD) width, effective radius/mean diameter, liquid water content, sedimentation rate, etc. Many (if not most) of these observations have been analyzed in other recent studies, and indeed many of the conclusions reached by the authors are "consistent with" (or similar language) the results of these other papers. The frequency with which conclusions are followed by such qualifiers gives the impression that there is not much new added by the manuscript. It would be more effective to give a condensed overview of in situ work on the ACE-ENA and SOCRATES campaigns in which you lay out what has already been done. Then in the results section, devote a subsection (or a couple paragraphs, whatever) to explain how your work fits with what's already been published. This would be easier to digest than the piecemeal and repetitive referencing in the manuscript's current state. I do see novelty in the specific focus on interactions among aerosol, clouds and precipitation (abbreviated ACI or ACPI, depending on the context), which to my knowledge have not been addressed in the literature for either field campaign (ACE-ENA and SOCRATES) analyzed.

I have serious concerns about the lack of justification for combining analysis of the field campaigns used here as well as the scientific reasoning leading to the point that turbulence-enhanced collision-coalescence explains more intense drizzle during one campaign (ACE-ENA winter) versus the other two. In fact, I think the differences in precipitation attributed to turbulence can be explained much more simply without appealing to turbulence-microphysics interactions *at all* (see 3[rd] major comment below). While there is clearly a body of legitimate analysis presented in the study, I recommend the manuscript either undergo VERY major revisions or that it be rejected so the authors have sufficient time to rewrite the manuscript before resubmitting it.

**GENERAL COMMENTS**

1. **The unifying thread tying together measurements from the two regions is not clear to me, and I ask that the authors further emphasize/clarify the scientific motivation for combining the campaigns.** This would give a stronger basis for communicating the significance of your results. As it stands, there are no previously unexplored commonalities across the 3 campaigns. Yes, cloud effective radius increases with height for all the campaigns; yes, drizzle mean diameter increases from cloud top to base – these results (among others) are expected, and simply demonstrate that atmospheric physics as we know it isn't completely "broken." But beyond that, what purpose does this comparison serve?

   The aerosol and meteorology driving the clouds in each regime are quite different, so it's somewhat of a trivial conclusion that the microphysical properties differ as well. The argument that "SOCRATES and ACE-ENA both took place in the midlatitudes, so they're directly comparable" is insufficient. The SO region sampled by SOCRATES is more consistently impacted by midlatitude cyclone systems than ENA during summer, which is more often dominated by the nearby Azores high (i.e., ENA is borderline subtropical during summer). In addition, I do not buy that these aircraft campaigns can be taken as "representative" samples of their respective latitude bands/ocean basins; Mechem et al. (2018) show significant interannual variability in the synoptic conditions experienced at the ENA site, and the summer ACE-ENA IOP was characterized by anomalously low BL heights *and* substantial BL decoupling.

2. The authors go to some length to justify their assertion that turbulence-enhanced collision-coalescence is the reason for stronger precipitation during winter at ENA, but the evidence given does not prove their hypothesis. **The discussion of TKE is illustrative but quantitatively insufficient.** For one, very few details are given on how the velocity perturbations are calculated; for example, what is the integral length scale obtained with a 10 s moving window (i.e., are you capturing the inertial subrange)? Is it the same for ACE-ENA and SOCRATES? (no) Is any window function applied or is this a simple "boxcar" moving average? Is 1 Hz data used or did you analyze high-rate data? Is "high-rate" the same for ACE-ENA and SOCRATES? (no) In addition, TKE is not the relevant quantity for evaluating turbulent enhancement of collisional growth; rather, it is the TKE dissipation rate $\varepsilon$ that is used in parameterizations (e.g., Grabowski and Wang 2013 as referenced in the manuscript). For another, $\epsilon \sim O(10^{-4}~\mathrm{m^2/s^2})$ in stratiform BL clouds while in shallow cumulus it is about an order of magnitude higher. Based on the sampling goals of both ACE-ENA and SOCRATES, mostly stratiform clouds were sampled, suggesting generally low turbulence intensities. A past modeling study on the feasibility of turbulence to overcome the "warm rain bottleneck" and accelerate drizzle formation via collision-coalescence enhancement in subtropical marine stratocumulus (Sc) showed a minor impact (Witte et al. 2019, doi: 10.1175/MWR-D-18-0242.1) – why is a different answer expected in the same cloud dynamical regime? Finally, cold pools are a dynamical forcing mechanism more prevalent during ACE-ENA winter than either of the other 2

campaigns; this is not mentioned at all.

For you to continue pushing the line of reasoning that turbulence directly *causes* stronger precipitation, at very minimum you need to demonstrate that $\varepsilon$ is substantially stronger for the ACE-ENA winter campaign than both typical marine Sc *and* SOCRATES/ACE-ENA summer (i.e., much greater than the 30-40% difference in mean cloud-top TKE shown). If you are unable to demonstrate this, I cannot support the heavy reliance on turbulence-enhanced collision-coalescence peppered throughout the manuscript. If you move forward with quantifying $\varepsilon$, please explicitly detail your approach in the methodology section – I recommend Siebert et al. (2010, doi: 10.1175/2009JAS3200.1) or Waclawczyk et al. (2017, doi: 10.5194/amt-10-4573-2017) as starting points for developing your own analysis.

I will note that your point that the *activation fraction* of available CCN to cloud droplets is highly correlated with turbulence intensity (as stated in the abstract, albeit differently worded) is valid, but this is not quantitatively demonstrated either; you could adopt a framework as in Hu et al. (2021, doi: 10.1029/2021JD035180) to explore this point further.

3.  Beyond the lack of quantitative evidence supporting the hypothesis that turbulence is the cause of increased drizzle production during ACE-ENA winter, **there is a simpler, more parsimonious explanation that I believe is given short shrift in the manuscript: that a combination of a deep cloud layer along with relatively clean aerosol conditions during ACE-ENA winter results in robust drizzle generation**. You do discuss the relationship between cloud depth and precipitation susceptibility very briefly in section 4.2, but it essentially reads as a footnote versus a primary point. Given the strong dependence of cloud base rain rate on cloud depth in the empirical relation discussed in this same section (4.2), it was a major oversight that you didn't explore this aspect further.

4.  The discussion of the role of thermodynamic decoupling is incomplete, but decoupling ostensibly plays a significant role in the low values of $f_{ad}$ encountered during all three campaigns. It would be well worth taking the next step and directly quantifying at least one decoupling metric from the observed profiles as defined by Jones et al. (2011).

5.  I am not familiar with "retrospecting" as discussed in section 4.3 and shown in Fig. 8. What is the procedure for performing this analysis? Please explain in the manuscript, as this appears to be simultaneously one of the most tentative aspects of the paper as well as something the authors are rather excited about.

**SPECIFIC COMMENTS**
Each comment refers to a specific line/passage, figure, table or caption. Specific line(s) are denoted by LXX (or LXX-YY for longer passages).

L24-26: the lack of sensitivity of precipitation to aerosol during SOCRATES suggests that it inhabits a different microphysical regime than ACE-ENA. In other words, there are sufficient CCN during SOCRATES that the ACPI are effectively "saturated" with respect to increasing aerosol loading. There are numerous references discussing such buffering effects (a good starting point is Stevens and Feingold 2009, doi: 10.1038/nature08281) that I recommend the authors consult to reframe this discussion.

L154-157: with respect to what are the "changes" in $\theta_l$ and $q_t$ evaluated? The mean of the cloud layer or surface layer? Based on Fig. S1, I assume cloud layer. I am confused by this definition because "mixed layer" typically implies surface mixed layer, while you are using it to describe an elevated mixed layer. In general, I found your analysis of decoupling to be incomplete.

L191-195: When you discuss "airmass origin," at what vertical level(s) are back trajectories taken? I believe you in terms of PBL airmass, but is this also true above the PBL?

L192 and elsewhere: Zhang et al. (2023) reference is missing in bibliography – I assume the correct reference is from (most of) the same authors in Atmosphere (doi: 10.3390/atmos14081246)?

L216-217: please add a location for the further discussion, i.e., "and will be further discussed in Section X.X" or "further discussed later in this section"

L259: are there any measurements that support your assertion that it's both coalescence-enlargement and sea salt contributing to heightened concentration at Dp>1 um?

L268-269: I don't think you've improved understanding of the first indirect effect. Rather, you're adding another data point that supports what we already understand about it. So it's more of a "confirmation" than anything novel.

L296: Verb tense disagreement. Recommend you make everything present tense: "...air **is** entrained into the boundary layer..."

L300-302: Does this difference in $r_c$ profiles tell you anything about mixing regime (i.e. homogeneous vs. inhomogeneous)?

L304: I can't imagine the water vapor source from entrainment evaporation is a leading order term in the BL $q_t$ budget, and I'm having some difficulty understanding the relevance of raising this point. As you state, the net impact of entrainment on BL $q_t$ should be negative (i.e., entrainment mixes in drier air, so BL-mean $q_t$ should decrease), which would imply this evaporative source is a relatively minor offset to the entrainment drying sink. Eyeballing it from Fig. 3c, it looks like there's maybe 0.2 g/m3 of vapor that is liberated from the clouds (extrapolating the midcloud $q_l$ lapse rate to $z_i$=1) – but I can't assess this any further since you don't show any mean $q_t$ or $q_v$ profiles. Both the G1 and the GV have open path hygrometers

from which $q_v$ can be accurately measured in cloud – if you want to get into a discussion of the vapor budget, it would be helpful to explicitly show some of these measurements.

L305-310: What evidence do you have for re-condensation beyond the inferences made from bulk profiles? And shouldn't there be *more rapid* growth on smaller drops since condensation rate is inversely proportional to surface area? You have the full DSDs to demonstrate the validity of the generalizations you're drawing. Please evidence for these assertions.

L313: What is gained by quantifying $\Delta r_c$? Is this not just a different way of expressing the subadiabatic $q_l$ lapse rate via the relation $r_e = k r_v$ where $r_v \propto (q_l/N)^{1/3}$ ?

L337-340: Please define what terms are being used to calculate the "reduction of $LWC_c$" – it's not clear to me what you're doing here.

L368-369: Water vapor competition matters in a water-vapor limited regime (which seems quite obvious when stated that way…), but it seems to me that ACE-ENA winter is more of an aerosol limited regime. Appealing to water vapor competition is not a "one size fits all" conclusion that can be universally applied.

L370: do you quantify skewness or is this a qualitative description?

L371-372: can you say with certainty whether coarse mode aerosols are drizzle residuals vs. "primary production" of sea salt from the surface? Seems like a difficult "chicken and egg" problem to assess from in situ data without either aerosol composition information or some modeling work to back up the statement.

L436: I assume you mean liquid water *content*, but this is not stated

L490-491: What are the uncertainties in $S_0$? The correlations do not look very strong in Fig. 7a.

L514: Double check the equation, it looks like something is not formatted correctly or there are some extra characters.

L536: Should there be a minus sign in the 2nd parenthetical?

L538-539: This sentence needs to be restructured, it is currently a fragment.

L542-543: you could expand upon this point more, it's a bit too concisely expressed to be easily understood.

L569: it is rather counterintuitive that the "pristine" environment has the strongest aerosol loading. This is paradoxical because we often use the terms "pristine" vs "polluted" to imply low vs. high aerosol loading, respectively. You clearly mean it in the sense that the SO region is

minimally impacted by anthropogenic emissions. So a little word-smithing is needed to resolve this incongruity.

L594: fully agreed that the assumption of constant $f_{ad}$ used in satellite $N$ retrievals is problematic, but how do the campaign-average profiles presented here improve this situation? On a profile-by-profile (or, from the satellite perspective, pixel-by-pixel) basis, is there anything from the measurements that suggest potential predictors of $f_{ad}$? Or do you view your contribution as simply another data point showing that an assumed value of 0.8 is unrealistic?

Figure 2: This figure could be compressed in the horizontal, which would accentuate the shape of the distribution in a manner pleasing to the eye. As it currently stands, this looks "stretched out" and there aren't many interesting detail/wiggles in any of the curves that merit such a long aspect ratio.

Figure 3: What do the shaded regions denote? Interquartile range? Standard deviation? 5th-95th percentiles? No info in caption.

Figure S2: put the two panels on the same plot so they can be directly compared.

Figure S3: why does this size range needs to be separated from Fig. 2 in the main manuscript?

Figure S4: please add uncertainty shading as you did in Fig. 3 of the manuscript, it would be helpful to see the variability of subadiabaticity within campaigns

Table S1: Please include f_ad in this table.

Table S2: it looks like this table is cutoff. Are there more variables not shown?

---

## Author Comment (AC1)

**Response to Reviewer #1**

We appreciate your time and effort in thoroughly reviewing our manuscript. We are truly grateful for your constructive comments and insightful suggestions, which encourage and help us to improve the manuscript. We have revised the manuscript carefully based on your comments. In the response below, your comments are provided in black text and our responses are provided in blue text.

**Response:**

**GENERAL COMMENTS:**

This study uses aircraft in-situ measurements from the ACE-ENA and SOCRATES field campaigns to illustrate vertical profiles of cloud microphysical and precipitation properties and their relationships with above- and below-cloud aerosols. The paper is well-written with appropriate references. There is tremendous detail in Section 2 (uncertainties in airborne observations, formulae used to calculate cloud properties, thresholds used to define in-cloud and above/below-cloud regions) to ensure the results can be reproduced. The discussion of the results is well structured, and the text is backed up with appropriate figures. The study draws proper conclusions and provides appropriate evidence. This is a comprehensive study that deals with numerous elements of aerosol-cloud-precipitation interactions – a single paper is used to show findings that could have spread across multiple studies if the individual elements were investigated further.

While the work is comprehensive, the novelty of this study comes from the fact that these in-situ observations come from regions that have not been sampled extensively by aircraft or the data examined in detail. Additionally, very distinct regions with unique cloud and aerosol characteristics are compared. In isolation, many findings might not strictly be new as they corroborate results from many previous studies that have used aircraft observations. Nevertheless, this study is an important addition to the literature because of the wide range of topics discussed and the fact that the authors contrast cloud and aerosol properties from multiple field campaigns from two different locations. The effort put into this work is

commendable and the paper would be a valuable addition to the journal and the literature. Given the number of topics discussed and potential for deeper dives into each topic, I can see multiple studies coming out of further investigation into the results presented here.

Thanks for the thorough assessment of our manuscript, we have tried our best to carefully consider and address your comments and suggestions.

I have some considerations for the authors before the study is published:

1. The introduction felt a bit too broad and generic - the authors could better motivate and highlight the unique aspect of this work by guiding the reader through the distinct nature of these regions, by contrasting the existing knowledge of cloud properties from these regions - somewhat done for SOCRATES but not for ACE-ENA, highlighting the 'climatic significance' of clouds observed in these two regions – this was only done for MBL clouds in general, and the need for in situ observations from these locations.

Is there a reason behind comparing these two specific field campaigns? Are we comparing apples to apples? - describing the cloud regime, type, or morphology in these regions could help interpret the differences in the results presented in later sections.

We have reconstructed our discussion to review the previous works on ACE-ENA and SOCRATES in the introduction, and our results in comparison with the previous study in the summary and conclusion section. Please refer to the revised manuscript.

Furthermore, to demonstrate the feasibility and justification of combining these two campaigns, we calculated the composite of the selected case periods during the ACE-ENA and SOCRATES using the hourly ERA5 reanalysis, as shown in Figure S1 below. The composite 850 hPa geopotential height is denoted by black contours, and the height anomaly from the 20-year climatology (2000-2020) is portrayed by the shaded area.

The discussion in the introduction has been modified to better motivate our study, as follows:

[revised manuscript text omitted]

2. The authors could better and use terminology to refer to cloud processes discussed throughout the text. Terms like "collision-coalescence", "in-cloud coalescence", "coalescence-scavenging" are used (often interchangeably it seems) - they can be merged or defined using more commonly used terminology.

Line 57: the authors could separate what they term the 'coalescence-scavenging effect' into two parts. While the drizzle drops are within the cloud layer, the process described in Line 58 can be described as the 'collision-coalescence' process. Once the drizzle drops are below cloud base, the process described in Line 59 can be described as the 'precipitation scavenging' process. Using such terminology would prevent confusion caused by using the same term for two separate processes that occur in-cloud and out-of-cloud, respectively. Please ensure consistency in using the terms as the paper currently uses "collision-coalescence process" in some spots and different terms at other spots while describing a similar process.

We have changed the term 'in-cloud coalescence' to 'collision-coalescence' throughout the paper. Additionally, we have added more explanation on 'coalescence-scavenging' in order to distinguish it from the precipitation of wet scavenging outside the cloud.

'Conversely, precipitation has been shown to exert a substantial influence on the MBL aerosol and cloud condensation nuclei (CCN) budget through the coalescence-scavenging effect, as multiple aerosols combine into a single aerosol core inside the cloud droplet during collision-coalescence. As the drizzle drops descend, they are enlarged by collecting more cloud droplets within the cloud layer. However, the drizzle drops, once falling out of the cloud base, can result in net reductions in sub-cloud aerosols and CCN budgets also via the precipitation scavenging processes'.

**SPECIFIC COMMENTS:**

ABSTRACT:

Line 15: Could you provide quantitative estimates with the number and size used to state "larger number" and "smaller cloud droplets"?

We have added those quantities in this sentence:

'SOCRATES clouds have a larger number (148.3 cm$^{-3}$) and smaller cloud droplets (8.0 µm) compared to ACE-ENA summertime (89.4 cm$^{-3}$ and 9.0 µm) and wintertime clouds (70.6 cm$^{-3}$ and 9.8 µm).'

SECTION 1:

Line 34: The authors mention 'cloud-top longwave radiative cooling' here which is very important and should be mentioned but it is not discussed later. In contrast, there is an excellent discussion of cloud top entrainment mixing and droplet evaporation near cloud top later, but it is not introduced here. The authors could better motivate their results by introducing cloud processes in Section 1 if they are discussed later.

We have added the discussion on the cloud-top longwave radiative cooling and entrainment mixing in the introduction as follows:

'Precipitation, particularly in the form of drizzle, is common in MBL clouds (Wood et al., 2015; Wu et al., 2020), and the turbulence forced by stratocumulus cloud-top radiative cooling can increase the cloud liquid water path, and contributing to drizzle production (Ghate et al., 2019, 2021). The drizzle formation and growth processes are deeply entwined with the MBL aerosols and dynamics. Aerosols have been found to suppress the precipitation frequency and strength by constantly buffering cloud droplet number concentrations via activation, hence increasing cloud precipitation susceptibility (Feingold and Seibert, 2009; Lu et al., 2009; Sorooshian et al., 2009; Duong et al., 2011). Furthermore, the assessments of precipitation susceptibility are examined to be under the influences of methodology (Terai et al., 2012), cloud morphology (Sorooshian et al., 2009; Jung et al., 2016), ambient aerosol concentrations (Duong et al., 2011; Jung et al., 2016; Gupta et al., 2022), and cloud thickness (Terai et al., 2012; Jung et al., 2016; Gupta et al., 2022). The in-cloud turbulence and wind shear can effectively enhance collision-coalescence efficiency, stimulating drizzle formation and growth, and consequently leading to enhanced precipitation (Chen et al., 2011; Wu et al., 2017). Cloud-top entrainment of dryer and warmer air can potentially deplete small cloud droplets and shrink large droplets via

evaporation, thereby impacting cloud top microphysical processes depending on the homogeneous or inhomogeneous mixing regimes (Lehmann et al., 2009; Jia et al., 2019).'

Line 42: This statement is only true under conditions of comparable cloud water content. Please update accordingly and provide appropriate references.

We have changed this statement to:

'…the aerosol-cloud interaction (ACI), can be typically viewed as decreased cloud droplet effective radii ($r_c$) and increased number concentrations ($N_c$) with more aerosol intrusion, under conditions of comparable cloud water content (Feingold and McComiskey, 2016).'

Line 44: Instead of listing references at the end of the sentence, a reader would benefit much more if references for specific elements followed the corresponding text. For example, "…investigated by different observational platforms, such as aircraft (Diamond et al., 2018; Painemal et al., 2020), model simulations (Hill et al. 2009)…". Please directly state some of the "different maritime regions". Something like "….over different maritime regions like the southeast Pacific (Painemal and Zuidema, 2011), northeast Pacific (Braun et al., 2018), southeast Atlantic (Diamond et al., 2018), and eastern North Atlantic (Zheng et al., 2022a)."

We have changed this discussion accordingly:

'The ACIs have been extensively investigated by different observational platforms, such as aircraft (Hill et al., 2009; Diamond et al., 2018; Gupta et al., 2022), ground-based and satellite observations (Painemal et al., 2020; Zheng et al., 2022a), and model simulations (Wang et al., 2020) over different maritime regions like the southeast Pacific (Painemal and Zuidema, 2011), northeast Pacific (Braun et al., 2018), southeast Atlantic (Gupta et al., 2022), and eastern North Atlantic (Zheng et al., 2022a).'

Line 53: More recent studies of precipitation susceptibility (e.g., Gupta et al., 2022; Jung et al., 2016; Terai et al., 2012) have been built upon the studies cited here. To my knowledge, Feingold and Seibert (2009) introduced the term precipitation susceptibility

and the study should be cited here. I see the authors briefly compared their results with some of these previous studies (good to see) - but the studies should be cited in the introductory text. Also suggest discussing the different issues with estimating/interpreting $S_o$ based on results from previous studies.

The suggested references (Gupta et al., 2022; Jung et al., 2016; Terai et al., 2012) are added to the text.

'The assessments of $S_o$ are examined to be under the influences of methodology (Terai et al., 2012), cloud morphology (Sorooshian et al., 2009; Jung et al., 2016), ambient aerosol concentrations (Duong et al., 2011; Jung et al., 2016; Gupta et al., 2022), and cloud thickness (Terai et al., 2012; Jung et al., 2016; Gupta et al., 2022).'

Line 75: Please also provide the months for the ACE-ENA IOPs along with the years.

This information is added:

'…the recent Aerosol and Cloud Experiments in the Eastern North Atlantic (ACE-ENA) aircraft campaign (J. Wang et al., 2022) were conducted in the summer (June and July) 2017 (ACEENA Sum) and winter (January and February) 2018 (ACEENA Win).'

Line 80: Please provide the duration for the SOCRATES austral summer IOP.

This information is added:

'…the Southern Ocean Clouds Radiation Aerosol Transport Experimental Study (SOCRATES) field campaign (McFarquhar et al., 2021) was conducted during the austral summer (January and February 2018),'

SECTION 2:

I want to commend the authors for the discussion in this section. However, the section does lack some context about the cloud sampling locations. A map of the sampling

locations/flights tracks or a list of the range of latitude-longitude coordinates where clouds were sampled during the IOPs would be very useful.

Thanks for your recognition.

We have added Figure S1 to illustrate the meteorological patterns of the study domains, and the sampling locations of the selected cases are indicated by the white dots in Figure S1.

Line 106: I believe by "resolutions" you actually mean - "the size bins of the probe were 1 to 3 um wide"?

The description is changed as follows:

'The Fast Cloud Droplet Probe (FCDP) onboard aircraft during ACE-ENA can detect droplets with diameter ($D_p$) ranging from 1.5 μm to 50 μm, with the size bins of the probe between 1 and 3 μm (Glienke and Mei, 2020). While the SOCRATES used a similar CDP to measure droplets from 2 μm to 50 μm at a 2 μm probe size bin width.'

Line 111: Can you provide a reference for the phase classification product? If not, a short description of the methodology for it would be useful.

This information is added:

'…the University of Washington Ice–Liquid Discriminator product, which is a Machine-learning-based single-particle phase classification of the 2DS images (Atlas et al., 2021)'

Line 128: Can cite Hansen and Travis, 1974 where the term "effective radius" and the associated formula were introduced.

We have added this reference to the revised manuscript.

Line 160: Do your results depend on the selection of the value of 200 m to determine the distance for above-cloud aerosols that are important for the analysis? Gupta et al. (2021) conducted a sensitivity test to determine if their analysis of cloud microphysical properties was affected by this number for distance from above-cloud aerosols. A comment on the sensitivity of the results on this value would be useful.

We have added the discussion as follows:

'The above-cloud aerosols and CCN are selected between the cloud top and 200 m above. Note that the selection criteria of 200 m above the cloud top would inevitably induce uncertainty in the cloud top ACI assessment, depending on the vertical trend of the individual aerosol profile. Over the Southeast Atlantic, Gupta et al. (2021) conducted an analysis focusing particularly on the differing impacts when biomass burning aerosols are in contact with marine stratocumulus cloud tops, using 100 m above as the demarcation, versus when they are separated by various distances, and found that significant differences were observed in cloud microphysics, owing to different droplet evaporation and nucleation, compared to separated profiles. That result is in agreement with the modeling sensitivity study over the Eastern North Atlantic by Wang et al. (2020), who found that aerosol plumes can exert impacts on the cloud-top microphysics only when they are in close contact with the cloud layer. In most cases, the ACE-ENA feature is a rather stable or slightly decreasing profile within a couple hundred meters above the cloud top, while the long-range transports, particularly during summertime, will induce an elevated aerosol layer in higher altitudes that is not in contact with the cloud layer. While the frequent new particle formation events during SOCRATES will significantly alter the free-troposphere Aitken mode aerosol budget, they would need to further subside down to impact the cloud (McCoy et al., 2021; Zhang et al., 2023). Therefore, the 200 m criteria used in this study are in the reconciliation of getting the close-to-cloud aerosol plumes and enough sample size for statistical analysis.'

SECTION 3:

Line 192: These aerosol concentrations for SOCRATES seem high for the Southern Ocean, which was typically viewed to be a pristine environment. The authors refer to previous studies from SOCRATES to explain the aerosol size distribution or composition, do these studies have similar aerosol concentrations?

We have added the discussion as follows:

'Previously, McCoy et al. (2021) reported average values of 680.69 cm$^{-3}$, 546.28 cm$^{-3}$ and 465.05 cm$^{-3}$ for mid-troposphere, above and below cloud for the multiple SOCRATES cases, respectively. While for the individual cases the above cloud aerosols vary from a couple hundred to over a thousand (McCoy et al., 2021; Zhang et a., 2023). These aerosols are predominantly produced from the oxidation of biogenic gases, notably the dimethyl sulfide (DMS) emitted by marine biological productivity (Sanchez et al., 2018; McCoy et al., 2020). The rising air currents in MBL transport these particles into the free troposphere (FT) with dominant aerosol population over the SO (McCoy et al., 2021; Sanchez et al., 2021). And hence, it reinforces the notion that the SO represents a pre-industrial marine environment where the influence of anthropogenic and biomass-burning aerosols is mostly negligible (McCoy et al., 2020, 2021).'

Line 213: Do you mean that the 'sub-cloud Nacc values' are more than double the 'above-cloud Nacc values'? Suggest rewording this sentence.

We have reworded this sentence as:

'Notice that the sub-cloud $N_{Acc}$ values from three IOPs are more than double of the above-cloud $N_{Acc}$ values, and most of the sub-cloud accumulation mode aerosol can be activated to become CCN at SS of 0.35%.'

Line 268: "These results have further improved the understanding of the aerosol first indirect effect". This statement is a bit overreaching. There are many studies that have

shown similar results. I suggest rewording to "These results are consistent with the understanding of the aerosol first indirect effect"

We totally agree. We have reworded this statement to 'These results have further confirmed and reassured our understanding of the aerosol first indirect effect'

Line 271: It is interesting that the average Nc for ACE-ENA winter is greater than both the sub-cloud Nacc and Nccn. Do you have any comments on why this is the case? Has this been observed elsewhere? Are the values of Nc influenced by above-cloud aerosols or sub-cloud Na?

We have added a brief discussion on this:

'Note that the $N_{CCN0.35\%}$ and $N_c$ values are lower than $N_c$ values during the ACE-ENA winter IOP, which is also confirmed in previous studies (J. Wang et al., 2022; Wang et al., 2023), which is also confirmed in previous studies (J. Wang et al., 2022; Wang et al., 2023). This interesting phenomenon can potentially be attributed to a combination of factors including lower MBL aerosol sources, stronger in-cloud coalescence-scavenging depletion of sub-cloud aerosols, and the aircraft snapshots capturing the equilibrium states of aerosols and cloud due to enhanced aerosol activations induced by stronger updrafts during the ACE-ENA winter (J. Wang et al., 2022). This thereby compels further investigation into the potential impacts of precipitation on the MBL CCN budget.'

Line 310: This is an excellent discussion of entrainment mixing and its competing influence on droplet size/liquid water content and likely highlights two different modes of cloud top mixing – homogeneous and inhomogeneous mixing that depend on the entrainment rate (Lehmann et al., 2009; Lu et al., 2011) – do the authors have any comments based on their calculated entrainment rates? The authors could cite examples of previous studies that show similar vertical profiles of cloud properties where the effects of entrainment mixing on cloud microphysical properties were evident.

We have added the discussion as follows:

'For the three IOPs, the $N_c$ and $LWC_c$ exhibited a stable trend from the cloud base, followed by a noticeable decrease near the cloud top mixing zone, while the changes in $r_c$ trend were not as dramatic as the others. Such characteristics of the cloud microphysics vertical profiles indicate the signal of inhomogeneous mixing, which occurs when dry and warm air mixes unevenly and not rapidly with the cloud air, hence partially evaporating the cloud droplets (Lehmann et al., 2009; Lu et al., 2011). The results are consistent with findings in stratocumulus clouds over multiple field campaigns (Brenguier et al., 2011; Jia et al., 2019) and with the findings in Sanchez et al. (2020) for five stratocumulus cases during the SOCRATES. However, further quantification of the entrainment-mixing mechanisms requires high-frequency eddy dissipation and accurate examination of the mixing time scale (Lehmann et al., 2009; Lu et al., 2011), which is of interest for future study.'

Line 370: The skewness of a distribution can actually be calculated as a statistical parameter rather than having a visual comparison. I leave it to the authors to decide if they would like to add this parameter to the study.

We have calculated the skewness values for the cloud DSDs and include them in the discussion: 'For the four cloud portions from cloud base to cloud top, the skewness of summertime (wintertime) cloud DSDs are 0.627 (0.271), 0.358 (0.175), 0.098 (-0.063), and -0.362 (-0.554), respectively.'

SECTION 4:

Line 434: This statement would be more accurate if the Liquid Water Path (LWP; vertical integral of the LWC) values were compared across campaigns rather than the mean LWC. Suggest adding LWP values or rewording the sentence.

We have changed this statement to:

'Furthermore, the similarity in the vertical integral of $LWC_c$ (as shown in Fig. 3c) provides comparable liquid water between three IOPs'

Line 468: Do you want to mention some of these aircraft campaigns - VOCALS, ORACLES, ACTIVATE, etc.?

We have expanded the discussion here:

'However, a more comprehensive investigation into the cloud microphysical responses to CCN intrusions under a larger range of various water supply conditions, and further untangling the ACI from the meteorological influences, will require additional aircraft cases from more field campaigns, for instance the VAMOS Ocean-Cloud-Atmosphere-Land Study (VOCALS), the Cloud System Evolution over the Trades (CSET), the ObseRvations of CLouds above Aerosols and their intEractionS (ORACLES), and the Aerosol Cloud meTeorology Interactions oVer the western ATlantic Experiment (ACTIVATE).'

Line 475: These are some very interesting results. While the ACI$_{r,CT}$ values are close to what one might expect (droplets are too large near cloud top for above-cloud aerosols to exert a significant influence on r near cloud top), it is interesting to note the ACI$_{N, CT}$ values reported here. Do the authors have any explanation or hypotheses for what causes these values to not be closer to 0 like ACI$_{r, CT}$?

We have added a brief explanation to this:

'Compared to the $ACI_{N,CB}$ and $ACI_{r,CB}$, the $ACI_{N,CT}$ and $ACI_{r,CT}$ are much weaker, especially for $ACI_{r,CT}$, as the near cloud top droplets are too large for above-cloud aerosols to exert a significant influence on $r_c$ (Diamond et al., 2018; Gupta et al., 2022). While the weaker cloud top $N_c$ dependence on the $N_{CCN,0.35\%}$ could be due to the legacy of the sub-cloud CCN impacts on $N_c$ being conveyed to the cloud top. This occurs because FT aerosols and CCN can be entrained down to the MBL before and during the cloud process, as observed in the assessment of inter-cloud cases. These weaker relationships support the notion that though the aerosols entrained into the upper-cloud region can affect the cloud microphysics to a certain degree, the effects are less pronounced than those from the sub-cloud aerosols (Diamond et al., 2018, Wang et al., 2020) because the MBL cloud $N_c$ and

$r_c$ variations are dominated by the condensational growth process, collision-coalescence process, and cloud top entrainment mixing near the cloud top.'

Line 480: I think the authors should also mention cloud top entrainment mixing over here.

We have changed this statement to:

'…the MBL cloud $N_c$ and $r_c$ variations are dominated by the condensational growth process, collision-coalescence process, and cloud top entrainment mixing near the cloud top.'

Line 484: As mentioned earlier, would be good to also cite Feingold and Seibert, 2009 when defining $S_o$, which to my knowledge was the first study to define the term.

This reference is added.

Line 491: The authors should provide the correlation coefficient values for the $S_o$ calculations and contrast these with previous studies. At least two recent studies did this – Jung et al. 2016 and Gupta et al. 2022.

Line 501: "…due to decreasing $So$ within the thicker cloud (Terai et al., 2012)". This is oversimplifying the problem. The value of $S_o$ depends not only on cloud thickness but also on the calculation methodology as shown by Terai et al, the cloud type – cumulus versus stratocumulus (Sorooshian et al., 2009; Jung et al., 2016) and the above- and below-cloud aerosol concentration (Duong et al., 2011; Jung et al., 2016; Gupta et al., 2022). Having more information on cloud type/morphology in the introduction would give context to these $S_o$ values and other results in the study.

Response to comments on L491 & L501:

We have added the suggested reference in the introduction, and the following discussion in the Section 4.2 (precipitation susceptibility):

'The $S_0$ values are 0.979, 1.229, and 1.638, with the absolute values of correlation coefficients being 0.33, 0.29, and 0.45 for SOCRATES, ACE-ENA summer and winter, respectively. These correlation coefficient values fall within the reasonable range found in previous studies on precipitation susceptibility in MBL stratus and stratocumulus clouds (Jung et al., 2016; Gupta et al., 2022), and indicate statistically significant dependences of $R_{CB}$ on $N_c$. Previous study by Terai et al. (2012) found that the $S_o$ values decrease with the increasing cloud thickness over the southeast Pacific, and Jung et al. (2016) found that the $S_o$ is more pronounced within the medium-deep clouds with thickness ~300-400 m in the MBL stratocumulus over the eastern Pacific. While Gupta et al. (2022) found that the $S_o$ values are generally higher under low ambient $N_a$ condition in the southeastern Atlantic MBL. In this study, $R_{CB}$ for the ACE-ENA winter is more susceptible to the layer-mean $N_c$ than the ACE-ENA summer and SOCRATES, which can be partially attributed to the existence of more large drizzle drops (as shown in Fig. 4d) near the cloud base. As previously discussed, the ACE-ENA winter feature with enhanced collision-coalescence and the drizzle-recirculating processes, especially under low $N_c$ conditions with more larger drizzle drops, leading to the increase of $S_o$ values. In comparison, the higher ambient aerosol and CCN concentrations during SOCRATES lead to relatively narrower drizzle DSDs and may induce effective aerosol buffering effects, where the warm-rain processes in cloud are already fairly suppressed, hence diminishing the sensitivity of $R_{CB}$ to $N_c$ (Stevens and Feingold, 2009; Fan et al., 2020; Gupta et al., 2022).'

Line 531: What are the units of the CCN loss rate? Here, the values are reported with units of "cm$^{-3}$" which does not include a unit of time, this is likely an error?

We have corrected the unit to cm$^{-3}$h$^{-1}$, thanks for catching this.

SECTION 5:

Line 568: The differences can also be attributed to the different size distributions which are then due to the sources discussed in the following sentences. That would then nicely lead to the discussion of aerosol modes toward the end of the paragraph.

Line 569: I don't think using the words 'pristine natural environment' is appropriate when the previous sentence claimed the aerosol concentrations are highest for SOCRATES.

Response to comments on L568 & L569:

We have reworded the following discussion as:

'The differences can be attributed to the differences in aerosol size distributions between ACE-ENA and SOCRATES, which are largely due to the aerosol sources in those regions. The SOCRATES features the pre-industrial natural environment enriched by aerosols from marine biological productivity and without the contamination of anthropogenic aerosols…'

Line 580: Can you also list the percentage increase in r from cloud base to top since these campaigns had different cloud thickness values?

We have changed this statement to:

'…the $r_c$ growths (and percentage increases), from cloud base to top, being 4.03 µm (0.66%), 4.78 µm (0.68%), and 5.85 µm (0.79%) for SOCRATES, ACE-ENA summer, and winter, respectively.'

Lines 255 and 576: How does in-cloud coalescence cause an increase in the size of sub-cloud aerosols? In-cloud coalescence would increase the size of a cloud drop as it accumulates water by colliding and coalescing with other droplets. Once this drop evaporates in the sub-cloud to expose the residual aerosol, the aerosol/CCN core size should be the same unless the CCN is modified during droplet growth. Is this related to the condensation of sulfuric gas onto aerosol cores as described in Line 245? If so, is there a way to verify this based on these observations? If not, this should be stated as a hypothesis rather than a conclusion?

We have added the discussion as follows:

'Coalescence scavenging refers to the process in which cloud or drizzle droplets, containing aerosol particles, merge with each other. Upon the collision-coalescence of

cloud droplets, the dissolved aerosol masses within the cloud droplets also collide and merge into a larger aerosol core, leading to larger aerosol particles upon droplet evaporation. The sub-cloud aerosols are then replenished into the cloud layer, experiencing growth within the cloud through cloud and drizzle droplet collision-coalescence, and subsequently falling and evaporating outside the cloud again. Eventually, the residual aerosols undergoing this cloud-processing cycle will gradually decrease in number concentration and increase in size (Flossmann et al., 1985; Feingold et al., 1996; Hudson and Noble, 2020; Hoffmann and Feingold, 2023).'

Lines 591-597: I don't fully understand or agree with the conclusion drawn here. The studies that the authors cited here/earlier (among many others) have calculated $f_{ad}$ using in-situ aircraft data from other locations and shown that assuming $f_{ad} = 0.8$ could lead to errors in satellite estimates of droplet concentration. While the calculation of $f_{ad}$ and stating the regional values is important, these $f_{ad}$ values do not "shed light on the further understanding of the satellite retrievals, particularly the satellite-based aerosol-cloud interaction assessment". The authors can state the $f_{ad}$ values and perhaps add a comment on the need to use these values when calculating droplet concentration for these regions using satellite retrievals, but I suggest removing lines 594-597.

We have eased our tone and modified our discussion to:
'While satellite retrievals of droplet number concentration heavily rely on the adiabatic cloud assumption and are usually given as a constant of $f_{ad} = 0.8$, the in-situ observational evidence found in this study further confirms the unrealistic nature of this assumption. It will be of interest to utilize multiple aircraft measurements (campaigns) to explore the variability of MBL cloud and drizzle microphysical properties over different marine regions. This can help examine potential predictors for $f_{ad}$, which will aid in satellite-based retrievals and aerosol-cloud interaction assessments (Painemal and Zuidema, 2011; Grosvenor et al., 2018; Painemal et al., 2021).'

Line 614: I don't understand what is meant by "the aircraft assessment provides more connected circumstances between the aerosols and cloud layer."

We have changed this statement to:

'…the aircraft assessment of ACI is based on measurements where the aerosols are in direct contact with the cloud layer.'

**TECHNICAL CORRECTIONS/CLARIFICATIONS:**

Line 12: Could use the term "aerosol-cloud-precipitation" given the terminology in the title? Also, change to "interactions" given the verb "are" in the next sentence?

We have changed this sentence as suggested.

Line 94: "aerosol, cloud, and drizzle"?

We have changed this statement to:

'This study targets the similarities and differences in the MBL aerosol, cloud, drizzle properties, their distribution and evolution, and more appealingly, the ACIs and ACPIs between the two campaigns.'

Line 105: "onboard the aircraft"?

We have changed this sentence as suggested.

Line 110: "large, ice particles"? Large ice particles would be a lot larger than 200 um.

We have changed this statement to:

'The 2DS in-situ measurements will be used as additional screening to eliminate the ice particles with diameters larger than 200 μm.'

Line 286: "To ensure the representativeness of the vertical profiles"?

We have changed this sentence as suggested.

Line 443: The sentence should probably end with "during the winter".

We have changed this sentence as suggested:

'…indicating that $N_c$ is more sensitive to the sub-cloud $N_{CCN,0.35\%}$ during the winter.'

Line 582: Do you mean "The mean cloud-top entrainment rates ($we$ ) are a function of cloud top virtual potential temperature and vertical velocity and their values are…."

We have changed this sentence to:

'Given the valid cloud top virtual potential temperature and vertical velocity measurements for the selected cloud cases, the averaged $w_e$ values are 0.570±0.834 cm s$^{-1}$, 0.581±0.560 cm s$^{-1}$, and 0.960±1.127 cm s$^{-1}$ for SOCRATES, ACE-ENA summer and winter, respectively.'

Figure 1: Please mention which statistical metrics are provided in the legends. Suggest adding that to the figure caption.

We have added the following to the caption:

'The statistical metrics in the legends denote the mean and standard deviation values for all samples in three IOPs.'

Figure 2: The caption lists the incorrect size range for the inner plots. Should be "Aitken mode size distribution ($Dp$ = 0.01 to 0.06 µm)"

The caption is corrected, thanks for catching.

---

## Author Comment (AC2)

**Response to Reviewer #2**

We appreciate your time and effort in thoroughly reviewing our manuscript. We are truly grateful for your constructive comments and insightful suggestions, which encourage and help us to improve the manuscript. We have revised the manuscript carefully based on your comments. In the response below, your comments are provided in black text and our responses are provided in blue text.

**Response:**

**SUMMARY**

The authors present a comparative study of aircraft in situ aerosol and cloud microphysical measurements of liquid phase boundary layer (BL) clouds from two different regions: over the Eastern North Atlantic (ENA) region near the Azores and the Southern Ocean (SO) in an area spanning south of Tasmania. The ENA measurements are from the summer and winter seasons, while the SO measurements are from summertime only. The overall conclusions are that:

1. Clouds across different regions and seasons have differing collections of microphysical properties
2. These differing microphysical regimes exhibit different susceptibilities to aerosol perturbations
3. Drizzle has a big impact on the BL CCN budget
4. Turbulence plays a leading role in enhanced precipitation seen in the ENA winter regime

The primary data analyzed in the study are aerosol and cloud microphysical properties averaged over all full cloud soundings of warm BL clouds during each campaign. These properties include number concentration (total, modal, fully size-resolved, etc.), measures of drop size distribution (DSD) width, effective radius/mean diameter, liquid water content, sedimentation rate, etc.

Many (if not most) of these observations have been analyzed in other recent studies, and indeed many of the conclusions reached by the authors are "consistent with" (or similar language) the results of these other papers. The frequency with which conclusions are followed by such qualifiers gives the impression that there is not much new added by the manuscript. It would be more effective to give a condensed overview of in situ work on the ACE-ENA and SOCRATES campaigns in which you lay out what has already been done. Then in the results section, devote a subsection (or a couple paragraphs, whatever) to explain how your work fits with what's already been published. This would be easier to digest than the piecemeal and repetitive referencing in the

manuscript's current state. I do see novelty in the specific focus on interactions among aerosol, clouds and precipitation (abbreviated ACI or ACPI, depending on the context), which to my knowledge have not been addressed in the literature for either field campaign (ACE-ENA and SOCRATES) analyzed.

I have serious concerns about the lack of justification for combining analysis of the field campaigns used here as well as the scientific reasoning leading to the point that turbulence- enhanced collision-coalescence explains more intense drizzle during one campaign (ACE-ENA winter) versus the other two. In fact, I think the differences in precipitation attributed to turbulence can be explained much more simply without appealing to turbulence-microphysics interactions *at all* (see $3^{rd}$ major comment below). While there is clearly a body of legitimate analysis presented in the study, I recommend the manuscript either undergo VERY major revisions or that it be rejected so the authors have sufficient time to rewrite the manuscript before resubmitting it.

Thanks for the thorough assessment of our manuscript, we have tried our best to carefully consider and address your comments and suggestions.

We have reconstructed our discussion to review the previous works on ACE-ENA and SOCRATES in the introduction, and our results in comparison with the previous study in the summary and conclusion section. Please refer to the response to general comment 1 and the revised manuscript.

**GENERAL COMMENTS**

1. **The unifying thread tying together measurements from the two regions is not clear to me, and I ask that the authors further emphasize/clarify the scientific motivation for combining the campaigns**. This would give a stronger basis for communicating the significance of your results. As it stands, there are no previously unexplored commonalities across the 3 campaigns. Yes, cloud effective radius increases with height for all the campaigns; yes, drizzle mean diameter increases from cloud top to base – these results (among others) are expected, and simply demonstrate that atmospheric physics as we know it isn't completely "broken." But beyond that, what purpose does this comparison serve? The aerosol and meteorology driving the clouds in each regime are quite different, so it's somewhat of a trivial conclusion that the microphysical properties differ as well.

The argument that "SOCRATES and ACE-ENA both took place in the midlatitudes, so they're directly comparable" is insufficient. The SO region sampled by SOCRATES is more consistently impacted by midlatitude cyclone systems than ENA during summer, which is more often dominated by the nearby Azores high (i.e., ENA is borderline subtropical during summer). In addition, I do not buy that these aircraft campaigns can be taken as "representative" samples of their respective latitude bands/ocean basins; Mechem et al. (2018) show significant interannual variability in the synoptic conditions experienced at the ENA site, and the summer ACE-ENA IOP was characterized by anomalously low BL heights *and* substantial BL decoupling.

To demonstrate the feasibility and justification of combining these two campaigns, we calculated the composite of the selected case periods during the ACE-ENA and SOCRATES using the hourly ERA5 reanalysis, as shown in Figure S1 below. The composite 850 hPa geopotential height is denoted by black contours, and the height anomaly from the 20-year climatology (2000-2020) is portrayed by the shaded area.

The discussion in the introduction has been modified to better motivate our study, as follows:

[revised manuscript text omitted]

**2.** The authors go to some length to justify their assertion that turbulence-enhanced collision-coalescence is the reason for stronger precipitation during winter at ENA, but the evidence given does not prove their hypothesis. **The discussion of TKE is illustrative but quantitatively insufficient.** For one, very few details are given on how the velocity perturbations are calculated; for example, what is the integral length scale obtained with a 10 s moving window (i.e., are you capturing the inertial subrange)? Is it the same for ACE-ENA and SOCRATES? (no) Is any window function applied or is this a simple "boxcar" moving average? Is 1 Hz data used or did you analyze high-rate data? Is "high-rate" the same for ACE-ENA and SOCRATES? (no) In addition, TKE is not the relevant quantity for evaluating turbulent enhancement of collisional growth; rather, it is the TKE dissipation rate $\epsilon$ that is used in parameterizations (e.g., Grabowski and Wang 2013 as referenced in the manuscript). For another, $\epsilon \sim O(10^{-4} m^2/s^2)$ in stratiform BL clouds while in shallow cumulus it is about an order of magnitude higher. Based on the sampling goals of both ACE-ENA and SOCRATES, mostly stratiform clouds were sampled, suggesting generally low turbulence intensities. A past modeling study on the feasibility of turbulence to overcome the "warm rain bottleneck" and accelerate drizzle formation via collision- coalescence enhancement in subtropical marine stratocumulus (Sc) showed a minor impact (Witte et al. 2019, doi: doi: 10.1175/MWR-D-18-0242.1) – why is a different answer expected in the same cloud dynamical regime? Finally, cold pools are a dynamical forcing mechanism more prevalent during ACE-ENA winter than either of the other 2 campaigns; this is not mentioned at all.

For you to continue pushing the line of reasoning that turbulence directly *causes* stronger precipitation, at very minimum you need to demonstrate that $\epsilon$ is substantially stronger for the ACE-ENA winter campaign than both typical marine Sc *and* SOCRATES/ACE-ENA summer (i.e., much greater than the 30-40% difference in mean cloud-top TKE shown). If you are unable to demonstrate this, I cannot support the heavy reliance on turbulence-enhanced collision-coalescence peppered throughout the manuscript. If you move forward with quantifying, please explicitly detail your approach in the methodology section – I recommend Siebert et al. (2010, 10.1175/2009JAS3200.1) or Waclawczyk et al. (2017, doi: 10.5194/amt-10-4573-2017) as starting points for developing your own analysis.

Thanks for the great suggestions. After carefully considering and examining this aspect, we found that the 1Hz used in this study is not sufficient to capture the small-scale turbulent structures. Thus, we have added the method description, and the following discussion of limitations:

'The turbulent perturbations of vertical ($\overline{w'^2}$) and horizontal ($\overline{u'^2}$ and $\overline{v'^2}$) components are calculated as the simple moving variance in a 10s window centered at the measurement time, without window weighting function, using 1Hz data for all three IOPs. The $w$ data is confined to an absolute aircraft roll angle of less than 5° (Cooper et al., 2016). Given the average aircraft ground speed of ~140 m/s and vertical speed of ~5 m/s (Altas et al., 2020), the smallest resolved wavelength is 140 m. Hence, within the 10s moving window, the ~50 m in the integral vertical range is able to resolve the eddies smaller than 140 m, and preserve the potential of capturing the inertial subrange.

The use of TKE provides an illustration that in-cloud turbulence during ACE-ENA might be slightly stronger than that observed during SOCRATES. That being said, the quantitative evaluation of the turbulent enhancement of collision-coalescence requires access to the eddy dissipation rate, as typically used in model parameterizations (Grabowski and Wang, 2013; Wittle et al., 2019). The smallest scales resolvable with the 1Hz measurement used in this study are on the order of 140 meters, thus capturing only the larger-scale end of the inertial subrange and larger turbulent motions. Consequently, the ability to resolve smaller eddies and turbulent structures, crucial for understanding the energy cascade within the inertial subrange, is limited by the too-coarse spatial and temporal resolutions and aliasing issues (Siebert et al., 2010; Muñoz-Esparza et al., 2018; Kim et al., 2022). Therefore, to fully resolve the spectrum of turbulence and quantitatively examine energy dissipation and mixing processes, access to higher-frequency measurements is required to capture smaller eddies within the inertial subrange (Siebert et al., 2010; Lu et al., 2011; Waclawczyk et al., 2017). Additionally, the further quantification of the entrainment-mixing mechanisms also requires high-frequency eddy dissipation and accurate examination of the mixing time scale (Lehmann et al., 2009; Lu et al., 2011) for individual profile. Though currently beyond the scope of this study, those mechanisms will be of interest for future investigations.'

I will note that your point that the activation fraction of available CCN to cloud droplets is highly correlated with turbulence intensity (as stated in the abstract, albeit differently worded) is valid, but this is not quantitatively demonstrated either; you could adopt a framework as in Hu et al. (2021, doi: 10.1029/2021JD035180) to explore this point further.

Following the notion in Hu et al. (2021), we have calculated the correlation coefficients between the CCN activation ratio and the TKE, as an inter-cloud assessment. The following discussions are added to Section 4.1:

'Previous studies have shown that the enhanced vertical turbulence (updraft velocity) can effectively facilitate CCN replenishment into the cloud layer (Hu et al., 2021; Zheng et al., 2022a&b) and increase the actual in-cloud supersaturation (Brunke et al., 2022), thus leading to a more efficient cloud droplet formation, enhancing the $ACI_{N,CB}$. By correlating the mean TKE values with the CCN activation ratio ($N_c/N_{CCN,0.35\%}$) for all individual cloud cases, the three IOPs show moderate but statistically significant correlation coefficients of 0.36, 0.55, and 0.51 for ACE-ENA summer, winter, and SOCRATES, respectively. This result reinforces the notion that the CCN activation fractions, particularly during the wintertime ACE-ENA, are significantly correlated with in-cloud turbulence intensities.'

And in the summary section:
'The moderate but statistically significant correlation coefficients between the CCN activation fractions and the TKE agree with a previous study that found the local activation fraction of CCN to be strongly associated with increased updrafts (Hu et al., 2021).'

3. Beyond the lack of quantitative evidence supporting the hypothesis that turbulence is the cause of increased drizzle production during ACE-ENA winter, **there is a simpler, more parsimonious explanation that I believe is given short shrift in the manuscript: that a combination of a deep cloud layer along with relatively clean aerosol conditions during ACE-ENA winter results in robust drizzle generation**. You do discuss the relationship between cloud depth and precipitation susceptibility very briefly in section 4.2, but it essentially reads as a footnote versus a primary point. Given the strong dependence of cloud base rain rate on cloud depth in the empirical relation discussed in this same section (4.2), it was a major oversight that you didn't explore this aspect further.

We agree with your argument and have revised and included the following discussion on the end of Section 3.3:

'Drizzle formation and evolution in the ACE-ENA winter clouds are noticeably stronger than in the other two IOPs, which could be attributed to multiple factors. First, the ambient aerosols and CCN during winter are substantially fewer, featuring clean environments that promote the formation of generally larger cloud droplets due to the availability of more water content per droplet. Larger cloud droplets are more likely to collide and coalesce into drizzle drops, leading to relatively heavier precipitation (Chen et al., 2011; Duong et al., 2011; Mann et al., 2014). Furthermore, the wintertime clouds feature deeper cloud layers with mean thickness of (392.4 m) compared to the summertime clouds (336.3). In a thicker cloud layer with sufficient turbulence, the residence times of large cloud droplets and drizzle drops are elongated, and the chance of collision-coalescence growth can be effectively increased by recirculating the drizzle drops (Brost et al., 1982; Feingold et al., 1996; Magaritz et al., 2009; Ghate et al., 2021). Additionally, the prevalence of precipitation-evaporation-induced MBL cold pools, which disturb the MBL thermodynamics and contribute to turbulent mixing (Zuidema et al., 2017), during the wintertime might provide strong dynamical forcing to the warm-rain process (Jenson et al., 2021; J. Wang et al., 2022). As a result, the ACE-ENA wintertime drizzle DSD is sufficiently broadened, and the $D_{mmd}$ is enlarged toward the cloud base. In comparison, although the SOCRATES exhibits even thicker clouds (487.4 m), the drizzle processes are seemingly suppressed by the much higher ambient aerosol and CCN concentrations.'

4. The discussion of the role of thermodynamic decoupling is incomplete, but decoupling ostensibly plays a significant role in the low values of fad encountered during all three campaigns. It would be well worth taking the next step and directly quantifying at least one decoupling metric from the observed profiles as defined by Jones et al. (2011).

We adapted the notion of Jones et al. (2011) and calculate the coupled layer and the degree of decoupling for every individual profile, as described in the following discussion in last paragraph of Section 2.1:

'Jones et al. (2011) suggested that the MBL would be in a well-mixed and coupled condition when the difference in liquid water potential temperature ($\theta_L$) and total water mixing ratio ($q_t$) between the bottom of MBL and the inversion layer are less than 0.5 K and 0.5 g/kg, respectively. In this regard, since the coupled and decoupled MBL conditions coexist in the selected cloud cases in this

study, particularly in ACE-ENA summer, which is characterized by anomalously low BL heights and substantial BL decoupling. Previous studies found that, under the decoupling condition, the aerosols, CCN, and moisture sources near the surface are disconnected from the cloud layer aloft, hence exerting much less effective impact on the cloud microphysics (Zheng et al., 2022a; Christensen et al., 2023). Therefore, we adapt and modify the metric in Jones et al. (2011) to calculate the sub-cloud coupled layer, in order to ensure the aerosols and CCN measured sub-cloud are in a well-mixed state and can represent the actual interaction (or contact) with the cloud layer. In this study, the $q_t$ and $\theta_L$ at the cloud base are calculated, and then their vertical variations are examined starting from the altitude of cloud base ($z_b$) and looking downward. As such, the coupled point altitude ($z_{cp}$) is defined as the altitude where the vertical changes in $q_t$ and $\theta_L$ exceed 0.5 K and 0.5 g/kg, respectively. Hence, the coupled layer ($H_{cp} = z_t - z_{cp}$) is defined as the layer between the cloud top altitude ($z_t$) and coupled point altitude ($z_{cp}$), hence the selection of the aerosols and CCN within the below-cloud part of the coupled layer can be viewed as in contact with the cloud. An example of the coupled layer identification is shown in Figure S2. Therefore, the degree of MBL decoupling ($D_{cp}$) can be quantified as the ratio of the coupled sub-cloud MBL thickness to the sub-cloud MBL thickness, where $D_{cp} = 1 - (H_{cp} - H_c)/z_b$. As shown in Table S1, the ACE-ENA summer feature with highest degree of decoupling (averaged $D_{cp}$=0.504), compared to the ACE-ENA winter ($D_{cp}$=0.370) and SOCRATES ($D_{cp}$=0.277).'

5. I am not familiar with "retrospecting" as discussed in section 4.3 and shown in Fig. 8. What is the procedure for performing this analysis? Please explain in the manuscript, as this appears to be simultaneously one of the most tentative aspects of the paper as well as something the authors are rather excited about.

We have added the following discussion on the methodology:

'In order to examine the potential impact of the aforementioned processes on the $ACI$ assessment, a sensitivity analysis is conducted by simply retrospecting the sub-cloud $N_{CCN0.35\%}$ according to their $L_{CCN}$. For each retrospective time step $\Delta T$, the $r_c$ values are held unchanged, and the retrospective $N_{CCN0.35\%}$ values for individual cloud cases are given by $N_{CCN0.35\%} - L_{CCN} * \Delta T$, and then the $ACI_{r,CB}$ can be recalculated. Note that assuming a constant $r_c$ value over time inevitably induces uncertainty and biases, as it does not consider the microphysical processes affecting the cloud

droplet mean size. However, previous numerical experiments show that the noticeable impact on the cloud mean radius through collision-coalescence necessitates a high degree of CCN depletion, and the quantified percentage changes in droplet mean sizes are several times less than the changes in CCN depletion (Feingold et al., 1996). Hence, the retrospective method, from an observational snapshot point of view, provides a direction that enables the assessment of $ACI_{r,CB}$ as if before the sub-cloud aerosols and CCN are scavenged by in-cloud coalescence-scavenging and precipitation scavenging processes.'

**SPECIFIC COMMENTS**

Each comment refers to a specific line/passage, figure, table or caption. Specific line(s) are denoted by LXX (or LXX-YY for longer passages).

L24-26: the lack of sensitivity of precipitation to aerosol during SOCRATES suggests that it inhabits a different microphysical regime than ACE-ENA. In other words, there are sufficient CCN during SOCRATES that the ACPI are effectively "saturated" with respect to increasing aerosol loading. There are numerous references discussing such buffering effects (a good starting point is Stevens and Feingold 2009, doi: 10.1038/nature08281) that I recommend the authors consult to reframe this discussion.

We have added the following discussion in Section 4.2 when discussing the precipitation susceptibility:

'…the ACE-ENA winter feature with enhanced collision-coalescence and the drizzle-recirculating processes, especially under low $N_c$ conditions with more larger drizzle drops, leading to the increasing the $S_o$ values. In comparison, the higher ambient aerosol and CCN concentrations during SOCRATES lead to relatively narrower drizzle DSDs and may induce effective aerosol buffering effects, where the warm-rain processes in cloud are already fairly suppressed, hence diminishing the sensitivity of $R_{CB}$ to $N_c$ (Stevens and Feingold, 2009; Fan et al., 2020; Gupta et al., 2022).'

L154-157: with respect to what are the "changes" in $\theta_L$ and $qt$ evaluated? The mean of the cloud layer or surface layer? Based on Fig. S1, I assume cloud layer. I am confused by this definition because "mixed layer" typically implies surface mixed layer, while you are using it to describe an

elevated mixed layer. In general, I found your analysis of decoupling to be incomplete.

We have refined our discussion, please refer to our previous response on the MBL decoupling discussion.

L191-195: When you discuss "airmass origin," at what vertical level(s) are back trajectories taken? I believe you in terms of PBL airmass, but is this also true above the PBL?

We have expanded the statements on the airmass origin as follows:

'In the SOCRATES region, according to the previous studies involving back-trajectory analyses, dominant air masses within the MBL primarily originate from the south or from the west, skirting the Antarctic coast (Zhang et al., 2023), while the air masses above the MBL follow a similar transport pathway, they can also originate from the tip of southern Africa and transport southeast along the warm conveyor belt (McCoy et al., 2021).

Conversely, the ENA region experiences aerosols of varied origins, spanning maritime air masses to those heavily influenced by continental emissions from North America or Northern Europe, especially during the summertime (Logan et al., 2014; Wang et al., 2020). The summertime air mass back-trajectories within the MBL strongly feature recirculating flow around the Azores high. During the wintertime, however, the air masses predominantly originate in the FT, are transported above the MBL, and are then further entrained down to the MBL by large-scale subsidence, indicating less influence from continental pollution (Y. Wang et al., 2021b).'

L192 and elsewhere: Zhang et al. (2023) reference is missing in bibliography – I assume the correct reference is from (most of) the same authors in Atmosphere (doi: 10.3390/atmos14081246)?

Yes, we have added the correct reference to the reference list. Thanks for catching.

L216-217: please add a location for the further discussion, i.e., "and will be further discussed in Section X.X" or "further discussed later in this section"

We have added the location for further discussion as: '…and will be further discussed in Section 3.1.'

L259: are there any measurements that support your assertion that it's both coalescence-enlargement and sea salt contributing to heightened concentration at Dp>1 um?

We have added the reference on previous case analysis during the summer ACE-ENA (Zheng et al., 2022b), and the long-term statistics on the coarse-mode aerosol seasonal variation over the ARM-ENA site (Zheng et al., 2018) to back up this statement.

'The elevation in sub-cloud coarse mode aerosols observed for both ACE-ENA IOPs (as seen in Fig. 2) can be attributed both to the coalescence-enlargement process and the intrusion of sea spray aerosols (e.g., sea salt). As illustrated and analyzed based on a case study during summertime that exhibits the signal of cloud-processing aerosols (Zheng et al., 2022b), as well as the long-term aerosol physicochemical properties over the ARM-ENA ground-based observatory (Zheng et al., 2018), particularly during the winter season where the production of sea spray aerosol is prevalent.'

L268-269: I don't think you've improved understanding of the first indirect effect. Rather, you're adding another data point that supports what we already understand about it. So it's more of a "confirmation" than anything novel.

We have eased the tone and changed this statement to: 'These results have further confirmed and reassured our understanding of the aerosol first indirect effect.'

L296: Verb tense disagreement. Recommend you make everything present tense: "…air **is** entrained into the boundary layer…"

We have changed this statement to present tense as 'The warmer and drier air near the cloud top entrains into the cloud layer and further mixes downward, often resulting in the evaporation of small cloud droplets and the shrinking of droplet sizes, which oppose condensational growth (Desai et al., 2021).'

L300-302: Does this difference in $r_c$ profiles tell you anything about mixing regime (i.e. homogeneous vs. inhomogeneous)?

We have added the following discussion on the entrainment mixing regime:

'For the three IOPs, the $N_c$ and $LWC_c$ exhibited a stable trend from the cloud base, followed by a noticeable decrease near the cloud top mixing zone, while the changes in $r_c$ trend were not as dramatic as the others. Such characteristics of the cloud microphysics vertical profiles indicate the signal of inhomogeneous mixing, which occurs when dry and warm air mixes unevenly and not rapidly with the cloud air, hence partially evaporating the cloud droplets (Lehmann et al., 2009; Lu et al., 2011). The results are consistent with findings in stratocumulus clouds over multiple field campaigns (Brenguier et al., 2011; Jia et al., 2019) and with the findings for selected cases during the ACE-ENA (Yeom et al., 2021), and the SOCRATES (Sanchez et al., 2020). While the near-cloud $r_c$ profiles for the ACE-ENA cases exhibit more constant variation, which could be possibly attributed to more effective mixing due to the stronger entrainment rate, particularly during the ACE-ENA winter, eventually reaching a smaller equilibrium in terms of mean sizes.'

L304: I can't imagine the water vapor source from entrainment evaporation is a leading order term in the BL $qt$ budget, and I'm having some difficulty understanding the relevance of raising this point. As you state, the net impact of entrainment on BL $qt$ should be negative (i.e., entrainment mixes in drier air, so BL-mean $qt$ should decrease), which would imply this evaporative source is a relatively minor offset to the entrainment drying sink. Eyeballing it from Fig. 3c, it looks like there's maybe 0.2 g/m3 of vapor that is liberated from the clouds (extrapolating the midcloud $ql$ lapse rate to $zi$=1) – but I can't assess this any further since you don't show any mean $qt$ or $qv$ profiles. Both the G1 and the GV have open path hygrometers from which $qv$ can be accurately measured in cloud – if you want to get into a discussion of the vapor budget, it would be helpful to explicitly show some of these measurements.

We are not intended to suggest that the evaporation will cause a difference in an order of magnitude. We appreciate you consideration on this argument, however, we have considered to delete this statement to avoid further confusion, and changed it to:

'As cloud-top entrainment mixing can shrink large cloud droplets via evaporation, depending on the entrainment mixing rate, the nearly constant $r_c$ values (at $z_i > 0.8$) might represent the equilibrium balance between two competing processes: cloud droplet condensational and collision-coalescence growths, and the entrainment mixing evaporation effects.'

L305-310: What evidence do you have for re-condensation beyond the inferences made from bulk profiles? And shouldn't there be *more rapid* growth on smaller drops since condensation rate is inversely proportional to surface area? You have the full DSDs to demonstrate the validity of the generalizations you're drawing. Please evidence for these assertions.

L313: What is gained by quantifying $\Delta r_C$? Is this not just a different way of expressing the subadiabatic $ql$ lapse rate via the relation $r_e = kr_v$ where $r_v \propto (q_l/N)^{1/3}$?

Response to L305-310 & L313:

We are trying to provide bulk descriptive discussion to introduce the discussion on the cloud microphysical responses on aerosols later, we have removed the argument on re-condensation and changed the discussion here as:

'When dry air entrainment occurs at the cloud top, some of the upper-level smaller cloud droplets will evaporate, which leads to decreases in $N_c$ (Fig. 3a). As cloud-top entrainment mixing can shrink large cloud droplets via evaporation, depending on the entrainment mixing rate, the nearly constant $r_c$ values (at $z_i > 0.8$) might represent the equilibrium balance between two competing processes: cloud droplet condensational and collision-coalescence growths, and the entrainment mixing evaporation effects.

The increases of $r_c$ ($\Delta r_c$) from cloud base to cloud top are 4.03 µm, 4.78 µm and 5.85 µm, with percentage increases of 66%, 68% and 79%, for SOCRATES, ACE-ENA summer and winter, respectively. Even though, theoretically, the condensational growth effect would be more pronounced on smaller cloud droplets due to their smaller surface area (Wallace and Hobbs, 2006), SOCRATES exhibits the thickest mean cloud thickness but experienced the least $r_c$ increases among the three IOPs. This suggests that high aerosol loadings are limiting the overall growth of the cloud DSD in SOCRATES clouds, while the ACE-ENA winter clouds show the strongest $r_c$ increase, in contrast. This comparison suggests different cloud microphysical responses to aerosol

perturbations in the three IOPs, which will be further discussed in Section 4.1.'

L337-340: Please define what terms are being used to calculate the "reduction of $LWC_C$" – it's not clear to me what you're doing here.

We are comparing the $LWC_c$ deficit between the three IOPs, from where it starts to decrease according to the mean profiles. We have added more description to the discussion:

'Considering the near cloud-top proportion of cloud where the $LWC_c$ experienced decrease, the difference in $LWC_c$ (between the cloud top value the upper-middle cloud maximum for the mean profiles) for the ACE-ENA summer (-0.032 g m$^{-3}$) is higher than the reductions in winter (-0.018 g m$^{-3}$) and SOCRATES (-0.009 g m$^{-3}$).'

L368-369: Water vapor competition matters in a water-vapor limited regime (which seems quite obvious when stated that way…), but it seems to me that ACE-ENA winter is more of an aerosol limited regime. Appealing to water vapor competition is not a "one size fits all" conclusion that can be universally applied.

We have added the following discussion in this regard:

'In addition, the discrepancies in ε between the three IOPs may be attributed to the sub-cloud aerosol differences, which essentially resided in different microphysical regimes. Y. Wang et al. (2021a) stated that higher aerosol loading would lead to increased ε due to the water vapor competition effect, supporting the discrepancy between SOCRATES and ACE-ENA summer IOPs, which can be categorized as a water-vapor-limited regime. Meanwhile, the ACE-ENA wintertime IOP exhibits characteristics of an aerosol-limited regime, in which the cloud DSDs tend to be narrower than in the water-limited regime, due to enhanced droplet growth, and the ε values further decrease with height via the condensational narrowing effect (J. Chen et al., 2018).'

L370: do you quantify skewness or is this a qualitative description?
We have calculated the skewness values for the cloud DSDs and include them in the discussion:
'For the four cloud portions from cloud base to cloud top, the skewness of summertime (wintertime) cloud DSDs are 0.627 (0.271), 0.358 (0.175), 0.098 (-0.063), and -0.362 (-0.554), respectively.'

L371-372: can you say with certainty whether coarse mode aerosols are drizzle residuals vs. "primary production" of sea salt from the surface? Seems like a difficult "chicken and egg" problem to assess from in situ data without either aerosol composition information or some modeling work to back up the statement.

Since both campaigns lack continuous coarse mode chemical compositions and the offline analyzed samples are inadequate for the select cases. We have added the following discussion regarding this issue:

'These coarse mode aerosols, whether from primary production of sea spray or from the residuals of evaporated drizzle drops, are more easily activated (or re-activated) into larger cloud droplets when they intrude (or recirculate) into the cloud layer (Hudson and Noble, 2020; Hoffmann and Feingold, 2023). Nevertheless, it is challenging to pinpoint the actual origins of coarse mode aerosols from the perspective of aircraft observational snapshots, thus requiring further numerical modeling work.'

L436: I assume you mean liquid water *content*, but this is not stated
We have changed this statement to:

'Furthermore, the similarity in the vertical integral of $LWC_c$ (as shown in Fig. 3c) provides comparable liquid water between three IOPs'

L490-491: What are the uncertainties in $S_0$? The correlations do not look very strong in Fig. 7a.

The $S_0$ values are 0.979, 1.229 and 1.638, with the absolute values of correlation coefficients are 0.33, 0.29 and 0.45, for SOCRATES, ACE-ENA summer and winter, respectively. These correlation coefficient values fall within the reasonable range found in previous studies on precipitation susceptibility in MBL stratus and stratocumulus clouds (Jung et al., 2016; Gupta et al., 2022), and indicate statistically significant (but not strong) dependences of $R_{CB}$ on $N_c$, since the $R_{CB}$ is not solely dependent on the $N_c$.

L514: Double check the equation, it looks like something is not formatted correctly or there are some extra characters.

We have corrected the equation:

$$R_{CB} = 1.73e^{-10} H_c^{3.6} N_a^{-1}$$

L536: Should there be a minus sign in the 2$^{nd}$ parenthetical?

Yes, we have corrected it.

L538-539: This sentence needs to be restructured, it is currently a fragment.

We have changed it to 'As the results indicate, the ACE-ENA clouds experience more substantial sub-cloud CCN loss than SOCRATES, especially in wintertime precipitating clouds.'

L542-543: you could expand upon this point more, it's a bit too concisely expressed to be easily understood.

We have further expanded on this discussion as:

'Recall that the assessment of $ACI_{r,CB}$ relies on the relative changes of $r_c$ and $N_{CCN}$, while the different $L_{CCN}$ for individual cases can result in the shrinking of the $N_{CCN}$ variation ranges (imagine the abundant CCN are depleted by the coalescence-scavenging). In other words, the given change in $r_c$ corresponds to a narrowed change in $N_{CCN}$. Mathematically speaking, the assessment of $ACI_{r,CB}$ depends on the ratio of the numerator (change in $r_c$) and the denominator (change in $N_{CCN}$). Under the circumstances of substantial cloud-processing to the aerosols, the altered sub-cloud CCN budgets are reflected as a smaller denominator, versus the less altered numerator, hence mathematically presented as an enlarged $ACI_{r,CB}$.'

L569: it is rather counterintuitive that the "pristine" environment has the strongest aerosol loading. This is paradoxical because we often use the terms "pristine" vs "polluted" to imply low vs. high aerosol loading, respectively. You clearly mean it in the sense that the SO region is minimally impacted by anthropogenic emissions. So a little word-smithing is needed to resolve this incongruity.

We have changed the term 'pristine' to 'pre-industrial' in order to describe the SOCRATES nature, as well as in the relevant discussion throughout the revised manuscript.

E.g., 'The SOCRATES features the pre-industrial natural environment enriched by aerosols from marine biological productivity and without the contamination of anthropogenic aerosols.'

L594: fully agreed that the assumption of constant $f_{ad}$ used in satellite $N$ retrievals is problematic, but how do the campaign-average profiles presented here improve this situation? On a profile-by-profile (or, from the satellite perspective, pixel-by-pixel) basis, is there anything from the measurements that suggest potential predictors of $f_{ad}$? Or do you view your contribution as simply another data point showing that an assumed value of 0.8 is unrealistic?

We have eased our tone and modified our discussion to:

'While satellite retrievals of droplet number concentration heavily rely on the adiabatic cloud assumption and are usually given as a constant of $f_{ad} = 0.8$, the in-situ observational evidence found in this study further confirms the unrealistic nature of this assumption. It will be of interest to utilize multiple aircraft measurements (campaigns) to explore the variability of MBL cloud and drizzle microphysical properties over different marine regions. This can help examine potential predictors for $f_{ad}$, which will aid in satellite-based retrievals and aerosol-cloud interaction assessments (Painemal and Zuidema, 2011; Grosvenor et al., 2018; Painemal et al., 2021).'

Figure 2: This figure could be compressed in the horizontal, which would accentuate the shape of the distribution in a manner pleasing to the eye. As it currently stands, this looks "stretched out" and there aren't many interesting detail/wiggles in any of the curves that merit such a long aspect ratio.

Figure 2 is replotted with smaller aspect ratio.

Figure 3: What do the shaded regions denote? Interquartile range? Standard deviation? 5th-95th percentiles? No info in caption.

The shaded region denotes the inter-cloud-case standard deviation. We have added this information to the captions.

Figure S2: put the two panels on the same plot so they can be directly compared.

Figure S2 is updated accordingly.

Figure S3: why does this size range needs to be separated from Fig. 2 in the main manuscript?

The coarse-mode range is added to the new Fig. 2, and the original Fig. S3 is removed.

Figure S4: please add uncertainty shading as you did in Fig. 3 of the manuscript, it would be helpful to see the variability of subadiabaticity within campaigns
The standard deviations are added as shading areas in Fig. S4.

Table S1: Please include f_ad in this table.

The $f_{ad}$ values are included.

Table S2: it looks like this table is cutoff. Are there more variables not shown?

We have fixed the table appearance.

---

## Referee Report (RR1)

**Review #2, Zheng et al., "Distinctive aerosol-cloud-precipitation interactions in marine boundary layer clouds from the ACE-ENA and SOCRATES aircraft field campaigns," Egusphere/ACP**

**SUMMARY**

Thanks to the authors for their responses to my and my fellow reviewer's comments. The reframing of the physical process discussion to de-emphasize turbulence as the primary contributor to increased collision-coalescence in the ACE-ENA winter regime was particularly appreciated. I still question the use of 1 Hz velocity data for quantifying turbulence but given the more qualitative discussion of the role of turbulence, this is sufficient. I have a few comments on additions to the manuscript and numerous typographical/language notes, but overall these should require only minor revisions.

On the latter topic of typographical comments, I understand the pressure to respond to reviews in a quick and timely fashion, but I encourage you to take a closer final editing pass before resubmitting future manuscripts. There are numerous errors in the added text that could easily have been avoided with one more close reading of the revised text. I can't speak for all reviewers, but this particular reviewer is always more agreeable/less cranky when a resubmitted paper is free (or at least, very close to free) of these minor and entirely avoidable errors, which are distracting and add significant time/effort to my review process.

**MINOR COMMENTS**

L148-149: re: "the enhanced large-scale subsidence would lead to a deeper stratocumulus-topped MBL" -- This is paradoxical - enhanced subsidence ostensibly compresses the boundary layer, so how does it lead to a deeper cloud-topped MBL? Maybe you're getting at the fact that enhanced subsidence also tends to sharpen the inversion?

L184-185: re: "[the SOCRATES region] is under more consistent influence of mid-latitude cyclone systems than over the ACE-ENA region" – I'd say the dominant impact on winter weather at ACE-ENA is very much mid-latitude systems.

L302: "the ratio of the coupled sub-cloud MBL thickness to the sub-cloud MBL thickness" – I think you mean "cloud thickness" instead of the 2nd "sub-cloud MBL thickness"?

L302: Why such a complicated expression for $D_{cp}$? Assuming $H_c = z_t - z_b$ (which you haven't stated in the text up to this point) and inserting your definition of $H_{cp}$, I come up with $D_{cp} = 1 - ((z_t - z_{cp}) - (z_t - z_b))/z_b = 1 - (z_b - z_{cp})/z_b = 1 - 1 - z_{cp}/z_b = z_{cp}/z_b$.

L743: There *are* high-rate measurements available from both SOCRATES (25 Hz; https://data.eol.ucar.edu/dataset/552.005) and ACE-ENA (20 Hz; https://adc.arm.gov/discovery/#/results/id::6747_aimms_sfcmet_met-air_airborne_horizwind?measurementsView=true&showDetails=true), so it's misleading

that you frame this as a lack of access. You don't need higher rate data than that to estimate turbulence properties, although aliasing is still an issue.

Table S2: Add units for all variables. I also recommend replacing "nan" with "—" but this is not a requirement.

**TYPOGRAPHICAL/LANGUAGE COMMENTS**

L145: "both summer and winter IOPs of ACE-ENA  featured  anomalously strong high-pressure" – remove "are," "with" and "er" from "stronger"; add "ly" to anomalous

L151: "while the winter IOP  prevalently featured  precipitation-generated…" – remove "is" and "with prevalent"; add "prevalently" before "featured"

L155: "In  recent years many observational studies based on  ACE-ENA data have…" – remove "Over the" and commas from this phrase; add "In" to start of sentence

L181-182: "anomalously strong" instead of "anomaly-stronger"

L190: what do you mean by "the functioning physical processes?" Your usage of "functioning" doesn't make sense to me. Do you mean the "dominant" or "first order" processes? Or are you instead aiming to compile a comprehensive list of every process operating within these clouds?

L272-274: Sentence starting "In this regard…" is a sentence fragment. Please restructure.

L274: "the decoupling conditions" ➜ "decoupled conditions"

L278: "in order to ensure the aerosols and CCN…" – I don't think "ensure" is the right word. I'd go for something like "to quantify the degree to which aerosols and CCN…"

L329: "which is described in the last section" – add "the"

L344: "subside " – the word "subside" implies downward motion; this is redundant

L345-346: "are in the reconciliation of getting the close-to-cloud…" – I'm not sure what you mean by "are in the reconciliation of"

L360: "and transport southeast" ➜ "and be transported southeast"

L364: new sentence starting with "While…" should be combined with previous sentence.

L365: "a thousand" – add units after number

L382: superscript "-3" – I didn't exhaustively catalog this issue, please double check unit superscripts throughout

L394: "double  the above-cloud…"

L477-479: remove duplicated phrase "which is also confirmed in…"

L586: "between the cloud top value and the upper-middle cloud…" – add "and"

L600: seems like "slowly" would be more appropriate than "not rapidly"

L604: what do you mean by "constant variation" – the nature of the variability is constant across cases? Please clarify.

L642: "which can be attributed…" – add "d" to end of "attribute"

L683: Correct reference "Altas et al., 2020" => "Atlas et al., 2020"

L698: Correct reference "Wittle et al., 2019" => "Witte et al., 2019"

L880: "featured  enhanced…" – remove "with"; add "d" to "feature"

L881: "more large" – remove "r" from "larger"

L896: $R_{cb}$ equation – still not happy with this. I would prefer you explicitly write out "1.73x10$^{-6}$" as it is not clear whether you are saying "natural base e to -10 power" or if you are referring to engineering notation, in which case the -10 would not be superscript and the "e" would be capitalized.

L900: remove "where is"

L1064: remove comma between "evolution" and "during"

---

## Author Response (AR2)

**Response to Reviewers**

In accordance with the color schemes policy of EGU publications, the original color scheme in the manuscript has been updated. The colors for ACE-ENA summer (originally red), ACE-ENA winter (originally blue), and SOCRATES (originally green) have been changed to pink, purple, and green, respectively. Only the colors have been altered; all the data remain original and intact. The updated figures were checked using the Coblis Color Blindness Simulator (https://www.color-blindness.com/coblis-color-blindness-simulator/) to ensure they allow readers with color vision deficiencies to interpret our findings correctly.

**Response to Reviewer #1**

Thanks for accepting our manuscript. In the response below, your comments are provided in black text and our responses are provided in blue text.

I have a single edit for the authors:

Line 386: There is repeating text: "which is also confirmed in previous studies (J. Wang et al., 2022; Wang et al., 2023)."

Thanks for catching it. The repeating text has been removed.

**Response to Reviewer #2**

We appreciate your time and effort in thoroughly reviewing our manuscript in the second round. We are truly grateful for your constructive comments and patient suggestions on the typographical issues. We have revised the manuscript based on your comments, and we have carefully gone through the manuscript for grammar errors and clarity. In the response below, your comments are provided in black text and our responses are provided in blue text.

**SUMMARY**

Thanks to the authors for their responses to my and my fellow reviewer's comments. The reframing of the physical process discussion to de-emphasize turbulence as the primary contributor to increased collision-coalescence in the ACE-ENA winter regime was particularly appreciated. I still question the use of 1 Hz velocity data for quantifying turbulence but given the more qualitative discussion of the role of turbulence, this is sufficient. I have a few comments on additions to the manuscript and numerous typographical/language notes, but overall these should require only minor revisions. On the latter topic of typographical comments, I understand the pressure to respond to reviews in a quick and timely fashion, but I encourage you to take a closer final editing pass before resubmitting future manuscripts. There are numerous errors in the added text that could easily have been avoided with one more close reading of the revised text. I can't speak for all reviewers, but this particular reviewer is always more agreeable/less cranky when a resubmitted paper is free (or at least, very close to free) of these minor and entirely avoidable errors, which are distracting and add significant time/effort to my review process.

Thanks for thoroughly reviewing our revised manuscript; please find our point-to-point response below.

**MINOR COMMENTS**

L148-149: re: "the enhanced large-scale subsidence would lead to a deeper stratocumulus-topped MBL" -- This is paradoxical - enhanced subsidence ostensibly compresses the boundary layer, so

how does it lead to a deeper cloud-topped MBL? Maybe you're getting at the fact that enhanced subsidence also tends to sharpen the inversion?

We have revised this statement to '…where the enhanced large-scale subsidence would lead to stronger and sharper temperature inversion above the stratocumulus-topped MBL…'

L184-185: re: "[the SOCRATES region] is under more consistent influence of mid-latitude cyclone systems than over the ACE-ENA region" – I'd say the dominant impact on winter weather at ACE-ENA is very much mid-latitude systems.

We have revised our statement of the SOCRATES and ACE-ENA winter IOP as follows:

'The region of selected SOCRATES cloud cases crosses a larger latitudinal zone and is under more consistent influence of mid-latitude cyclone systems than the ACE-ENA during the summer IOP.'

'The winter IOP was under the frequent impacts of the mid-latitude systems and prevalently featured precipitation-generated cold pools, where evaporative cooling alters the thermodynamical structure of the MBL, sustains and enhances turbulence mixing, hence contributes to dynamical perturbations that can influence the behavior of the MBL (Terai and Wood, 2013; Zuidema et al., 2017; Jenson et al., 2021; J. Wang et al., 2022; Smalley et al., 2024).'

**Reference:**

Smalley, M. A., Witte, M. K., Jeong, J.-H., and Chinita, M. J.: A climatology of cold pools distinct from background turbulence at the Eastern North Atlantic observations site, EGUsphere [preprint], https://doi.org/10.5194/egusphere-2024-1098, 2024.

L302: "the ratio of the coupled sub-cloud MBL thickness to the sub-cloud MBL thickness" – I think you mean "cloud thickness" instead of the 2nd "sub-cloud MBL thickness"?

L302: Why such a complicated expression for Dcp? Assuming $H_c = z_c - z_b$ (which you haven't stated in the text up to this point) and inserting your definition of $H_{cp}$, I come up with $D_{cp} = 1 - ((z_t - z_{cp}) - (z_c - z_b))/z_b = 1 - (z_b - z_{cp})/z_b = 1 - 1 - z_{cp}/z_b = z_{cp} - z_b$.

Answer to these two comments:

Thanks for catching it. We have simplified the expression of $D_{cp}$ to:

'Therefore, the degree of MBL decoupling $(D_{cp})$ can be quantified as the ratio of the coupled point height $(z_{cp})$ to the cloud base height $(z_b)$, where $D_{cp} = z_{cp}/z_b$.'

And we have added the description of cloud thickness in the previous paragraph:

'The detailed selected cloud profiles, with their cloud base heights $(z_t)$, cloud top heights $(z_b)$ and cloud thicknesses $(H_c = z_t - z_b)$ are listed in Table S1, along with the cloud profile macrophysics.'

L743: There are high-rate measurements available from both SOCRATES (25 Hz; https://data.eol.ucar.edu/dataset/552.005) and ACE-ENA (20 Hz; https://adc.arm.gov/discovery/#/results/id::6747_aimms_sfcmet_met-air_airborne_horizwind?measurementsView=true&showDetails=true), so it's misleading that you frame this as a lack of access. You don't need higher rate data than that to estimate turbulence properties, although aliasing is still an issue.

Thanks for the suggestion. We have revised the following statement to further clarify:

'Though currently beyond the scope of this study, utilizing the high-rate measurements of velocities available from SOCRATES (at 25Hz) and ACE-ENA (at 20Hz) to explore those mechanisms further will be of interest to future investigations.'

Table S2: Add units for all variables. I also recommend replacing "nan" with "—" but this is not a requirement.

The units are added in Tables S1 & S2, and all 'nan' are replaced by '−' in Table S2.

**TYPOGRAPHICAL/LANGUAGE COMMENT**

L145: "both summer and winter IOPs of ACE-ENA  featured  anomalously stronger high-pressure" – remove "are,""with" and "er" from "stronger"; add "ly" to anomalous

This sentence is revised to:

'…both summer and winter IOPs of ACE-ENA featured anomalously strong high-pressure systems…'

L151: "while the winter IOP is prevalently featured  precipitation-generated…" – remove "is" and "with prevalent"; add "prevalently" before "featured"

This sentence is revised to:

'The winter IOP was under the frequent impacts of the mid-latitude systems and prevalently featured precipitation-generated cold pools…'

L155: "In  recent years, many observational studies, based on  ACE-ENA data have…" – remove "Over the" and commas from this phrase; add "In" to start of sentence

This sentence is revised to:

'In recent years, many observational studies based on ACE-ENA data have focused on the seasonal contrasts of the aerosol distributions and sources…'

L181-182: "anomalously strong" instead of "anomaly-stronger"

Change is made accordingly.

L190: what do you mean by "the functioning physical processes?" Your usage of "functioning" doesn't make sense to me. Do you mean the "dominant" or "first order" processes? Or are you instead aiming to compile a comprehensive list of every process operating within these clouds?

We have changed it to '…the dominant physical processes…'

L272-274: Sentence starting "In this regard…" is a sentence fragment. Please restructure.

We have restructured this sentence to:

'The cases selected for this study feature both coupled and decoupled MBL conditions, particularly during ACE-ENA summer, which is characterized by anomalously low MBL heights and substantial MBL decoupling.'

L274: "the decoupling conditions" → "decoupled conditions"

Change is made accordingly.

L278: "in order to ensure the aerosols and CCN…" – I don't think "ensure" is the right word. I'd go for something like "to quantify the degree to which aerosols and CCN…"

We have revised this sentence to:

'Therefore, we adapt and modify the metric in Jones et al. (2011) to calculate the sub-cloud coupled layer, in order to quantify the degree to which aerosols and CCN measured sub-cloud are in a well-mixed state and can represent the actual interaction (or contact) with the cloud layer.'

L329: "which is described in the last section" – add "the"

The word 'the' is added.

L344: "subside down" – the word "subside" implies downward motion; this is redundant

The word 'down' is removed.

L345-346: "are in the reconciliation of getting the close-to-cloud…" – I'm not sure what you mean by "are in the reconciliation of"

We have revised this sentence to:

'Therefore, the 200 m criterion used in this study captures the close-to-cloud aerosol plumes and provides enough sample size for statistical analysis.'

L360: "and transport southeast" → "and be transported southeast"

Corrected.

L364: new sentence starting with "While…" should be combined with previous sentence.

L365: "a thousand" – add units after number

Answer to these two comments:

We have revised the sentence to 'For individual cases, the above cloud aerosols vary from a couple hundred to over a thousand particles per cubic centimeter'

L382: superscript "-3" – I didn't exhaustively catalog this issue, please double check unit superscripts throughout

We have corrected all the superscripts throughout the manuscript.

L394: "double  the above-cloud…"

Corrected.

L477-479: remove duplicated phrase "which is also confirmed in…"

The duplicated phrase is removed.

L586: "between the cloud top value and the upper-middle cloud…" – add "and"

Added.

L600: seems like "slowly" would be more appropriate than "not rapidly"

We have revised the sentence to 'which occurs when dry and warm air mixes unevenly and slowly with the cloud air, hence partially evaporating the cloud droplets', as suggested.

L604: what do you mean by "constant variation" – the nature of the variability is constant across cases? Please clarify.

This statement is changed to 'The near-cloud top $r_c$ profiles ($z_i > 0.8$) for the ACE-ENA cases exhibit fewer increases compared to the SOCRATES'

L642: "which can be attributed…" – add "d" to end of "attribute"

Added.

L683: Correct reference "Altas et al., 2020" => "Atlas et al., 2020"

Corrected.

L698: Correct reference "Wittle et al., 2019" => "Witte et al., 2019"

Corrected.

L880: "featured  enhanced…" – remove "with"; add "d" to "feature"

Corrected.

L881: "more larger" – remove "r" from "larger"

'r' is removed.

L896: Rcb equation – still not happy with this. I would prefer you explicitly write out "1.73x10-6 " as it is not clear whether you are saying "natural base e to -10 power" or if you are referring to engineering notation, in which case the -10 would not be superscript and the "e" would be capitalized.

We have corrected the equations to the formatting using -10 power in the manuscript and on Figure 7. For instance:

'Note that the relationship for SOCRATES in this study reveals a similar $R_{CB}$ dependence on $N_c$ but a smaller dependence on the cloud thickness than the study by Kang et al. (2024), who concluded a relationship of $R_{CB} = 1.41 \times 10^{-9} \, H_c^{3.1} N_a^{-0.8} \ldots$'

L900: remove "where is"

Removed.

L1064: remove comma between "evolution" and "during"

Removed.

---

## Author Response (AR3)

**Response to Editor**

Dear Editor, we appreciate your time and effort thoroughly reviewing our manuscript. We are truly grateful for your detailed comments and patient suggestions. We have revised the manuscript based on your comments. In the response below, your comments are in black text and our responses are in blue text.

After reading the revised manuscript, I found several unclear sentences, English mistakes, typos, and errors such as cross-referencing. Below is the list of my suggestions and questions. I don't think I found all mistakes and typos. Please do very careful reading and perform corrections and adjustments. Also, I found in some places the explanation of the physical processes mostly come from the existing studies and these explanations are not investigated by your analysis so there is no guarantee that these physical explanations hold for these IOPs. You seem to be making "sound reasonable" arguments without showing evidence. For these parts, please clearly make a separation from what your analysis shows to what other studies show and clearly state the arguments with other studies are reasonable/seems applicable but not necessarily true. I also suggest to use ACE-ENA Sum and ACE-ENA Win in the text.

Thank you for the suggestions. We have carefully and clearly made the relevant statements in the revised test to indicate the separations between what our current analysis shows and the possible hypotheses adopted from previous studies. Additionally, Dr. Tim Logan has thoroughly reviewed the manuscript and provided grammatical corrections and adjustments.

L13: are being examined => are examined
Corrected.

L16: larger number => larger number concentration
Corrected.

L17: cloud droplets => cloud droplet effective radius
Corrected.

L19: drizzle formation and growth due to => drizzle formation via droplet growth through

Corrected.

L23: sufficient water => sufficient water vapor (or moisture)

Corrected.

L24: You have not defined the aerosol-limited regime, yet. Maybe "ACE-ENA winter is in the aerosol-limited regime such that aerosols are more likely activated..." (L22-L24)

We have revised this to:

'The ACE-ENA winter season features relatively fewer aerosols, which are more likely activated into cloud droplets under the conditions of sufficient water vapor availability and strong turbulence.'

L26: water-vapor-limit regime => water-vapor-limited regime

We have revised this to:

'The enriched aerosol loading during ACE-ENA summer and SOCRATES generally leads to smaller cloud droplets competing for the limited water vapor and exhibiting a stronger ACI.'

L27: Please use more appropriate verb in stead of "pronounced" or describe how pronounced.

We have revised this to:

'…the precipitation susceptibilities are stronger during the ACE-ENA…'

L51: "more and smaller cloud droplets not only extend cloud longevity and spatial coverage" I don't think this always holds. Please clarify it.

We have changed this to:

'Furthermore, a larger number of small cloud droplets can sometimes extend cloud longevity and spatial coverage and modulate the precipitation processes in the MBL clouds, reflecting the cloud adjustments to aerosol disturbances (Albrecht, 1989; Bellouin et al., 2020)'

L57: "Aerosols have been found to suppress the precipitation frequency and strength by constantly buffering cloud droplet number concentrations via activation, hence increasing cloud

precipitation susceptibility" doesn't make sense. Buffering implies that multiple process work together to reduce the response to an aerosol perturbation but aerosol should increase cloud droplet number after it is activated. Also, how does this increase cloud precipitation susceptibility? Please clarify in plain language.

We have revised the statements to:

'Frequent aerosol intrusions in the MBL have been found to have to lower the efficiency of collision-coalescence-induced which results in the suppression of precipitation frequency and strength. Such phenomenon can be quantified and assessed via the cloud precipitation susceptibility.'

L67: cloud top => cloud-top

Corrected.

L80: CCN budgets also via => CCN budget via

Corrected.

L88: for such studies => for studying ACPIs / for studies of ACPIs

Corrected.

L94: were => was

Corrected.

L95: Remove "(ACE-ENA Sum)" and "(ACE-ENA Win)" since these are not used in other part of the text. Or use these names in stead of, e.g., ACE-ENA winter, ACE-ENA summer, winter ACE-ENA, and summer ACE-ENA in the text. The latter may be better so that you list like "SOCRATES, ACE-ENA Sum, and ACE-ENA Win" in stead of "SOCRATES, ACE-ENA winter and summer."

Throughout the manuscript, we have changed those terms to ACE-ENA Sum, and ACE-ENA Win where relevant.

L115: as well as => and

Corrected.

L117: aerosol, clouds and precipitation => aerosol, clouds, and precipitation

Corrected.

L134: numerous => many

Corrected.

L136: colder nature => colder weather/climate/atmospheric condition? near-freezing condition?

We have changed this to 'colder climate'.

L137: "compositely speaking" sounds odd to me. "our composite analysis of ... shows that" is better

We have changed this to 'our composite analysis of the synoptic pattern shows that the SOCRATES cloud…'

L139: The region of selected... => Because/Since the region of selected...

We have changed this to 'Since the region of selected...'

L162: ...(Glienke and Mei, 2000). While... => ...(Glienke and Mei, 2000), while... Or remove "while" and start with "SOCRATES..." "the" in front of "SOCRATES" is unnecessary here. This should be generally applicable throughout the text.

We have changed this to '… (Glienke and Mei, 2000). SOCRATES used a…'. And we have corrected similar circumstances in the text.

L166: will be => are

Corrected.

L207: along with => , and

Corrected.

L208: cloud base heights (zt), cloud top heights (zb) and => cloud-base heights (zb), cloud-top heights (zt), and
Corrected.

L213: g/kg => g kg^-1 (superscript)
Corrected.

L224: g/kg => g kg^-1
Corrected.

L235: which counts => which count
Corrected.

L243: ACE-ENA is => ACE-ENA are
Corrected.

L262: "separated profiles" Please make it clear.
We have changed this to '… compared to profiles which aerosols and cloud layer are separated.'

L265: decreasing profile => decreasing aerosol profile
Corrected.

L266: during summertime, will induce => during summertime, induce
Corrected.

L266: What is the typical distance between the bottom of the elevated aerosol layer and the cloud top for both ACE-ENA and SOCRATES? Why is 200 m chosen in stead of 100 m which is used by the referenced study? Some clarification is necessary here.

The 100 m threshold in Gupta et al. (2021) was used in the ORACLES campaign analysis, which might be different from our study regions, and we have added the following discussion.

'Note that from previous studies on ACE-ENA and SOCRATES, the aerosol vertical profiles within ~200 m above the cloud layers are typically found to have less variation (Wang et al., 2020; Wang et al., 2021; McCoy et al., 2021; Zhang et al., 2023), hence representing the aerosol layers in contact with the cloud. Hence, the 200 m criterion used in this study provides a sufficient sample size population for statistical analysis.'

L268: SOCRATES will significantly => SOCRATES significantly
Corrected.

L268: aerosol budget, they would need => aerosol budget, and (or which?) they need
We have changed this to 'the aerosols would need…'

L310: Figure 1a reveals => Figures 1a and 1d reveal
Corrected.

L313: 1a&b => 1a and 1b
Corrected.

L314: 1b&c => 1b and 1c
Corrected.

L320: during SOCRATES => for SOCRATES
Corrected.

L321: 1e&f => 1e and 1f
Corrected.

L322: Section 3.1 => Section ? The paragraph is in Section 3.1.
We have changed this to '…will be further discussed in the following paragraphs.'

L343: entrained down => entrained

Corrected.

L369: Bulk cloud microphysical properties distribution => Distribution of bulk cloud microphysical properties

Corrected.

L372: "The results in Figure 1 have demonstrated that aerosol/CCN sources and concentrations, especially from the sub-cloud regime, play an important role in cloud droplet formation and evolution." I don't see any discussion regarding this point in Section 3.1. Are you mentioning about L318-L320? If you are mentioning the sentences following this sentence, then adjust the sentence appropriately.

This sentence was originally used to introduce the following context about the aerosols and cloud droplets differences between the three IOPs. We have removed the sentence to avoid further confusion.

L387: "Note that the NCCN0.35% and NC values are lower than NC values during the ACE-ENA winter IOP, which is also confirmed in previous studies" I don't get this. Which NCCN and NC are you comparing with NC of ACE-ENA winter?

We have changed this to 'Note that the $N_{CCN0.35\%}$ are lower than $N_c$ values during the ACE-ENA Win, which is also confirmed in previous studies'

L406: "due to cloud-top entrainment" requires an immediate explanation. I think some text re-organization may be necessary to combine the sentences in the next paragraph. Or maybe just removing "due to cloud-top entrainment" from the sentence just work nicely, i.e., use the current paragraph to describe the general shape of the profile and use the next paragraph to give physical insights/interpretations.

Thanks for the suggestion, we have removed this statement from the sentence.

L429: Define DSD here.

Corrected.

L435: "In addition, the cloud adiabaticity is defined as ..." is not connected well to the previous sentence. Please elaborate it.

We have elaborated on this: 'Cloud adiabaticity is a key parameter as it provides insight into the degree of mixing and microphysical processes occurring within clouds. The sub-adiabatic conditions indicate that the $LWC_c$ is less than what would be expected in an adiabatic scenario, often due to processes such as in-cloud collision-coalescence and entrainment mixing (Hill et al., 2009; Braun et al., 2018; Gao et al., 2020; Wu et al., 2020b). In addition, the cloud adiabaticity is defined as $f_{ad} = LWC_c/LWC_{ad}$, where the $LWC_{ad}$ denotes adiabatic LWC (Wu et al., 2020b).'

L450: cloud top => cloud-top

Corrected.

L455: "generally weaker cloud-top inversions" In L461, the temperature jumps are listed, which should be listed here for clarity. Or move "Within the above-cloud inversion layer, the temperature (water vapor mixing ratio) differences ∆T (∆q) are 1.76 K (-1.75 g kg-1), 1.54 K (-1.66 g kg-1) and 1.48 K (-1.09 g kg-1) for SOCRATES, ACE-ENA summer and winter, respectively." right after the sentence.

We have modified the narrative structure as suggested.

L455: "stronger near-cloud top turbulence" In Fig. 5, TKE near the cloud top for ACE-ENA Win is strongest. So add a cross-reference of (Figure 5).

Cross-reference is added.

L457: cloud top => cloud-top

Corrected.

L460: How about influence on the water vapor jump? ACE-ENA Win is lowest and SOCRATES is strongest so entrained air could be dryer? It is better to use virtual potential temperature.

We have elaborated this to:

'Within the above-cloud inversion layer, the temperature (water vapor mixing ratio) differences $\Delta T$ ($\Delta q$) are 1.76 K (-1.75 g kg$^{-1}$), 1.54 K (-1.66 g kg$^{-1}$) and 1.48 K (-1.09 g kg$^{-1}$) for SOCRATES, ACE-ENA Sum, and ACE-ENA Win, respectively. The virtual potential temperature differences $\Delta\theta_V$ are 4.90 K, 5.16 K, and 3.82 K, for SOCRATES, ACE-ENA Sum, and ACE-ENA Win, respectively, indicating relatively dryer entrained airmasses during SOCRATES and ACE-ENA Sum.'

L462: Remove "Therefore".

Done.

L465: exhibited => exhibit

Corrected.

L490: droplet size distributions (DSD) => DSDs. DSD should be defined at L429.

Corrected.

L515: Move "For the four cloud portions from cloud base to cloud top, the skewness of summertime (wintertime) cloud DSDs are 0.627 (0.271), 0.358 (0.175), 0.098 (-0.063), and -0.362 (-0.554), respectively." to L508 before "Notably, the cloud DSDs..." and adjust both sentences.

We have modified the narrative structure as suggested.

L520: Make new paragraph starting with "Note that in the upper region..." and adjust the sentence.

Done. We have added the new paragraph starting with 'In the upper region of the cloud (Fig. 4a)…'

L541: m/s => m s^-1

Corrected.

L546: 5c & d => 5c and 5d

Corrected.

L554: might be slightly => is

Corrected.

L578: are elongated => would become longer (a time scale does not have width)

Corrected.

L578: growth can be => growth could be

Corrected.

L581: which disturb => which would disturb

Corrected.

L583: "As a result, the ACE-ENA..." The discussion above the sentence is all hypothesis/speculations based on past studies and there is no guarantee that these occur in ACE-ENA. Please correct it.

We have modified the discussion to 'The physical hypotheses from previous studies could potentially serve as the explanation for the phenomena that the ACE-ENA Win drizzle DSD is sufficiently broadened, and the $D_{mmd}$ is enlarged toward the cloud base.'

L630: higher => higher than ACE-ENA

Corrected.

L632: "relatively less Nccn,0.35% condition for SOCRATES" In Fig. 1f, Nccn,0.35% is largest for SOCRATES. Please clarify this.

The 'relatively less' was a typo, should be 'relatively more'.

We have corrected the discussion to 'Recall that the sub-cloud $N_{CCN,0.35\%}$ during SOCRATES is generally higher than ACE-ENA and is constituted by more small-sized aerosols (as indicated in Fig. 2b). Consequently, after activation, the lower part of the cloud exhibits a higher number of smaller cloud droplets for SOCRATES, as shown in Fig. 4d.'

L636: According to Fig.1 "under the relatively more CCN condition" is not true for ACE-ENA compared with SOCRATES. Please clarify this.

We have corrected this to '…under the relatively less CCN condition …'

L640: add appropriate references for VOCALS, CSET, ORACLES, ACTIVATE

The following references are added:

'VAMOS Ocean-Cloud-Atmosphere-Land Study (VOCALS; Wood et al., 2011), the Cloud System Evolution over the Trades (CSET; Albrecht et al., 2019), the ObseRvations of CLouds above Aerosols and their intEractionS (ORACLES; Redemann et al., 2021), and the Aerosol Cloud meTeorology Interactions oVer the western ATlantic Experiment (ACTIVATE; Sorooshian et al., 2019)'

L645: Move "The AI indices from three IOPs are ..." between "...the cloud base)." and "Note that the availability..." in L607 or move it to a better place before discussing these indices

We have modified the narrative structure as suggested.

L648: Include Fig. S5 into Fig. 6. Also state that the LWC near cloud top for the three IOPs are not comparable each other.

We have added Fig. S5 into Fig. 6, and adjusted the discussions accordingly.

'To investigate the ACI indices at the upper level of the cloud, the $N_c$ and $r_c$ at the upper cloud ($z_i > 0.8$) are plotted against the above-cloud $N_{CCN,0.35\%}$ in Figures 6c and 6d…'

and

'Note that the $LWC_c$ near the cloud top for the three IOPs are not comparable to each other, which might also induce uncertainty in the near-cloud-top ACI assessment.'

L682: near the cloud base => near the cloud base in ACE-ENA winter

Corrected.

L682: "the ACE-ENA winter featured enhanced collision-coalescence and drizzle-recirculating processes" has not been shown by data analysis. Or are you mentioning L617-L621? But it still lacks drizzle-recirculating process. Please clarify it.

We have modified the discussion to:

'As previously discussed, the ACE-ENA Win featured enhanced collision-coalescence suggested by the stronger in-cloud turbulence, and a possible drizzle-recirculating process as indicated by the previous study. And such mechanisms might explain the low $N_c$ conditions with more large drizzle drops, leading to the increase of $S_o$ values ACE-ENA Win.'

L684: "the higher ambient aerosol and CCN concentrations ... may induce effective aerosol buffering effects" is unclear. Are you saying that aerosol of SOCRATES is mainly composed of fine Aitken mode aerosol (I think you mentioned somewhere in the text), which results in smaller cloud droplets, thus collision-coalescence is not effective, which means that the warm-rain processes are suppressed? Please explain clearly.

We have modified the discussion to:

'In comparison, the aerosol of SOCRATES is largely composed of fine Aitken mode aerosol, which results in smaller cloud droplets. Thus, collision-coalescence is ineffective during SOCRATES, which leads to the relatively narrower drizzle DSDs, where the warm-rain processes are suppressed, and in turn, diminishing the sensitivity of $R_{CB}$ to $N_c$'

L695: Do you really think the difference of R^2 between 0.165 and 0.295 for SOCRATES is significant and think SOCRATES are more related to Nc? These numbers as well as correlation coefficients for S0 (L672) are generally small and I am not sure if these are statistically meaningful. How reliable are these values with uncertainty of observed data?

We have clarified the discussions to:

'The $S_0$ values are 0.979, 1.229, and 1.638, with the absolute values of correlation coefficients being 0.33, 0.29, and 0.45 for SOCRATES, ACE-ENA Sum, and ACE-ENA Win, respectively. The regression relationships are statistically significant with p<0.05 for all three IOPs.'

and

'The statistical coefficient of determination ($R^2$) values of $R_{CB}$ against $H_c$ ($N_c$) are 0.696 (0.177), 0.419 (0.212) and 0.165 (0.295), for the ACE-ENA Sum, winter and SOCRATES, respectively, suggesting that the $R_{CB}$ in ACE-ENA clouds may be more determined by $H_c$, while the $R_{CB}$ in SOCRATES clouds could be less dependent on both $H_c$ and $N_c$.'

L754: Sentences below this line are unclear and need elaboration. "Hence ..." This is unclear and need better explanation. "the time needed, ..." is also unclear to me. What budget are you talking about? The sub-cloud CCN budget? "restore the sub-cloud CCN to the budget" sounds strange since these units are different.

We have modified the discussions to:

'Hence, the retrospective period used here might quickly exceed the actual time of cloud-processing to become effective on aerosol and CCN. In other words, the actual time needed to trace back to the sub-cloud CCN concentration before they were cloud-processed, is shorter than the retrospective time tested here in Figure 8.'

L756: Thus, => This

Corrected.

L775: "Physical processing like in-cloud Brownian capture can reduce Aitken mode aerosols, while the chemical processes transform Aitken mode aerosols to larger sizes, moving them toward the accumulation mode. In addition, the in-cloud coalescence processes shift sub-cloud aerosol residuals to larger sizes, as multiple aerosols combine into a single aerosol core inside the cloud droplet during collision-coalescence" These are not shown from your analysis. These from the existing studies are used so that "the observed increase in the tail-end of the aerosol distribution for all IOPs" is somehow explained. So, these statements are speculation or can be hypothesis for future research. Please distinguish the existing studies from your findings.

We have modified the discussions to:

'According to previous studies, physical processing like in-cloud Brownian capture can reduce Aitken mode aerosols, while the chemical processes transform Aitken mode aerosols to larger sizes, moving them toward the accumulation mode. In addition, the in-cloud coalescence processes could also shift sub-cloud aerosol residuals to larger sizes, as multiple aerosols combine into a single aerosol core inside the cloud droplet during collision-coalescence. Those physical mechanisms could potentially explain the observed increase in the tail of the aerosol size distribution for all IOPs, and it will be of interest for future research to prove such hypotheses.'

L794: "which substantially enhances the collision-coalescence and the drizzle re-circulating processes" is not demonstrated by your analysis and this is another speculation.

We have modified the discussions to:

'The ACE-ENA Win clouds feature more prominent drizzle formation and evolution owing to the combined effects of relatively cleaner environment, deeper cloud layer, and slightly stronger in-cloud vertical turbulence, which is speculated to substantially enhance the collision-coalescence and the drizzle re-circulating processes, compared to the other two IOPs.'

L802: cloud base => cloud-base

Corrected.

L807: "a result of turbulence-driven in-cloud droplet interactions, especially under low NC condition" is not shown by your analysis and this is a speculation/hypothesis.

We have modified the discussions to:

'This could be possibly hypothesized as the result of turbulence-driven in-cloud droplet interactions, which could result in much higher $R_{CB}$ induced by larger drizzle drops near the cloud base for ACE-ENA, especially under low $N_c$ condition.'

L809: "The relationships established in this study indicate that ACE-ENA clouds, are largely determined by Hc, while SOCRATES clouds are more influenced by the Nc." This needs to be corrected because of small R2.

We have modified the discussions to:

'The relationships established in this study indicate that the $S_o$ in ACE-ENA clouds can be partially determined by $H_c$, while in SOCRATES clouds the $S_o$ is less influenced by $H_c$ and $N_c$.'

L810: "The combination of a deeper cloud layer and relatively lower ambient aerosol concentration, eventually leading to stronger drizzle production and evolution during ACE-ENA, especially during the winter season, results in more robust precipitation susceptibility" Even this is a sort of speculation. This has not been demonstrated by your analysis. A sound reasonable explanation doesn't mean that that is true or happening without evidence.

We have modified the discussions to:

'Based on the physical mechanisms found in the previous study, a possible hypothesis can be leveraged to explain the observed results. That is, the combination of a deeper cloud layer and relatively lower ambient aerosol concentration, eventually leading to stronger drizzle production and evolution during ACE-ENA, especially during the winter season, results in more robust precipitation susceptibility. And further numerical simulations and experiments are warranted to prove this hypothesis.'

Fig. 1: "ACE-ENA summer, winter and SOCRATES" => "ACE-ENA Sum, ACE-ENA Win, and SOCRATES"; "pink, purple and green" => "pink, purple, and green"
Corrected.

Fig. 2: "Accumulation mode, Aitken mode and Coarse mode" => "Accumulation mode, Aitken mode, and Coarse mode"; "ACE-ENA summer, winter and SOCRATES" => "ACE-ENA Sum, ACE-ENA Win, and SOCRATES"; "pink, purple and green" => "pink, purple, and green"
Corrected.

Fig. 3: "ACE-ENA summer, winter and SOCRATES" => "ACE-ENA Sum, ACE-ENA Win, and SOCRATES"; "pink, purple and green" => "pink, purple, and green"
Corrected.

Fig. 4: "ACE-ENA summer, winter and SOCRATES" => "ACE-ENA Sum, ACE-ENA Win, and SOCRATES"; "pink, purple and green" => "pink, purple, and green"

Corrected.

Fig. 5: "u'2 (c) and v'2" => "u'2 (c), and v'2": "ACE-ENA summer, winter and SOCRATES" => "ACE-ENA Sum, ACE-ENA Win, and SOCRATES"; "pink, purple and green" => "pink, purple, and green"

Corrected.

Fig. 6: Add label (a) and (b). Include Fig. S5 here; "ACE-ENA summer, winter and SOCRATES" => "ACE-ENA Sum, ACE-ENA Win, and SOCRATES"; "pink, purple and green" => "pink, purple, and green"

Corrected.

Fig. 7: "ACE-ENA summer, winter and SOCRATES" => "ACE-ENA Sum, ACE-ENA Win, and SOCRATES"; "pink, purple and green" => "pink, purple, and green"

Corrected.

---

## Author Response (AR4)

**Response to Editor**

Dear Editor, we appreciate your time and effort for the additional check of our manuscript. We are truly grateful for your suggestions on the technical corrections. We have revised the manuscript based on your comments. In the response below, your comments are in black text, and our responses are in blue text.

I think the manuscript will be ready for publication after some technical corrections listed below:
- line 57: Remove "have to" so that "... aerosol intrusions ... have been found to lower the efficiency..."

Corrected.

- line 57: "collision-coalescebce-induced" is not a noun so "the efficiency of collision-coalescence-induced which results in..." is grammatically wrong. I guess "-induced" should be removed???

Corrected.

- line 262: "profiles which aerosols and cloud layer are separated" should be "profiles in which aerosols and cloud layer are separated"

Corrected.

- line 305: FT is not defined yet. Or since FT is used at only 4 locations, use either "free troposphere" (noun) or "free tropospheric" (adjective), e.g., free tropospheric aerosol.

We have changed all the FT to "free troposphere" or "free tropospheric".

- line 314: "Figure 1a and 1d reveals" => "Figures 1a and 1d reveal"

Corrected.